# Instance-Dependent Regret Bounds
# for Nonstochastic Linear Partial Monitoring

**Federico Di Gennaro**[*,†]
EPFL, Lausanne, Switzerland
digennarof.00@gmail.com

**Khaled Eldowa**[*,‡]
Università degli Studi di Milano, Milan, Italy
& Politecnico di Milano, Milan, Italy
khaled.eldowa3@gmail.com

**Nicolò Cesa-Bianchi**
Università degli Studi di Milano, Milan, Italy
& Politecnico di Milano, Milan, Italy
nicolo.cesa-bianchi@unimi.it

## Abstract

In contrast to the classic formulation of partial monitoring, linear partial monitoring can model infinite outcome spaces, while imposing a linear structure on both the losses and the observations. This setting can be viewed as a generalization of linear bandits where loss and feedback are decoupled in a flexible manner. In this work, we address a nonstochastic (adversarial), finite-actions version of the problem through a simple instance of the exploration-by-optimization method that is amenable to efficient implementation. We derive regret bounds that depend on the game structure in a more transparent manner than previous theoretical guarantees for this paradigm. Our bounds feature instance-specific quantities that reflect the degree of alignment between observations and losses, and resemble known guarantees in the stochastic setting. Notably, they achieve the standard $\sqrt{T}$ rate in easy (locally observable) games and $T^{2/3}$ in hard (globally observable) games, where $T$ is the time horizon. We instantiate these bounds in a selection of old and new partial information settings subsumed by this model, and illustrate that the achieved dependence on the game structure can be tight in interesting cases.

## 1 Introduction

Partial monitoring models sequential decision-making problems with general feedback structures. A generic nonstochastic partial monitoring problem is characterized by an action space $\mathcal{A}$, an outcome space $\mathcal{Z}$, an observation space $\Sigma$, a loss function $Y \colon \mathcal{A} \times \mathcal{Z} \to \mathbb{R}$, and an observation function $\Phi \colon \mathcal{A} \times \mathcal{Z} \to \Sigma$. All of these components are known to the learner, who interacts with an unknown environment in a series of $T$ rounds. In each round $t$, the learner picks an action $A_t \in \mathcal{A}$, observes $\Phi(A_t, Z_t)$, and incurs the (unobserved) loss $Y(A_t, Z_t)$, where $Z_t \in \mathcal{Z}$ is a latent outcome. The learner aims to minimize the regret, defined as the difference between the cumulative losses of played actions and the cumulative losses incurred by the best action in hindsight. This general decoupling between losses and observations results in an extraordinary modeling power. Not only does this framework encompass full-information scenarios [23, 13], where the latent outcome can be inferred from the observation, and bandit scenarios [4, 11], where the loss and the observation coincide;

---

[*]Equal contribution.

[†]Now at ETH Zürich, Zürich, Switzerland.

[‡]Now at Univ. Grenoble Alpes, Inria, CNRS, Grenoble INP, LJK; Grenoble, France.

39th Conference on Neural Information Processing Systems (NeurIPS 2025).

it can model problems that interpolate between these two extremes, like feedback graphs [2], and problems that go beyond bandit feedback, like dynamic pricing [14]. The primary object of study in this framework is how the structure of the game affects its difficulty, measured via the notion of minimax regret (the best achievable worst-case regret).

When $\mathcal{A}$ and $\mathcal{Z}$ are both finite, one recovers the well-studied *finite* partial monitoring problem. An extensive line of work over the last few decades [48, 14, 3, 20, 6, 38, 39] has shown that all finite games can be classified into one of four classes, each characterized by the achievable minimax rate in terms of $T$, which can only be $\Omega(T)$, $\Theta(T^{2/3})$, $\Theta(T^{1/2})$, or $0$. These classes, respectively, consist of: hopeless games, hard—or (strictly) globally observable—games, easy—or locally observable— games, and trivial games. What delineates these categories are precise 'observability' conditions on the loss and observation functions. An important work in our context is that of Lattimore and Szepesvári [40], who introduced the *exploration-by-optimization* (EXO) method (see Section 3 for a detailed description) and achieved—via an intuitive and efficient algorithm—the best known rates for the problem, free from arbitrary and redundant game-dependent constants that appear in prior results.

Nevertheless, the restriction to finite outcome spaces remains a limitation of this model. On the other hand, the *linear* partial monitoring framework [42, 15, 31, 32] allows infinite outcomes, but restricts that, for a given outcome, the loss and observation functions are linear in some representation of the actions. This framework still encompasses a broad spectrum of problems (see Section 2), and arguably, provides a more intuitive observation structure, free from the combinatorial complications of the finite model (which can nonetheless be made to fit within the linear framework [32]). A simpler point of view sees this model as a generalization of the standard linear bandit problem [17, 10], which corresponds to the case when the loss and observation functions coincide. Kirschner et al. [31, 32] studied an instance of this model in the stochastic setting (where the hidden outcome is fixed over the rounds and the learner receives noisy observations), adopting a version of the information-directed sampling [49] algorithm. Remarkably, they showed that the classification theorem continues to hold in the linear setting with analogous observability criteria. Also of note is that their bounds feature certain *alignment* constants that reflect the difficulty of the game within its respective class in an arguably more interpretable manner compared to finite partial monitoring results.

In the nonstochastic (adversarial) setting, a linear partial monitoring problem was studied by Lattimore and Gyorgy [37], who also adopted the EXO approach. In fact, their setup is more general than what we described above since their observation function is taken as arbitrary. Nevertheless, they established an elegant result bounding the regret of a general EXO policy in terms of a generalized version of the information ratio [49], which characterizes the Bayesian regret of information-directed sampling. As such, however, their bounds are still of a largely opaque nature in terms of their dependence on problem-specific parameters. Moreover, in its general form, implementing their EXO algorithm involves solving a daunting optimization problem (over a possibly infinite dimensional space) at every step in order to compute a loss estimation function and a sampling distribution. Hence, though constructive, their approach serves more as a template than a concrete implementable policy. An expanded discussion on related works is provided in Appendix A.

## 1.1  Contributions

We address a finite-actions version of adversarial linear partial monitoring through a simple instance of the EXO method. Outlined in Section 3, this policy builds upon an exponential weights update and imposes a simple structure on the adopted loss estimator, which is made possible thanks to the linearity of the observation function. As a result, the aforementioned optimization problem is reduced to one of minimizing a convex objective over probability distributions over the actions, subject to a certain convex constraint. This problem is efficiently solvable against many loss (outcome) spaces of interest. Under the observability conditions identified by Kirschner et al. [31] in the stochastic setting, we prove new regret bounds for this policy in Sections 4 and 5 for locally and globally observable games respectively; the former of order $\sqrt{T}$ and the latter of order $T^{2/3}$, as one would expect.

The proofs rely on bounding the optimal value of the optimization program alluded to above via an application of the minimax theorem, following the analysis of EXO by Lattimore and Szepesvári [40] in the finite setting. However, to better exploit the linear structure of our setting, we employ new arguments, including a more natural way of constructing exploration distributions in locally observable games. The resulting bounds feature interpretable instance-dependent quantities, similar to the alignment constants of Kirschner et al. [31, 32], that gauge the difficulty of the game within its

observability class. We then illustrate the versatility of the bounds by instantiating them in a selection of both locally and globally observable games, proving their near-optimality in some cases.

## 2 Preliminaries

**Notation.** For a positive integer $d$, $[d]$ denotes the set $\{1, \ldots, d\}$. We use the Iverson bracket notation $\llbracket \cdot \rrbracket$ to denote the indicator function. For a set $S \subseteq \mathbb{R}^d$, $\mathrm{co}(S)$ denotes its convex hull, and $\mathrm{ext}(S)$ the set of its extreme points. Moreover, its (absolute) polar set is defined as $S^\circ := \{x \in \mathbb{R}^d \mid \sup_{s \in S} |\langle x, s \rangle| \leq 1\}$. For a real matrix $X$, $\mathrm{tr}(X)$, $\mathrm{rk}(X)$, $\mathrm{col}(X)$, $X^\dagger$, and $\|X\|$ denote, respectively, its trace, rank, column space, Moore–Penrose inverse, and operator norm (induced by the Euclidean $L_2$-norm). For $p \in [0, \infty]$, let $\mathcal{B}_p(r)$ be the centered $L_p$ ball of radius $r > 0$. We use $\mathbf{1}_d$ and $\boldsymbol{I}_d$ to denote, respectively, the $d$-dimensional vector with all entries equal to 1 and the $d \times d$ identity matrix. The probability simplex in $\mathbb{R}^d$ is denoted by $\Delta_d := \{x \in [0, 1]^d \mid \|x\|_1 = 1\}$. Given a finite index set $Z$ and a collection of matrices $\{X_z \mid z \in Z\}$, all with the same number of rows, we denote by $\mathrm{span}\{X_z \mid z \in Z\}$ the span of all their columns.

### 2.1 Problem setting

Let $\mathcal{A} := [k]$ denote the action set, where $k$ is a positive integer. We associate every action $a \in \mathcal{A}$ with a feature vector $\psi_a \in \mathbb{R}^d$ and an observation matrix $M_a \in \mathbb{R}^{d \times n(a)}$, where $n(\cdot)$ maps every action to a positive integer. We will use $\mathcal{X}$ to denote the set $\{\psi_a \mid a \in \mathcal{A}\}$. Let $\mathcal{L} \subset \mathbb{R}^d$ be a convex and compact loss space, which, in our setting, is synonymous with the outcome space $\mathcal{Z}$ as used in the introduction. We assume here that $\mathcal{L}$ contains a centered $L_2$ ball $\mathcal{B}_2(r)$ of some radius $r > 0$. This simplifies the observation structure of the game and suffices to subsume typical cases such as when $\mathcal{L}$ represents the absolute polar set of $\mathcal{X}$ or a ball of bounded norm. We study the following game between a learner and an oblivious adversary over $T$ rounds. Before the game starts, the adversary secretly chooses a sequence of loss vectors $(\ell_t)_{t \in [T]}$, where $\ell_t \in \mathcal{L}$. Then, at every round $t \in [T]$, the learner: i) selects (possibly at random) an action $A_t \in \mathcal{A}$; ii) observes the signal $\phi_t := M_{A_t}^\top \ell_t \in \mathbb{R}^{n(A_t)}$, which, in general, is distinct from the incurred (but unobserved) loss $\psi_{A_t}^\top \ell_t \in \mathbb{R}$. The feature mapping and the observation matrices are known to the learner. For brevity, let $y_t(a) := \psi_a^\top \ell_t$ for every $a \in \mathcal{A}$. In the notation used in the introduction, one can see that for $a \in \mathcal{A}$ and $\ell \in \mathcal{L}$, the loss map in the setting becomes $Y(a, \ell) = \psi_a^\top \ell$, while the observation map becomes $\Phi(a, \ell) = M_a^\top \ell$.

For round $t \in [T]$, let $\mathcal{H}_t := (A_s, \phi_s)_{s=1}^t$ denote the interaction history up to the end of the round, and let $\mathcal{F}_t := \sigma(\mathcal{H}_t)$ denote the $\sigma$-algebra generated by $\mathcal{H}_t$. The learner's policy can be represented via a sequence $(\pi_t)_{t \in [T]}$ of probability kernels such that $\pi_t$ maps the history $\mathcal{H}_{t-1}$ to a distribution over the actions, from which $A_t$ is to be sampled. The learner's goal is to minimize the following standard notion of regret

$$R_T := \mathbb{E}\left[\sum_{t=1}^T y_t(A_t)\right] - \min_{a \in \mathcal{A}} \sum_{t=1}^T y_t(a), \tag{1}$$

where the expectation is with respect to the internal randomness of the algorithm. The dependence of the regret on the learner's strategy $(\pi_t)_t$ and the sequence of losses $(\ell_t)_t$ is suppressed from the notation for brevity. The minimax regret is defined as $R_T^* := \inf_{(\pi_t)_t} \sup_{(\ell_t)_t} R_T$. where the inf is over all (possibly randomized) learning algorithms and the sup is over all sequences of $T$ loss vectors from $\mathcal{L}$. Similarly to [31, 32], we impose a standard boundedness assumption throughout this work:

**Assumption 1.** *It holds that* $\max_{a, b \in \mathcal{A}, \ell \in \mathcal{L}} |(\psi_a - \psi_b)^\top \ell| \leq 2$.

As a notational remark, we define $\mathbb{E}_t[\cdot] := \mathbb{E}[\cdot \mid \mathcal{F}_{t-1}]$, with $\mathcal{F}_0$ being the trivial $\sigma$-algebra. Moreover, throughout the paper, we treat functions over $\mathcal{A}$ as vectors in $\mathbb{R}^k$. In particular, for $b \in \mathcal{A}(= [k])$, the indicator vector $e_b \in \mathbb{R}^k$ corresponds to the function $a \mapsto \llbracket a = b \rrbracket$ over $\mathcal{A}$.

### 2.2 Observability conditions and lower bounds

Similarly to the finite partial monitoring setting, the observation structure of the game characterizes the achievable regret. Before we proceed, we introduce some relevant concepts. An action $a \in \mathcal{A}$

is called *Pareto optimal* if $\psi_a \in \text{ext}(\text{co}(\mathcal{X}))$; i.e., $\psi_a$ is an extreme point of the convex hull of $\mathcal{X}$. Let $\mathcal{A}^* \subseteq \mathcal{A}$ satisfy that $\text{ext}(\text{co}(\mathcal{X})) = \{\psi_a \mid a \in \mathcal{A}^*\}$ and contain no two actions with the same features. Additionally, let $k^* := |\mathcal{A}^*|$. This set is well-defined since $\text{ext}(\text{co}(\mathcal{X})) \subseteq \mathcal{X}$; moreover, it is the smallest subset of $\mathcal{A}$ that satisfies $\min_{a \in \mathcal{A}} \langle \psi_a, \ell \rangle = \min_{a^* \in \mathcal{A}^*} \langle \psi_{a^*}, \ell \rangle$ for every $\ell \in \mathbb{R}^d$. The set of actions that are optimal against a loss vector $\ell \in \mathbb{R}^d$ is denoted by $\mathcal{P}(\ell) := \{ a \in \mathcal{A} \mid \langle \psi_a - \psi_b, \ell \rangle \leq 0 \text{ for all } b \in \mathcal{A} \}$, which extends to sets $C \subseteq \mathbb{R}^d$ as $\mathcal{P}(C) := \cup_{\ell \in C} \mathcal{P}(\ell)$. We state below the observability conditions that determine the difficulty of the game (in terms of the scaling of the regret with $T$) as identified by Kirschner et al. [31] in the stochastic setting.

**Definition 1** (Observability conditions [31]). *A game is called* globally observable *if*

$$\psi_a - \psi_b \ \in \ \text{span}\{M_c \mid c \in \mathcal{A}\} \quad \text{for all } a, b \in \mathcal{A}. \tag{2}$$

*Further, a game is called* locally observable *if for every convex set $C \subseteq \mathbb{R}^d$,*

$$\psi_a - \psi_b \ \in \ \text{span}\{M_c \mid c \in \mathcal{P}(C)\} \quad \text{for all } a, b \in \mathcal{P}(C). \tag{3}$$

Note that the second condition implies the first by taking $C$ as $\mathbb{R}^d$ (or simply $\{\mathbf{0}\}$). Intuitively, in a globally observable game, one can use the observations to estimate the difference between the losses of any pair of actions. In locally observable games, one can do so efficiently; only using observations from actions that perform at least as good as the pair of interest. (See also the equivalent formulation provided in Lemma 4 below, or Lemma 5 in Appendix B.) Kirschner et al. [32] refined these conditions to account for cases when $\mathcal{L}$ is an arbitrary (and possibly low-dimensional) set. We do not pursue this direction here; as mentioned in the problem setting, we focus on the case when $\mathcal{L}$ has a non-empty interior (in particular, we assumed it contains a centered $L_2$ ball), rendering the conditions above sufficient to derive the following lower bounds.

**Proposition 1.** *Assume that the game has at least two non-duplicate Pareto optimal actions, and that $\mathcal{B}_2(r) \subseteq \mathcal{L}$ for some $r > 0$. Then, the minimax regret $R_T^*$ is $\widetilde{\Omega}(\sqrt{T})$ if the game is locally observable, $\widetilde{\Omega}(T^{2/3})$ if it is globally but not locally observable, and $\Omega(T)$ otherwise.*

These bounds are obtained in the same manner as similar results in the finite outcomes setting and the linear stochastic setting [31], see Appendix J. However, one must deal with some technical nuisances since in our case, the stochastic setting is not immediately subsumed by the adversarial one; besides boundedness issues, the manner in which noise is added to the observations can make the relationship a bit subtle, refer to [35, Chapter 29] for details. Over the coming section, we will derive regret upper bounds with matching rates and interpretable game-dependent constants for an EXO-based policy.

## 2.3  Examples

Owing to its flexibility, this setting can model many online learning problems, classic and contrived, as we now elaborate. In all these examples, $\mathcal{L}$ is typically taken as $\mathcal{X}^\circ$, which is equivalent to $\mathcal{B}_\infty(1)$ when $\psi_a = e_a$ for all $a \in \mathcal{A}$.

**Full information.** In this game, also known as prediction with expert advice [13], we have that $d = k$, $\psi_a = e_a$, and $M_a = \boldsymbol{I}_k$ for all $a \in \mathcal{A}$. Hence, the learner observes the entire loss vector upon playing any action, and the game is clearly locally observable.

**Feedback graphs.** An instance of this game [44, 1, 2] is characterized by a graph $\mathcal{G} = (\mathcal{A}, E)$ over the actions (we focus on undirected graphs). As in the previous game, $d = k$ and $\psi_a = e_a$; however, $M_a$ only has the vectors $(e_b)_{b \in N_{\mathcal{G}}(a)}$ as columns, where $N_{\mathcal{G}}(a)$ is the neighborhood of $a$ (which need not include $a$ itself). Globally observable games correspond to *weakly* observable graphs, where the loss of every action is observable; while locally observable games correspond to *strongly* observable graphs, where each action either has a self-loop, is connected to all other actions, or both.

**Linear bandits.** Adversarial linear bandits [17, 10] (with a finite action set) is recovered by simply fixing $M_a = \psi_a$ for all $a \in \mathcal{A}$. This is easily verifiable to be a locally observable game, which reduces to a standard bandit problem when $\psi_a = e_a$.

**Linear dueling bandits.** This game is a quantitative variation of the dueling bandits problem. It was proposed in [31, 32] in the stochastic linear partial monitoring setting, extending the finite-outcomes formulation in [24]. Here, the action set has the form $\mathcal{A} = [m] \times [m]$ for some positive integer $m$. Given a feature mapping $a \mapsto \psi_a$ from $[m]$ to $\mathbb{R}^d$, we extend it to $\mathcal{A}$ via $(a, b) \mapsto \psi_{a,b} := \psi_a + \psi_b$.

However, the observations are given by $M_{a,b} = \psi_a - \psi_b$; hence, the learner receives relative feedback. This game can also be shown to be locally observable [31, 32].

**Bandits with ill-conditioned observers.** Modifying a problem from [18], consider a game where $d = k$, $\psi_a = e_a$, and $M_a = (1 - \varepsilon)\mathbf{1}_k/k + \varepsilon e_a$ for some $\varepsilon \in (0, 1]$, such that $\varepsilon = 1$ recovers standard bandits. Note that the observations become, in a sense, less informative as $\varepsilon$ decreases. Nevertheless, the game is always locally observable as $e_a - e_b = {}^1\!/\varepsilon(M_a - M_b)$.

**Composite graph feedback.** We describe now another variant of the graph feedback problem, still characterized by an undirected graph $G = (\mathcal{A}, E)$, which we now assume to contain all self-loops. Here, again, $d = k$ and $\psi_a = e_a$; however, upon playing an action, the learner observes the *average* of the losses of its neighbors (which includes the played action itself). That is, if $A$ is the adjacency matrix of the graph and $D$ is the degree matrix, then $M_a$ is the $a$-th column of $AD^{-1}$. This is less informative than standard graph feedback, and draws inspiration from problems studied in [57, 25]. Global observability here corresponds to $AD^{-1}$ being invertible, whereas the only locally observable graph is the one corresponding to standard bandits, see Appendix I for more details.

## 3 Exploration-by-Optimization, the Linear Case

We describe in this section a simple instantiation of the EXO approach [40, 37] suited to our setting, resulting in a policy amenable to efficient implementation. We show in later sections that this policy suffices to obtain tight bounds, even if its specificity excludes it from enjoying the generic guarantees provided by Lattimore and Gyorgy [37] and Lattimore [36] for general EXO. The predictions made by this policy revolve around the predictions of an instance of the exponential weights algorithm, restricted over the set $\mathcal{A}^*$ of Pareto optimal actions (without duplicates). When executed against a sequence of loss functions $(\widehat{y}_t)_{t \in [T]}$ with $\widehat{y}_t \colon \mathcal{A} \to \mathbb{R}$, the exponential weights algorithm outputs at round $t$ a distribution $q_t \in \Delta_k$ given by

$$q_t(a) = \frac{[\![a \in \mathcal{A}^*]\!] \exp\!\big(-\eta \sum_{s=1}^{t-1} \widehat{y}_s(a)\big)}{\sum_{a' \in \mathcal{A}^*} \exp\!\big(-\eta \sum_{s=1}^{t-1} \widehat{y}_s(a')\big)} \,, \tag{4}$$

where $\eta > 0$ is a learning rate parameter. This would have been a sound approach on its own were we able to access the loss functions $(y_t)_t$ directly. Instead, we are to estimate these loss functions, which, in general, requires carefully exploring the actions. Incurring minimal loss while doing so, to the extent allowed by the structure of the game, is the essential goal of the EXO method. Towards making this tradeoff more explicit, the following lemma provides a generic regret bound for any policy by pivoting around the predictions $(q_t)_t$ produced against some sequence $(\widehat{y}_t)_t$ of surrogate loss functions satisfying certain conditions. (Proofs of this section's results are in Appendix D.)

**Lemma 1.** *Fix a learning policy and define $p_t$ as the law of $A_t$ conditioned on $\mathcal{F}_{t-1}$. Let $q_t$ be as given in (4) for some learning rate $\eta > 0$ and sequence of surrogate loss functions $(\widehat{y}_t)_t$ such that $\widehat{y}_t \in \mathbb{R}^k$ is $\mathcal{F}_t-$measurable and satisfies $\max_{a \in \mathcal{A}} |\eta \widehat{y}_t(a)| \leq 1$. Further, let $a^* \in \arg\min_{a \in \mathcal{A}^*} \sum_{t=1}^T y_t(a)$. Then, the regret of the policy satisfies*

$$R_T \leq \frac{\log k}{\eta} + \sum_{t=1}^T \mathbb{E}\left[\langle p_t - q_t, y_t\rangle + \eta\langle q_t, \mathbb{E}_t \widehat{y}_t^2\rangle + \langle q_t - e_{a^*}, y_t - \mathbb{E}_t \widehat{y}_t\rangle\right].$$

While not particularly insightful, results of this form nonetheless motivate the EXO approach, where one picks the predictions and the surrogate loss functions by solving an optimization problem at every step, seeking to minimize an upper bound on the second term in the regret guarantee of Lemma 1. More precisely, at every round $t$, reliant on the observed history thus far, one decides on a prediction $p_t \in \Delta_k$ and a function $g_t \colon \mathcal{A} \times \mathbb{R} \to \mathbb{R}^k$ so as to set $\widehat{y}_t = g_t(A_t, \phi_t)$, assuming $||g_t(A_t, \phi_t)||_\infty \leq {}^1\!/\eta$. To make the dependence on $\ell_t$ more explicit, we denote by $H$ the $d \times k$ matrix with $\{\psi_a\}_{a \in \mathcal{A}}$ as columns. In particular, this enables us to write $y_t = H^\top \ell_t$. Considering the $t$-th summand in the second term in the bound of Lemma 1, the term inside the expectation becomes

$$\langle p_t - q_t, H^\top \ell_t\rangle + \eta\big\langle q_t, \textstyle\sum_a p_t(a) g_t(a, M_a^\top \ell_t)^2\big\rangle + \big\langle q_t - e_{a^*}, H^\top \ell_t - \textstyle\sum_a p_t(a) g_t(a, M_a^\top \ell_t)\big\rangle, \tag{5}$$

where $g_t(a, M_a^\top \ell_t)^2$ is the entry-wise square of $g_t(a, M_a^\top \ell_t)$. The general strategy is to optimize jointly in $p_t$ and $g_t$ the expression above, maximized over the adversary's choice of $\ell_t$. Besides some

small deviations, this still aligns closely with the approach of Lattimore and Gyorgy [37].[1] Now, however, exploiting the structure of our setting, we sidestep the demanding task of optimizing over all possible loss estimators by committing to a simple structure commonly used in (adversarial) linear bandit algorithms [10, 35], modified slightly to suit our partial monitoring setup.

Before proceeding in that direction, we introduce some notation. Let $M$ denote the $d \times (\sum_a n(a))$ matrix obtained by stacking horizontally the observation matrices of all the actions; such that $\mathrm{span}\{M_a \mid a \in \mathcal{A}\} = \mathrm{col}(M)$. For any $\pi \in \Delta_k$, we define $Q(\pi) := \sum_{a \in \mathcal{A}} \pi(a) M_a M_a^\top$. Moreover, with $\delta \in (0, 1/2]$, let $Q_\delta(\pi) := Q((1-\delta)\pi + \delta \mathbf{1}_k/k)$. This forces the argument of $Q$ to be bounded away from zero, implying that $\mathrm{col}(Q_\delta(\pi)) = \mathrm{col}(M)$ for any $\pi \in \Delta_k$ (see Lemma 8 in Appendix C), which is a convenient property for the analysis. Still, $Q_\delta(\pi)$ need not be invertible as we allow that $\mathrm{rk}(M) < d$; hence, the ensuing presentation will feature quantities of the form $Q(\pi)^\dagger$ and $Q_\delta(\pi)^\dagger$. Returning to our task, the following lemma specifies a certain form for the pair $(p_t, g_t)$ and provides the resulting regret bound under global observability.

**Lemma 2.** *In the same setting as Lemma 1, let the predictions of the policy satisfy $p_t = (1-\delta)\widetilde{p}_t + \delta \mathbf{1}_k/k$ for some $\delta \in (0, 1)$ and $\widetilde{p}_t \in \Delta_k$, and let the surrogate loss functions satisfy $\widehat{y}_t = g_t(A_t, \phi_t)$ where $g_t(a, \phi) = (I_k - \mathbf{1}_k q_t^\top) H^\top Q(p_t)^\dagger M_a \phi$. Then, assuming global observability, it holds that*

$$
R_T \leq \frac{\log k}{\eta} + 2\delta T + \sum_{t=1}^T \mathbb{E}\left[ \langle \widetilde{p}_t - q_t, H^\top \ell_t \rangle + \eta \omega^2 \sum_{a,b \in \mathcal{A}} q_t(a) q_t(b) (\psi_a - \psi_b)^\top Q_\delta(\widetilde{p}_t)^\dagger (\psi_a - \psi_b) \right]
$$

*provided that $\max_{a,b \in \mathcal{A}} (\psi_a - \psi_b)^\top Q_\delta(\widetilde{p}_t)^\dagger (\psi_a - \psi_b) + \max_{c \in \mathcal{A}} \|M_c^\top Q_\delta(\widetilde{p}_t)^\dagger M_c\|_2 \leq \frac{2}{\eta \omega}$, where*

$$
\omega := \sup_{a,b,c \in \mathcal{A}, \, p \in \Delta_k, \, \ell \in \mathcal{L}} \frac{|(\psi_b - \psi_c)^\top Q_\delta(p)^\dagger M_a M_a^\top \ell|}{\|M_a^\top Q_\delta(p)^\dagger (\psi_b - \psi_c)\|} \leq \max_{a \in \mathcal{A}, \ell \in \mathcal{L}} \|M_a^\top \ell\|.
$$

Here, $Q(p_t)^\dagger M_{A_t} \phi_t =: \widehat{\ell}_t$ is an estimator of $\ell_t$ that is essentially unbiased.[2] Instead of taking the loss estimate of action $a \in \mathcal{A}$ as $\psi_a^\top \widehat{\ell}_t$ as common in linear bandit algorithms, we use a shifted version $(\psi_a - Hq_t)^\top \widehat{\ell}_t$ anchored about $Hq_t = \sum_{b \in \mathcal{A}} q_t(b) \psi_b$. Shifting the loss estimators in this manner (via any anchor in $\mathrm{co}(\mathcal{A})$) does not alter the resulting $q_t$; however, its effect is that the last term in the bound of Lemma 2 (known as the variance term) is expressed in terms of feature *differences*. This is crucial if one is to take advantage of either observability condition, since they only relate the observation matrices (in terms of which $Q_\delta(\cdot)$ is defined) to feature differences. Specifically using $Hq_t$ as the anchor is, in a certain sense, an optimal choice, see Lemmas 13 and 14 in Appendix C.

With the structures imposed upon $g_t$ and $p_t$ in Lemma 2, we can now reduce our task of minimizing the regret bound to solving a (constrained) optimization problem at every step $t$ (depending on the history only through $q_t$) over the distribution $\widetilde{p}_t \in \mathbb{R}^k$. In the following, let $L$ be a free parameter satisfying $L \geq \omega$. Define the function $\mathcal{E}: \mathcal{A} \times \mathcal{A} \times \Delta_k \to \mathbb{R}$ as $\mathcal{E}(a, b; p) := (\psi_a - \psi_b)^\top Q_\delta(p)^\dagger (\psi_a - \psi_b)$, and for every pair $q \in \Delta_k$ and $\eta > 0$, define the function $\Lambda_{\eta,q}: \Delta_k \times \mathcal{L} \to \mathbb{R}$ as[3]

$$
\Lambda_{\eta,q}(p, \ell) := \frac{1}{\eta} \langle p - q, H^\top \ell \rangle + L^2 \sum_{a,b \in \mathcal{A}} q(a) q(b) \mathcal{E}(a, b; p).
$$

Next, define $z: \Delta_k \to \mathbb{R}$ as $z(p) := \max_{a,b \in \mathcal{A}} \mathcal{E}(a, b; p) + \max_{c \in \mathcal{A}} \|M_c^\top Q_\delta(p)^\dagger M_c\|_2$ and its sub-level set $\Xi_\eta := \{p \in \Delta_k : z(p) \leq 2/\eta L\}$. Finally, let

$$
\Lambda_{\eta,q}^* := \min_{p \in \Xi_\eta} \max_{\ell \in \mathcal{L}} \Lambda_{\eta,q}(p, \ell) \qquad \text{and} \qquad \Lambda_\eta^* := \sup_{q \in \Delta_k^*} \Lambda_{\eta,q}^*,
$$

where $\Delta_k^* = \{q \in \Delta_k \mid q(a) = 0 \, \forall a \notin \mathcal{A}^*\}$. Algorithm 1 summarizes the policy we have arrived at: at round $t$, given $q_t$, we choose a distribution $\widetilde{p}_t$ satisfying $z(\widetilde{p}_t) \leq 2/\eta L$ and $\max_{\ell \in \mathcal{L}} \Lambda_{\eta,q_t}(\widetilde{p}_t, \ell) \leq$

---

[1]The core algorithm here corresponds to the FTRL algorithm run on the probability simplex over the actions, while they use FTRL with a general potential over the convex hull of the action set. The two approaches can be related, see [30, Section 5.2]. Moreover, we use a second order bound on the Bregman divergence (or stability) term, which introduces the constraint over the norm of the loss estimators.

[2]Its expectation is the projection of $\ell_t$ onto $\mathrm{col}(M)$.

[3]As an alternative form, Lemma 13 in Appendix C implies that $\Lambda_{\eta,q}(p, \ell) = (1/\eta)\langle p - q, H^\top \ell \rangle + 2L^2 \sum_{a \in \mathcal{A}} q(a)(\psi_a - Hq)^\top Q_\delta(p)^\dagger (\psi_a - Hq)$.

---
**Algorithm 1** Anchored Exploration-by-Optimization
---
1: **input:** learning rate $\eta > 0$, stability parameter $\delta \in (0, 1/2]$, sub-optimality tolerance $\varepsilon \geq 0$, scale parameter $L \geq \omega$
2: **initialize:** $\widehat{y}_0 = \mathbf{0}$
3: **for** $t = 1, \ldots, T$ **do**
4: $\quad \forall a \in \mathcal{A}$, set $q_t(a) \propto [\![a \in \mathcal{A}^*]\!] \exp\left(-\eta \sum_{s=1}^{t-1} \widehat{y}_s(a)\right)$
5: $\quad$ choose $\widetilde{p}_t \in \Delta_k$ such that $\max_{\ell \in \mathcal{L}} \Lambda_{\eta,q_t}(\widetilde{p}_t, \ell) \leq \Lambda_\eta^* + \varepsilon$ and $z(\widetilde{p}_t) \leq \frac{2}{\eta L}$
6: $\quad$ set $p_t = (1 - \delta)\widetilde{p}_t + \delta \mathbf{1}_k/k$
7: $\quad$ execute $A_t \sim p_t$ and observe $\phi_t = M_{A_t}^\top \ell_t$
8: $\quad \forall a \in \mathcal{A}$, set $\widehat{y}_t(a) = (\psi_a - Hq_t)^\top Q(p_t)^\dagger M_{A_t} \phi_t$ $\quad \triangleright$ or simply, $\widehat{y}_t(a) = \psi_a^\top Q(p_t)^\dagger M_{A_t} \phi_t$
9: **end for**
---

$\Lambda_\eta^* + \varepsilon$, for some tolerance $\varepsilon > 0$. In the following round, $q_{t+1}$ is computed via the exponential weights update in (4) with $\widehat{y}_t$ as given in Lemma 2. As shown in Appendix D (in the proof of Lemma 3 below), the functions $z(\cdot)$ and $\max_{\ell \in \mathcal{L}} \Lambda_{\eta,q}(\cdot, \ell)$ are both convex. Hence, the problem at hand is a (finite-dimensional) convex program, the structure of which depends on that of the loss space $\mathcal{L}$, over which the max in the first (and more troublesome) term of $\Lambda_{\eta,q}$ is taken. Typical examples of $\mathcal{L}$, like balls of bounded norm or the polar set of $\mathcal{X}$, can further simplify the form of the objective function, see Appendix G for details. Note that the parameter $\delta$ is introduced to maintain numerical stability. Turning back to the regret, the following bound readily follows from Lemma 2.

**Proposition 2.** *Under global observability, Algorithm 1 satisfies* $R_T \leq \dfrac{\log k}{\eta} + \eta \Lambda_\eta^* T + (2\delta + \eta\varepsilon)T$.

What remains now is deriving an upper bound on $\Lambda_\eta^*$ depending on the structure of the game, and tuning the algorithm's parameters accordingly. Note that an adequate tuning of $\eta$ would depend on $\Lambda_\eta^*$ (or a satisfactory upper bound), which might be difficult to compute. Following Lattimore and Szepesvári [40], we can use a learning rate sequence $(\eta_t)_t$, which is updated at round $t+1$ using the attained value at the optimization problem of round $t$. This still allows recovering essentially the same regret rates, details are included in Appendix H for completeness. Now, for bounding $\Lambda_\eta^*$, the properties of our $\Lambda_{\eta,q}$ allow us to invoke Sion's minimax theorem, as Lattimore and Szepesvári [40] did in the finite setting.

**Lemma 3.** *For any $\eta > 0$ and $q \in \Delta_k$, it holds that*

$$\Lambda_{\eta,q}^* = \max_{\ell \in \mathcal{L}} \min_{p \in \Xi_\eta} \Lambda_{\eta,q}(p, \ell)\,.$$

The next two sections will start from this result and derive bounds on $\Lambda_\eta^*$ for locally and globally observable games, exploiting the linear structure of the problem.

## 4 Locally Observable Games

As alluded to before, what distinguishes locally observable games is that exploration, roughly speaking, is cheap. The following lemma provides an alternative definition of local observability that appeals to this intuition.

**Lemma 4.** *A game is locally observable if and only if it holds for all $a \in \mathcal{A}^*$, $\ell \in \mathbb{R}^d$, and $a^* \in \arg\min_{a' \in \mathcal{A}} \psi_{a'}^\top \ell$ that $\psi_a - \psi_{a^*} \in \operatorname{span}\left\{M_b \mid \psi_b^\top \ell \leq \psi_a^\top \ell\right\}$.*

Fixing $q \in \Delta_k^*$ and $\eta > 0$, we show in this section how to exploit this property to bound $\Lambda_\eta^*$. We sketch here the main arguments, deferring details to Appendix E, where proofs of this section's results can be found. Note that Lemma 3 reduces the task of bounding $\Lambda_{\eta,q}^*$ to finding a uniform upper bound on $\min_{p \in \Xi_\eta} \Lambda_{\eta,q}(p, \ell)$ holding against any fixed loss vector $\ell$. With the clairvoyant-like knowledge of $\ell$ that this affords us, we start by constructing a specific distribution $p \in \Xi_\eta$ that depends on $\ell$ and derive an upper bound on $\Lambda_{\eta,q}$ when evaluated at this pair. Using this result, we can then bound $\Lambda_{\eta,q}^*$ and $\Lambda_\eta^*$ by passing to worst case over $\ell \in \mathcal{L}$ and $q \in \Delta_k^*$.

Our choice of $p$ will take the form $p = \gamma\pi + (1 - \gamma)\widehat{p}$, with $\gamma \in (0, 1/2]$ and $\pi, \widehat{p} \in \Delta_k$. For brevity, let $\bar{\gamma} = 1 - \gamma$. The role of $\pi$ is to ensure that $p \in \Xi_\eta$ and that of $\widehat{p}$ is to control the magnitude of

$\Lambda_{\eta,q}(p,\ell)$. As a starting point, it is easy to show that

$$\Lambda_{\eta,q}(\gamma\pi + \bar{\gamma}\widehat{p}, \ell)$$
$$\leq \frac{2\gamma}{\eta} + \frac{\bar{\gamma}}{\eta}\langle \widehat{p} - q, H^\top \ell \rangle + L^2 \sum_{a,b \in \mathcal{A}} q(a)q(b)\left(\psi_a - \psi_b\right)^\top Q_\delta(\gamma\pi + \bar{\gamma}\widehat{p})^\dagger \left(\psi_a - \psi_b\right) \quad (6)$$

$$\leq \frac{2\gamma}{\eta} + \frac{\bar{\gamma}}{\eta}\langle \widehat{p} - q, H^\top \ell \rangle + 2L^2 \sum_{a,b \in \mathcal{A}} q(a)q(b)\left(\psi_a - \psi_b\right)^\top Q_\delta(\widehat{p})^\dagger \left(\psi_a - \psi_b\right). \quad (7)$$

We proceed first with the selection of $\widehat{p}$ aiming to minimize the second and third terms, which exhibit a tradeoff between minimizing our loss against $\ell$ (compared to $q$) and sufficiently exploring to minimize the variance of the loss estimator. In finite partial monitoring [39, 40], the analogous predicament is resolved via a 'water transfer' operator. Leveraging properties of the same spirit as Lemma 4, this technique involves perturbing $q$ to ensure that each action receives comparable mass to actions whose estimators rely on the observations of the former, without incurring more loss in the process. While some adaptation of this approach can be utilized here (more details below), we will describe in what follows an alternative manner of redistributing the mass in $q$ that is arguably more natural (especially considering the structure of (7)), and that, moreover, only relies on the simple characterization of local observability given by Lemma 4, without explicit reference to the neighborhood relation used in finite partial monitoring.

To control the third term, it is helpful to first relate the feature differences $(\psi_a - \psi_b)_{a,b \in \mathcal{A}}$ to the observations matrices $(M_a)_{a \in \mathcal{A}}$ (or their collective form $M$) seeing that $Q_\delta(\widehat{p})$ is a linear combination of the latter. While this feat only requires global observability, the stronger assumption of local observability allows one to do so in a manner that facilitates controlling the second term simultaneously, as will be shortly shown. Towards this goal, we define some relevant objects. Let $n := \sum_{a \in \mathcal{A}} n(a)$ and $k^* := |\mathcal{A}^*|$. Note that any $v \in \mathbb{R}^n$ can, with a slight notation abuse, be decomposed as $(v(a))_{a \in \mathcal{A}}$ with $v(a) \in \mathbb{R}^{n(a)}$ such that $Mv = \sum_{a \in \mathcal{A}} M_a v(a)$. In locally observable games, Lemma 4 gives that for any loss vector $\ell \in \mathcal{L}$, action $a \in \mathcal{A}^*$, and optimal action $a^* \in \arg\min_{c \in \mathcal{A}} \psi_c^\top \ell$, there exists a vector $v \in \mathbb{R}^n$ such that $\psi_a - \psi_{a^*} = Mv$ and that, at the same time, having $v(b) \neq 0$ for some $b \in \mathcal{A}$ implies that $\psi_b^\top \ell \leq \psi_a^\top \ell$. Hence, the following set is not empty under local observability:

$$\mathcal{W}_\ell^{\text{loc}} := \Big\{ \lambda = (\lambda_a)_{a \in \mathcal{A}^*} \in \mathbb{R}^{k^* \times n} \mid \exists a^* \in \arg\min_{c \in \mathcal{A}} \psi_c^\top \ell \ \forall (a,b) \in \mathcal{A}^* \times \mathcal{A},$$
$$\psi_a - \psi_{a^*} = M\lambda_a \text{ and } \lambda_a(b) \neq 0 \Rightarrow \psi_b^\top \ell \leq \psi_a^\top \ell \Big\}.$$

Each member of this set is a sequence of weight-vectors in $\mathbb{R}^n$, one for each action in $\mathcal{A}^*$, that allows recovering the difference between its feature vector and that of a fixed optimal action by linearly combining columns from the observation matrices of better- (or similarly) performing actions. Finally, for $\lambda \in \mathcal{W}_\ell^{\text{loc}}$ and $a \in \mathcal{A}^*$, let $\text{supp}(\lambda, a) := \{b \in \mathcal{A} \mid \lambda_a(b) \neq 0\}$, and define $\text{supp}(\lambda) := \bigcup_{a \in \mathcal{A}^*} \text{supp}(\lambda, a)$; that is, the set of all actions whose observations are relied on.

Now, fixing some $\lambda \in \mathcal{W}_\ell^{\text{loc}}$ and $a^* \in \arg\min_{a' \in \mathcal{A}} \psi_{a'}^\top \ell$ (such that $\psi_a - \psi_{a^*} = M\lambda_a$ for all $a \in \mathcal{A}$), one can show that

$$\sum_{a,b \in \mathcal{A}} q(a)q(b)\left(\psi_a - \psi_b\right)^\top Q_\delta(\widehat{p})^\dagger \left(\psi_a - \psi_b\right) \leq 2\sum_{a \in \mathcal{A}} q(a)\left(\psi_a - \psi_{a^*}\right)^\top Q_\delta(\widehat{p})^\dagger \left(\psi_a - \psi_{a^*}\right)$$
$$\leq 2 \max_{s \in \mathcal{A}^*}\Big(\sum_{c \in \mathcal{A}} \|\lambda_s(c)\|\Big) \sum_{c \in \mathcal{A}} \sum_{a \in \mathcal{A}^*} q(a)\|\lambda_a(c)\| \big\|M_c^\top Q_\delta(\widehat{p})^\dagger M_c\big\|.$$

An easily demonstrated fact is that for any $r \in \Delta_k$, $\sum_{c \in \mathcal{A}} r(c)\big\|M_c^\top Q_\delta(r)^\dagger M_c\big\| \leq 2\,\text{rk}(M) \leq 2d$. In view of the bound reported above, one can exploit this property by setting $\widehat{p} = \sum_{a \in \mathcal{A}^*} q(a)\nu_a$, where $\nu_a \in \Delta_k$ is such that $\nu_a(c) \propto \|\lambda_a(c)\|$. That is, we choose $\widehat{p}$ as a mixture (weighted according to $q$) of $k^*$ distributions, each associated with a Pareto optimal action $a \in \mathcal{A}^*$ and supported on $\text{supp}(\lambda, a)$. In particular, action $a$ transfers a fraction of its total mass $q(a)$ to action $c$ proportionally to $\|\lambda_a(c)\|$, representing the 'importance' of $c$ to $a$. (For $a^*$, simply set $\nu_{a^*} = e_{a^*}$.) As an aside, though arrived at through a different manner, the exploration distribution used by Lattimore and Szepesvári [40] in the analysis of EXO in finite partial monitoring assumes a form similar to $\widehat{p}$, only

that, roughly speaking, $\nu_a$ is taken there as uniform over $\mathrm{supp}(\boldsymbol{\lambda}, a)$. We show in Appendix E.1 a way to incorporate this alternative choice (among others) into the analysis scheme showcased in this section, though resulting in a generally worse bound than what we report below for our choice of $\widehat{p}$.

Proceeding with the analysis, note that by the definition of $\mathcal{W}^{\mathrm{loc}}$, $\psi_b^\top \ell \le \psi_a^\top \ell$ for all $b \in \mathrm{supp}(\boldsymbol{\lambda}, a)$; hence, it holds that $\langle \nu_a - e_a, H^\top \ell \rangle \le 0$ for all $a \in \mathcal{A}^*$. And since $\widehat{p} - q = \sum_{a \in \mathcal{A}^*}(\nu_a - e_a)$, we see now that the second term in Equation (7) becomes non-positive. While for the third term, a refinement of the arguments laid above gives that

$$\sum_{a,b \in \mathcal{A}} q(a) q(b) \left(\psi_a - \psi_b\right)^\top Q_\delta(\widehat{p})^\dagger \left(\psi_a - \psi_b\right) \le 4\beta_{\boldsymbol{\lambda}}^2 \min\left\{\mathrm{rk}(\boldsymbol{M}), |\mathrm{supp}(\boldsymbol{\lambda})|\right\},$$

where $\beta_{\boldsymbol{\lambda}} \coloneqq \max_{a \in \mathcal{A}^*} \sum_{b \in \mathcal{A}} \|\boldsymbol{\lambda}_a(b)\|$ represents the 'difficulty' of recovering the features from the observations via $\boldsymbol{\lambda}$. The alternative bound featuring $|\mathrm{supp}(\boldsymbol{\lambda})|$ brings an improvement in problems with abundant feedback per played action, like the full-information and graph feedback problems. We then proceed by taking the infimum of the bound above over $\boldsymbol{\lambda} \in \mathcal{W}_\ell^{\mathrm{loc}}$, before passing to the worst case over $\ell \in \mathcal{L}$. What remains now is the less challenging task of bounding the first term in (7). This is done by choosing $\gamma \propto \eta$, provided $\eta$ is small enough, with the leading constant chosen to ensure that $z(p) \le 2/\eta L$; specifically, it is an upper bound on the value of an optimal-design-like criterion that $\pi$ is chosen to minimize. Ultimately, we arrive at the following result:

**Theorem 1.** *In locally observable games, it holds that*

$$\Lambda_\eta^* \le \max_{\ell \in \mathcal{L}} \inf_{\boldsymbol{\lambda} \in \mathcal{W}_\ell^{\mathrm{loc}}} 8L^2 \beta_{\boldsymbol{\lambda}}^2 \min\left\{\mathrm{rk}(\boldsymbol{M}), |\mathrm{supp}(\boldsymbol{\lambda})|\right\} + 2L\left(1 + \beta_{\mathrm{glo}}^2\right) \min\left\{\mathrm{rk}(\boldsymbol{M}), w^*\right\}$$

*provided* $\dfrac{1}{\eta} \ge 2L\left(1 + \beta_{\mathrm{glo}}^2\right) \min\left\{\mathrm{rk}(\boldsymbol{M}), w^*\right\}$, *where* $\beta_{\mathrm{glo}} \coloneqq \max_{a,b \in \mathcal{A}} \min_{\boldsymbol{v} \in \mathbb{R}^n \,:\, \psi_a - \psi_b = \boldsymbol{M}\boldsymbol{v}} \sum_{c \in \mathcal{A}} \|\boldsymbol{v}(c)\|$

*and* $w^* \coloneqq \min_{S \subseteq \mathcal{A}} |S| \max_{b \in \mathcal{A}} \left\| M_b^\top U \left(\sum_{s \in S} U^\top M_s M_s^\top U\right)^{-1} U^\top M_b \right\| \le k$.

Here, $U \in \mathbb{R}^{d \times \mathrm{rk}(\boldsymbol{M})}$ denotes a matrix whose columns $(u_i)_{i \in [\mathrm{rk}(\boldsymbol{M})]}$ form an orthonormal basis for $\mathrm{col}(\boldsymbol{M})$.[4] Let $\beta_{\mathrm{loc}} \coloneqq \max_{\ell \in \mathcal{L}} \min_{\boldsymbol{\lambda} \in \mathcal{W}_\ell^{\mathrm{loc}}} \beta_{\boldsymbol{\lambda}}$, which satisfies $\beta_{\mathrm{loc}} \ge \beta_{\mathrm{glo}}$ (see Lemma 17 in Appendix E). Assuming for simplicity that $L, \beta_{\mathrm{loc}} \ge 1$; the following simpler bound is obtained immediately:

$$\Lambda_\eta^* \le 12L^2 \beta_{\mathrm{loc}}^2 d.$$

We see that under local observability, similarly to finite partial monitoring [40], there exist constants $\alpha, B$ such that $\Lambda_\eta^* \le \alpha$ given that $\eta \le 1/B$. Assuming $\delta$ and $\varepsilon$ are sufficiently small, setting $\eta = \min\{1/B, \sqrt{\log k / \alpha T}\}$ yields via Proposition 2 that $R_T$ is $\mathcal{O}\left(\sqrt{\alpha T \log k}\right)$. We refer again to Appendix H for a discussion on adaptive learning rates following [40]. We also note here that $B$ can be chosen conservatively as it only affects the regret additively. We now use the bound of Theorem 1 to obtain regret bounds for the locally observable games discussed in Section 2. Omitted details and derivations are provided in Appendix I.

**Full information.** Here, playing any action is sufficiently informative; for any $\ell \in \mathcal{L}$, we can trivially pick $\boldsymbol{\lambda} \in \mathcal{W}_\ell^{\mathrm{loc}}$ supported only on an optimal action and satisfying $\beta_{\boldsymbol{\lambda}} \le \sqrt{2}$. Moreover, $w^* \le 1$ taking $S$ as a singleton. Hence, we recover the order-optimal $\mathcal{O}\left(\sqrt{T \log k}\right)$ regret bound [13].

**Strongly observable feedback graphs.** Generalizing the example above, we can, against any $\ell \in \mathcal{L}$, find $\boldsymbol{\lambda} \in \mathcal{W}_\ell^{\mathrm{loc}}$ supported on an independent set (a set of actions no two distinct members of which are neighbors) and satisfying $\beta_{\boldsymbol{\lambda}} \le 2$. This set can be iteratively constructed by first selecting an optimal action, removing its neighbors from the graph, and repeating this (proceeding with the best remaining action) until the graph is empty. This gives that $\beta_{\boldsymbol{\lambda}}^2 |\mathrm{supp}(\boldsymbol{\lambda})|$ is bounded up to a constant by $\alpha(\mathcal{G})$, the independence number of the graph (size of a maximal independent set). The same can be shown to hold for $w^*$, allowing us to recover the near-optimal $\mathcal{O}\left(\sqrt{\alpha(\mathcal{G}) T \log k}\right)$ regret bound for this problem [44, 2].

**Linear bandits.** Since the features and the observation vectors coincide, it is immediate that $\beta_{\mathrm{loc}} \le 2$. The regret bound is then of order $\sqrt{dT \log(k)}$ (as achieved in [10] and [35, Chp. 27]), which reduces to the near-optimal $\sqrt{kT \log(k)}$ rate for standard bandits.

---

[4]The inverse of $\sum_{s \in S} U^\top M_s M_s^\top U$ exists when $\mathrm{span}(\{M_s\}_{s \in S}) = \mathrm{col}(\boldsymbol{M})$, see Lemma 9 in Appendix C.

**Linear dueling bandits.** It also holds in this example that $\beta_{\mathrm{loc}} \leq 2$, implying again that the regret bound is of order $\sqrt{dT \log(k)}$.

**Bandits with ill-conditioned observers.** Since $e_a - e_b = {}^1\!/\varepsilon(M_a - M_b)$, we have that $\beta_{\mathrm{loc}} \leq {}^2\!/\varepsilon$. Thus, the regret bound is of order ${}^1\!/\varepsilon\sqrt{kT \log k}$. We show in Appendix J that this bound is tight up to logarithmic factors.

## 5 Globally Observable Games

On the other hand, exploration is generally costly in globally observable games. Here, starting from the decomposition in (6), we simply pick $\widehat{p} = q$ and place the burden of controlling the variance term entirely on $\pi$. Consequently, $\gamma$ must be kept sufficiently large; precisely, of order $\sqrt{\eta}$. This causes the final bound to scale with $1/\sqrt{\eta}$, hence becoming $\eta$-dependent, which aligns with the finite outcomes setting [40]. Building on this, the next theorem provides two bounds on $\Lambda_\eta^*$. The first resembles the square root of the second term in the bound of Theorem 1, while the second bound features a refined alignment constant but has a generally worse dependence on the structure of the observation matrices.

**Theorem 2.** *In globally observable games, it holds that*

$$\Lambda_\eta^* \leq 4\sqrt{1/\eta}\,L \min\Big\{ (1 + \beta_{\mathrm{glo}})\sqrt{\min\{\mathrm{rk}(\boldsymbol{M}), w^*\}},\, (1 + \beta_{2,\mathrm{glo}})\sqrt{u^*} \Big\}$$

*provided that* $\dfrac{1}{\eta} \geq (1 + L^2) \min\Big\{ (1 + \beta_{\mathrm{glo}}^2)\min\{\mathrm{rk}(\boldsymbol{M}), w^*\},\, (1 + \beta_{2,\mathrm{glo}}^2)u^* \Big\}$, *where*

$$\beta_{2,\mathrm{glo}} := \max_{a,b \in \mathcal{A}} \big\| \boldsymbol{M}^\dagger(\psi_a - \psi_b) \big\|, \text{ and } u^* := \min_{S \subseteq \mathcal{A}} |S| \big\| \boldsymbol{M}^\top U \big(\textstyle\sum_{s \in S} U^\top M_s M_s^\top U \big)^{-1} U^\top \boldsymbol{M} \big\| \leq k\,.$$

Note that $w^* \leq u^*$, and that $\beta_{2,\mathrm{glo}} = \max_{a,b \in \mathcal{A}} \min_{\boldsymbol{v} \in \mathbb{R}^n:\ \psi_a - \psi_b = \boldsymbol{M v}} \|\boldsymbol{v}\|$; hence, $\beta_{2,\mathrm{glo}} \leq \beta_{\mathrm{glo}} \leq \sqrt{k}\beta_{2,\mathrm{glo}}$. This theorem shows that in the present regime, there exist constants $\alpha, B$ such that $\Lambda_\eta^* \leq \alpha/\sqrt{\eta}$ given that $\eta \leq 1/B$. Assuming again that $\delta$ and $\varepsilon$ are sufficiently small, setting $\eta = \min\big\{ {}^1\!/B, \big({}^{\log k}\!/\alpha T\big)^{2/3} \big\}$ yields via Proposition 2 that $R_T$ is $\mathcal{O}\big((\alpha T)^{2/3}(\log k)^{1/3}\big)$. The adaptive tuning of $\eta$ is discussed in Appendix H. We now revisit the globally observable examples from Section 2.

**Weakly observable feedback graphs.** Here, it is immediate to verify that $\beta_{\mathrm{glo}} \leq 2$. A subset of vertices (actions) is said to be a total dominating set if the union of their neighborhoods covers all vertices. Moreover, the total domination number of the graph, $\delta(\mathcal{G})$, is defined as the size of a minimal total dominating set. It can be easily shown (see Appendix I) that $w^* \leq \delta(\mathcal{G})$, leading to a $\mathcal{O}\big((\delta(\mathcal{G}) \log k)^{1/3} T^{2/3}\big)$ regret bound. Further refining this bound to scale with the *weak* domination number (which only considers weakly observable nodes) used in [1] requires a careful combination of the analyses of this section and the previous one to eliminate unnecessary forced exploration over the strongly observable portion of the graph.

**Composite graph feedback.** It holds here that $w^* = u^* = \mathrm{rk}(\boldsymbol{M}) = k$; hence, the second bound is generally better. The regret bound is then $\mathcal{O}\big((k \log k)^{1/3}(\beta_{2,\mathrm{glo}} T)^{2/3}\big)$, the tightness of which remains to be studied. In Appendix I, we provide a more detailed discussion about the interpretation of $\beta_{2,\mathrm{glo}}$ in this problem, as well as two instantiations of which in simple families of graphs.

## 6 Limitations and Future Work

Besides enjoying the same rates in $T$ as the bounds derived in this work, the regret bounds derived by Kirschner et al. [31, 32] in the stochastic setting feature analogous alignment constants to $\beta_{\mathrm{loc}}$ and $\beta_{\mathrm{glo}}$, see [31, Lemma 13] or [32, Lemma 5]. Comparatively, our bounds enjoy some extra versatility as they easily yield nearly tight guarantees for feedback-rich problems like full-information and feedback graphs, which were not addressed in [31, 32]. However, our exposition is limited to full-dimensional loss spaces (like [31] but unlike [32]) and finite actions. The former aspect precludes us from recovering tight guarantees for finite partial monitoring as the observation conditions do not match in general, see [32, Example 8]. Addressing this aspect in the adversarial setting is left for future work. Likewise for studying generic compact action sets, for which a full classification theorem is lacking also in the stochastic setting. Lastly, characterizing how the minimax regret depends on instance-based constants remains a general challenging direction.

## Acknowledgments and Disclosure of Funding

KE and NCB acknowledge the financial support from the FAIR project, funded by the NextGenerationEU program within the PNRR-PE-AI scheme (M4C2, investment 1.3, line on Artificial Intelligence), the MUR PRIN grant 2022EKNE5K (Learning in Markets and Society), funded by the NextGenerationEU program within the PNRR scheme (M4C2, investment 1.1), the EU Horizon CL4-2022-HUMAN-02 research and innovation action under grant agreement 101120237, project ELIAS (European Lighthouse of AI for Sustainability), and the One Health Action Hub, University Task Force for the resilience of territorial ecosystems, funded by Università degli Studi di Milano (PSR 2021-GSA-Linea 6).

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

# A    Additional Related Works

The work of Rustichini [50] is generally recognized as the first work on partial monitoring. Notably, he studied a modified notion of regret under which non-trivial guarantees can be derived for hopeless games, see also [45, 43, 46, 34, 36]. Finite partial monitoring has also been extensively studied in the stochastic setting [7, 8, 58, 33, 53, 27, 28]. Among the works cited earlier in the adversarial (finite) setting, of most relevance here are those by Lattimore and Szepesvári [39, 40]. The first proved a minimax theorem for all games with finite actions, linking the minimax (adversarial) regret and the worst-case Bayesian regret. For finite partial monitoring, they bounded the latter by presenting an algorithm (Mario sampling) enjoying a bounded information ratio in the Bayesian setting, which in turn, implies a non-constructive guarantee on the former. The second work, as mentioned already, proposed the EXO method, and showed that its regret matches the non-constructive bounds of the first work. The subsequent work of Lattimore and Gyorgy [37] does, in some sense, link these two results in a general linear partial monitoring setup by showing that the regret of EXO can be bounded in terms of a general notion of information ratio. We note here that the results by Kirschner et al. [31, 32] in the *stationary* stochastic linear setting cannot be combined with these findings to yield bounds on the adversarial minimax regret or the regret of EXO, as that requires bounding (some form of) the information ratio for arbitrary (finitely supported) priors over *sequences* of latent outcomes.

In the same line of work, Lattimore [36] expanded on the results of Lattimore and Gyorgy [37] considering all games with infinite outcomes and finite actions. He showed that the optimal rate in $T$ is indeed determined by the behavior of the information ratio, and that for every $p \in [1/2, 1]$, there exist games where the minimax regret is $T^p$ up to subpolynomial factors. Further, Foster et al. [22] showed that the regret of EXO can also be bounded in terms of the Decision-Estimation Coefficient, a general complexity measure for interactive decision making [21], which too is rather opaque in terms of its dependence on game-specific parameters.

The first work on linear partial monitoring is by Lin et al. [42], followed by the closely related work of Chaudhuri and Tewari [15]. Both study the stochastic setting and only prove $T^{2/3}$ bounds under a global observability condition. On the other hand, they focus on designing efficient algorithms in the face of combinatorial action spaces that can be exponentially large. On a different thread, Ito and Takemura [30] employ a version of EXO in the standard linear bandit setting (hence our algorithm and theirs naturally follow a similar general template) and obtain best-of-both-worlds (BOBW) bounds (near optimal bounds in both the adversarial and the stochastic settings). Their techniques and contributions remain distinct from ours as they do not address partial monitoring. In a related line of work [54, 55, 56], EXO is used to obtain BOBW bounds in partial monitoring, but only for finite outcomes.

# B    The Geometry of a Linear Partial Monitoring Game

This section provides a brief overview of some partial monitoring concepts and terminology, see [3, 6, 31, 35], which will be used in the proofs, particularly those of the lower bounds in Appendix J. We remark that the definitions used here mostly concern the case when $\mathcal{B}_2(r) \subseteq \mathcal{L}$ for some $r > 0$, which is our focus in this work. Recall that $\mathcal{X} := \{\psi_a \mid a \in \mathcal{A}\}$, and define $\mathcal{V} := \mathrm{co}(\mathcal{X})$, which is a convex polytope. An action $a \in \mathcal{A}$ is called *Pareto optimal* if $\psi_a \in \mathrm{ext}(\mathcal{X})$; that is, $\psi_a$ is a vertex of $\mathcal{V}$. The convex cone

$$C_a := \{\ell \in \mathbb{R}^d \mid \forall b \in \mathcal{A}, \langle \psi_a - \psi_b, \ell \rangle \leq 0\} = \{\ell \in \mathbb{R}^d \mid \forall x \in \mathcal{V}, \langle \psi_a - x, \ell \rangle \leq 0\}$$

is called the *cell* of $a$. A Pareto optimal action $a$ satisfies that $\{\psi_a\} \cap \mathrm{co}(\mathcal{X} \setminus \{\psi_a\}) = \varnothing$ since $\psi_a$ is a vertex of $\mathcal{V}$. Hence, by the separating hyperplane theorem, there exists a vector $u \in \mathbb{R}^d$ such that $\psi_a^\top u < x^\top u$ for all $x \in \mathcal{X} \setminus \{\psi_a\}$, yielding that $\psi_a^\top u < \psi_b^\top u$ for all $b \in \mathcal{A}$ with $\psi_b \neq \psi_a$. It follows that $\dim(C_a) = d$, where $\dim(S)$ is the dimension of the affine hull of $S$, since we can construct a small enough Euclidean ball around $u$ where the same condition holds (recall that $\mathcal{A}$ is finite). Conversely, $\dim(C_a) = d$ implies that $a$ is Pareto optimal. This is because if $\psi_a$ can be written as convex combination of points in $\mathcal{X} \setminus \{\psi_a\}$, then for every such point $\psi_b$ (with positive weight), $\langle \psi_a - \psi_b, \ell \rangle = 0$ for all $\ell \in C_a$; hence, $C_a$ cannot be full-dimensional.

The polytope $\mathcal{V}$ induces a graph (the 1-skeleton of $\mathcal{V}$) whose vertices are the vertices of $\mathcal{V}$, two of which are adjacent in this graph if they are the endpoints of an edge of $\mathcal{V}$. Two non-duplicate (i.e., having different features) Pareto optimal actions $a$ and $b$ are said to be *neighbors* if $\psi_a$ and $\psi_b$ are

adjacent in the 1-skeleton of $\mathcal{V}$. It is known that this graph is connected [5]. For any two neighbors $a$ and $b$, it holds that $\dim(C_a \cap C_b) = d - 1$. This is true since
$$C_a \cap C_b = \left\{ \ell \in \mathbb{R}^d \mid \langle \psi_a - \psi_b, \ell \rangle = 0 \text{ and } \forall c \in \mathcal{A}, \langle \psi_a - \psi_c, \ell \rangle \le 0 \right\},$$
$\mathcal{A}$ is finite, and the fact that there exists (by virtue of $\psi_a$ and $\psi_b$ being endpoints of an edge of $\mathcal{V}$) a vector $u \in \mathbb{R}^d$ such that $\psi_a^\top u < x^\top u$ for all $x \in \mathcal{V} \setminus [\psi_a, \psi_b]$ (which also excludes duplicates of $a$ and $b$) and $\psi_a^\top u = \psi_b^\top u$. Conversely, if $\dim(C_a \cap C_b) = d - 1$ for two Pareto optimal actions $a$ and $b$, then they must be neighbors. To see this, note that each $\ell \in C_a \cap C_b$ can be mapped to a face of $\mathcal{V}$ containing both $\psi_a$ and $\psi_b$ in addition to points $x \in \mathcal{V}$ satisfying $\langle x, \ell \rangle = \langle \psi_a, \ell \rangle$. Since the intersection of two faces is also a face, the intersection of all faces containing $\psi_a$ and $\psi_b$ is still a face of $\mathcal{V}$. If $a$ and $b$ are not neighbors, this face must contain some vertex $\psi_c$ different from $\psi_a$ and $\psi_b$. Therefore, for any $\ell \in C_a \cap C_b$, $\langle \psi_c, \ell \rangle = \langle \psi_a, \ell \rangle = \langle \psi_b, \ell \rangle$, implying that $\dim(C_a \cap C_b) \le d - 2$ since $\psi_a, \psi_b$, and $\psi_c$ are affinely independent being all distinct vertices of $\mathcal{V}$. Lastly, for neighboring actions $a$ and $b$, define their neighborhood as
$$\mathcal{N}_{a,b} := \{ c \in \mathcal{A} \mid \psi_c \in [\psi_a, \psi_b] \} = \left\{ c \in \mathcal{A} \mid \psi_c = \alpha \psi_a + (1 - \alpha) \psi_b, \alpha \in [0, 1] \right\},$$
which includes the duplicates of $a$ and $b$. (Note that the neighborhood relation is defined only for non-duplicate Pareto optimal actions.)

Recall the definition $\mathcal{P}(\ell) := \{ a \in \mathcal{A} \mid \langle \psi_a - \psi_b, \ell \rangle \le 0 \text{ for all } b \in \mathcal{A} \}$ for $\ell \in \mathbb{R}^d$, extending to sets $D \subseteq \mathbb{R}^d$ via $\mathcal{P}(D) := \cup_{\ell \in D} \mathcal{P}(\ell)$. Lemma 26 in [31] shows that local observability as defined via Definition 1 is equivalent to the more usual local observability condition used in finite partial monitoring. The following lemma expands on that adding two more equivalent conditions, one of which was featured in Lemma 4.

**Lemma 5.** *The following conditions are equivalent.*

    *i) For every convex set $D \subseteq \mathbb{R}^d$,*
$$\psi_a - \psi_b \in \operatorname{span}\{ M_c \mid c \in \mathcal{P}(D) \} \quad \text{for all } a, b \in \mathcal{P}(D).$$

    *ii) For any pair of neighboring actions $a$ and $b$,*
$$\psi_a - \psi_b \in \operatorname{span}\{ M_c \mid c \in \mathcal{N}_{a,b} \}.$$

    *iii) For any Pareto optimal action $a$, $\ell \in \mathbb{R}^d$, and $a^* \in \arg\min_{a' \in \mathcal{A}} \psi_{a'}^\top \ell$,*
$$\psi_a - \psi_{a^*} \in \operatorname{span}\left\{ M_b \mid \psi_b^\top \ell \le \psi_a^\top \ell \right\}.$$

    *iv) For any pair of Pareto optimal actions $a$ and $b$, and any $\ell \in \mathbb{R}^d$,*
$$\psi_a - \psi_b \in \operatorname{span}\left\{ M_c \mid \psi_c^\top \ell \le \max\{ \psi_a^\top \ell, \psi_b^\top \ell \} \right\}.$$

*Proof. i) $\Leftrightarrow$ ii)* This is given by [31, Lemma 26]. (Note that, unlike [31], our setting allows duplicate actions; however, their proof extends immediately to the case when duplicates are present.)

*i) $\Rightarrow$ iii)* Since $a$ is Pareto optimal, $\psi_a \in \operatorname{ext}(\mathcal{V})$, implying that $\{\psi_a\} \cap \operatorname{co}(\mathcal{X} \setminus \{\psi_a\}) = \varnothing$. Hence, by the separating hyperplane theorem, there exists a vector $v \in \mathbb{R}^d$ such that $\psi_a^\top v < x^\top v$ for all $x \in \mathcal{X} \setminus \{\psi_a\}$, yielding that $\psi_a^\top v < \psi_b^\top v$ for all $b \in \mathcal{A}$ with $\psi_b \ne \psi_a$. Define $E$ as the line segment between $v$ and $\ell$, i.e.,
$$E := \left\{ \ell' \in \mathbb{R}^d \mid \ell' = (1 - \lambda)\ell + \lambda v, \lambda \in [0, 1] \right\}.$$
Now, considering that $a, a^* \in \mathcal{P}(E)$, Condition *i)* implies that
$$\psi_a - \psi_{a^*} \in \operatorname{span}\{ M_b \mid b \in \mathcal{P}(E) \}.$$
We now show that for every $b \in \mathcal{P}(E)$, $\psi_b^\top \ell \le \psi_a^\top \ell$, from which the lemma follows directly. If $\psi_b = \psi_a$, the inequality holds trivially. Suppose now that $\psi_b \ne \psi_a$. By the definition of $v$, $\psi_a^\top v < \psi_b^\top v$. Hence, since $b \in \mathcal{P}(E)$, we have that for some $\lambda \in [0, 1)$,
$$(1 - \lambda)\psi_b^\top \ell - (1 - \lambda)\psi_a^\top \ell \le \lambda \psi_a^\top v - \lambda \psi_b^\top v \le 0.$$
Dividing by $1 - \lambda$ (note that $\lambda < 1$) gives that $\psi_b^\top \ell \le \psi_a^\top \ell$ as required.

*iii) $\Rightarrow$ iv)* This can be seen immediately upon writing $\psi_a - \psi_b = \psi_a - \psi_{a^*} - (\psi_b - \psi_{a^*})$.

*iv) $\Rightarrow$ ii)* For neighbors $a$ and $b$, $[\psi_a, \psi_b]$ is a face (in particular, an edge) of the polytope $\mathcal{V}$; hence, there exists $r \in \mathbb{R}$ and $\ell \in \mathbb{R}^d$ such that $x^\top \ell = r$ for $x \in [\psi_a, \psi_b]$ and $y^\top \ell > r$ for $y \in \mathcal{V} \setminus [\psi_a, \psi_b]$. This gives, via the definition of $\mathcal{N}_{a,b}$, that for any $c \in \mathcal{A}$, $\psi_c^\top \ell \le \max\{ \psi_a^\top \ell, \psi_b^\top \ell \} \Rightarrow c \in \mathcal{N}_{a,b}$, which established the sought implication. $\qquad\square$

## C Auxiliary Results

We will use $\mathcal{S}^n_{\succeq}$ in the following to denote the set of (symmetric) positive semi-definite matrices in $\mathbb{R}^{n \times n}$, while $\succeq$ will refer to the Loewner order: for any $F, G \in \mathcal{S}^n_{\succeq}$, $F \succeq G$ is equivalent to $F - G \in \mathcal{S}^n_{\succeq}$.

**Lemma 6.** *Let $n$ and $m$ be two positive integers, and let $\mathcal{V}$ be a subspace of $\mathbb{R}^n$. Then, the mapping $(X, G) \mapsto X^\top G^\dagger X$ is convex on $\mathbb{R}^{n \times m} \times \{G \in \mathcal{S}^n_{\succeq} \mid \mathrm{col}(G) = \mathcal{V}\}$ with respect to the Loewner order.*

*Proof.* This is a direct consequence of Theorem 8 in [47]. $\qquad\square$

**Lemma 7.** *Let $n$ and $m$ be two positive integers. For $F, G \in \mathcal{S}^n_{\succeq}$ and $X \in \mathbb{R}^{n \times m}$ such that $\mathrm{col}(X) \subseteq \mathrm{col}(G)$, it holds that*

$$X^\top (F + G)^\dagger X \preceq X^\top G^\dagger X\,.$$

*Proof.* This is a direct consequence of Theorem 5 in [12]. $\qquad\square$

Recall that for $\pi \in \Delta_k$ and $\delta \in (0, 1)$,

$$Q(\pi) \coloneqq \sum_{a \in \mathcal{A}} \pi(a)\, M_a M_a^\top \qquad \text{and} \qquad Q_\delta(\pi) \coloneqq Q((1 - \delta)\pi + \delta \mathbf{1}_k/k)\,.$$

Recall also that $U \in \mathbb{R}^{d \times \mathrm{rk}(\boldsymbol{M})}$ is a matrix whose columns $(u_i)_{i \in [\mathrm{rk}(\boldsymbol{M})]}$ form an orthonormal basis for $\mathrm{col}(\boldsymbol{M})$, with $\boldsymbol{M}$ being the $d \times (\sum_a n(a))$ matrix obtained by horizontally stacking all the observations matrices $(M_a)_{a \in \mathcal{A}}$. Now, for $\pi \in \Delta_k$ and $\delta \in (0, 1)$, define

$$B(\pi) \coloneqq \sum_{a \in \mathcal{A}} \pi(a)\, U^\top M_a M_a^\top U = U^\top Q(\pi) U \quad \text{and} \quad B_\delta(\pi) \coloneqq B((1 - \delta)\pi + \delta \mathbf{1}_k/k)\,. \quad (8)$$

Finally, recall that given a finite index set $Z$ and a collection of matrices $\{X_z \mid z \in Z\}$, all with the same number of rows, we denote by $\mathrm{span}\{X_z \mid z \in Z\}$ the span of all their columns.

**Lemma 8.** *For any $\pi \in \Delta_k$, $\mathrm{col}(Q(\pi)) = \mathrm{span}\{M_a \mid \pi(a) > 0\}$.*

*Proof.* Let $n \coloneqq \sum_{a \in \mathcal{A}} n_a$, $v_\pi$ be the $n$-dimensional vector constructed by concatenating the vectors $\sqrt{\pi(a)}\mathbf{1}_{n(a)}$ in order, and $D_\pi$ be the $n \times n$ diagonal matrix with $v_\pi$ on the diagonal. Note then that $Q(\pi)$ can be written as $\boldsymbol{M} D_\pi^2 \boldsymbol{M}^T$. Hence,

$$\mathrm{col}(Q(\pi)) = \mathrm{col}(\boldsymbol{M} D_\pi^2 \boldsymbol{M}^T) = \mathrm{col}(\boldsymbol{M} D_\pi) = \mathrm{span}\{M_a \mid \pi(a) > 0\}\,.$$

$\qquad\square$

**Lemma 9.** *For any $\pi \in \Delta_k$, the following conditions are equivalent.*

    *i)* $\mathrm{span}\{M_a \mid \pi(a) > 0\} = \mathrm{col}(\boldsymbol{M})$;

    *ii)* $\mathrm{col}(Q(\pi)) = \mathrm{col}(\boldsymbol{M})$;

    *iii)* $B(\pi)$ *is positive definite*;

    *iv)* $\mathrm{span}\{U^\top M_a \mid \pi(a) > 0\} = \mathbb{R}^{\mathrm{rk}(\boldsymbol{M})}$.

*Proof.*
*i)* $\Leftrightarrow$ *ii)* Follows directly from Lemma 8.

*i)* $\Rightarrow$ *iii)* For any nonzero $v \in \mathbb{R}^{\mathrm{rk}(\boldsymbol{M})}$,

$$v^\top B(\pi)v = v^\top U^\top Q(\pi) U v = (Uv)^\top Q(\pi)(Uv) = \sum_{a \in \mathcal{A}} \pi(a) \left\| M_a^\top U v \right\|^2\,.$$

Note that $Uv \neq 0$ since $U$ has full rank, and $Uv \in \mathrm{col}(\boldsymbol{M})$ by the definition of $U$. Hence, thanks to Condition $i$), $M_a^\top Uv \neq 0$ for some $a \in \mathcal{A}$ with $\pi(a) > 0$, yielding that $\sum_{a \in \mathcal{A}} \pi(a) \left\| M_a^\top Uv \right\|^2 > 0$.

$iii) \Rightarrow iv)$ Applying an analogous argument to the one used in the proof of Lemma 8, one can show that
$$\mathrm{col}(B(\pi)) = \mathrm{span}\{U^\top M_a \mid \pi(a) > 0\}\,.$$

The required property then follows since $\mathrm{col}(B(\pi)) = \mathbb{R}^{\mathrm{rk}(\boldsymbol{M})}$ by the assumption that $B(\pi)$ is positive definite.

$iv) \Rightarrow i)$ For any $x \in \mathbb{R}^d$, Condition $iv$) implies that $U^\top x \in \mathbb{R}^{\mathrm{rk}(\boldsymbol{M})}$ can be written as $U^\top x = \sum_{a \in \mathcal{A}} U^\top M_a v_a$ for some collection of vectors $(v_a)_{a \in \mathcal{A}}$ such that $v_a \in \mathbb{R}^{n(a)}$ and $v_a = \boldsymbol{0}$ whenever $\pi(a) = 0$. Now, take $x$ to be in $\mathrm{col}(\boldsymbol{M})$. Since $\mathrm{col}(M_a) \in \mathrm{col}(\boldsymbol{M})$ and $UU^\top$ is the projection matrix onto $\mathrm{col}(\boldsymbol{M})$, we obtain that
$$x = UU^\top x = \sum_{a \in \mathcal{A}} UU^\top M_a v_a = \sum_{a \in \mathcal{A}} M_a v_a\,.$$

This shows that $\mathrm{col}(\boldsymbol{M}) \subseteq \mathrm{span}\{M_a \mid \pi(a) > 0\}$. The required property then readily follows as the converse statement is trivial.

$\square$

**Lemma 10.** *Fix some $\pi \in \Delta_k$ such that $B(\pi)$ is positive definite. Then, $U\,B(\pi)^{-1}\,U^\top = Q(\pi)^\dagger$.*

*Proof.* For brevity, let $Q \coloneqq Q(\pi)$, $B \coloneqq B(\pi)$, and $X \coloneqq UB^{-1}U^\top$. Also, recall that $B = U^\top QU$. Since $U$ has orthonormal columns, $U^\top U = \boldsymbol{I}_{\mathrm{rk}(\boldsymbol{M})}$ and $UU^\top$ is the orthogonal projection matrix onto $\mathrm{col}(U)$, which coincides with $\mathrm{col}(\boldsymbol{M})$. Additionally, note that the row and column spaces of $Q$ are the same thanks to symmetry, both of which also coincide with $\mathrm{col}(\boldsymbol{M})$ via Lemma 9. Hence,
$$UBU^\top = UU^\top QUU^\top = Q\,.$$

We now prove the sought identity, that $X = Q^\dagger$, by verifying the MoorePenrose conditions:
$$\begin{aligned}
QXQ &= UBU^\top UB^{-1}U^\top UBU^\top = UBU^\top = Q\,,\\
XQX &= UB^{-1}U^\top UBU^\top UB^{-1}U^\top = UB^{-1}U^\top = X\,,\\
(QX)^\top &= (UU^\top)^\top = UU^\top = QX\,,\\
(XQ)^\top &= (UU^\top)^\top = UU^\top = XQ\,.
\end{aligned}$$

$\square$

**Lemma 11.** *For any $\pi \in \Delta_k$ and $\delta \in (0,1)$, it holds that $\mathrm{col}(Q_\delta(\pi)) = \mathrm{col}(\boldsymbol{M})$, $B_\delta(\pi)$ is positive definite, and $U\,B_\delta(\pi)^{-1}\,U^\top = Q_\delta(\pi)^\dagger$.*

*Proof.* As $(1-\delta)\pi + \delta \boldsymbol{1}_k / k$ has full support, the requirements follow from Lemma 9, Lemma 10, and the definitions of $Q_\delta(\pi)$ and $B_\delta(\pi)$. $\square$

**Lemma 12** (Extension of Theorem 21.1 in [35]). *Define the functions $f, g \colon \Delta_k \to [-\infty, +\infty]$ as*
$$f(\pi) \coloneqq \log \det B(\pi) \qquad and \qquad g(\pi) \coloneqq \max_{a \in \mathcal{A}} \mathrm{tr}\big(M_a^\top UB(\pi)^{-1}U^\top M_a\big)\,.$$

*Then, for any $\pi^* \in \Delta_k$, the following are equivalent:*

    *i) $\pi^*$ is a minimizer of $g$;*

    *ii) $\pi^*$ is a maximizer of $f$;*

    *iii) $g(\pi^*) = \mathrm{rk}(\boldsymbol{M})$.*

*Proof.* For brevity, let $r := \mathrm{rk}(\boldsymbol{M})$ and $g_a(\pi) := \mathrm{tr}\big(M_a^\top U B(\pi)^{-1} U^\top M_a\big)$ for $a \in \mathcal{A}$. When $B(\pi)$ is ill-conditioned, $f$ and $g$ map to $-\infty$ and $+\infty$ respectively. It can be verified via standard arguments that $f$ and $g$ indeed attain their maximum and minimum respectively on $\Delta_k$.

Via the definition of $B$ (see (8)) and the fact that $d(\log \det X) = \mathrm{tr}(X^{-1}dX)$, we have that

$$\big(\nabla f(\pi)\big)_a \; = \; \mathrm{tr}\big(B(\pi)^{-1} U^\top M_a M_a^\top U\big) \; = \; \mathrm{tr}\big(M_a^\top U B(\pi)^{-1} U^\top M_a\big) = g_a(\pi)\,.$$

Using the linearity of the trace, we get that

$$\sum_{a \in \mathcal{A}} \pi(a) \big(\nabla f(\pi)\big)_a = \sum_{a \in \mathcal{A}} \pi(a) g_a(\pi) = \mathrm{tr}\Big(B(\pi)^{-1} \sum_{a \in \mathcal{A}} \pi(a) U^\top M_a M_a^\top U\Big) = \mathrm{tr}(\boldsymbol{I}_r) = r\,.$$

*ii) $\Rightarrow$ i)* By the concavity of $f$, the first-order optimality condition gives that for any $\pi \in \Delta_k$,

$$0 \geq \langle \nabla f(\pi^*), \pi - \pi^* \rangle = \sum_a \pi(a) g_a(\pi^*) - r\,.$$

Taking $\pi$ as a Dirac over some $a \in \mathcal{A}$ gives that $g_a(\pi^*) \leq r$; hence, $g(\pi^*) \leq r$. However, for any $\pi \in \Delta_k$, $g(\pi) \geq \sum_{a \in \mathcal{A}} \pi(a) g_a(\pi) = r$, so $\pi^*$ minimizes $g$ and $\min_{\pi \in \Delta_k} g(\pi) = r$.

*iii) $\Rightarrow$ ii)* If $g(\pi^*) = r$, then $g_a(\pi^*) \leq r$ for every $a \in \mathcal{A}$; hence,

$$\langle \nabla f(\pi^*), \pi - \pi^* \rangle = \sum_{a \in \mathcal{A}} \pi(a) g_a(\pi^*) - r \leq 0$$

for all $\pi \in \Delta_k$. The concavity of $f$ therefore implies that $\pi^*$ maximizes $f$.

*i) $\Rightarrow$ iii)* From the first part we know that $\min_{\pi \in \Delta_k} g(\pi) = r$; hence, $\pi^*$, being a minimizer of $g$, satisfies $g(\pi^*) = r$. $\qquad\square$

**Lemma 13.** *For any $p, q \in \Delta_k$, it holds that*

$$\sum_{a,b \in \mathcal{A}} q(a) q(b) (\psi_a - \psi_b)^\top Q_\delta(p)^\dagger (\psi_a - \psi_b) = 2 \sum_{a \in \mathcal{A}} q(a) (\psi_a - Hq)^\top Q_\delta(p)^\dagger (\psi_a - Hq)\,.$$

*Proof.* Note that via the definition of $H$, we have that $Hq = \sum_{b \in \mathcal{A}} q(b) \psi_b$. For brevity, let $X := Q_\delta(p)^\dagger$. It is also noteworthy that the only relevant property of $Q_\delta(p)^\dagger$ here is its symmetry. Starting with the left-hand-side, we have that

$$\sum_{a,b \in \mathcal{A}} q(a) q(b) (\psi_a - \psi_b)^\top X (\psi_a - \psi_b) = 2 \sum_{a \in \mathcal{A}} q(a) \psi_a^\top X \psi_a - 2 \sum_{a,b \in \mathcal{A}} q(a) q(b) \psi_a^\top X \psi_b$$

$$= 2 \sum_{a \in \mathcal{A}} q(a) \psi_a^\top X \psi_a - 2(Hq)^\top X Hq\,.$$

where the equivalence of the two cross terms in the initial expansion follows from the symmetry of $X$. Next, expanding the right-hand-side of the sought equality yields that

$$2 \sum_{a \in \mathcal{A}} q(a) (\psi_a - Hq)^\top X (\psi_a - Hq) = 2 \sum_{a \in \mathcal{A}} q(a) \psi_a^\top X \psi_a - 4 \sum_{a \in \mathcal{A}} q(a) \psi_a^\top X Hq + 2(Hq)^\top X Hq$$

$$= 2 \sum_{a \in \mathcal{A}} q(a) \psi_a^\top X \psi_a - 2(Hq)^\top X Hq\,,$$

which concludes the proof. $\qquad\square$

**Lemma 14.** *For any $p, q \in \Delta_k$, it holds that*

$$\sum_{a \in \mathcal{A}} q(a) (\psi_a - Hq)^\top Q_\delta(p)^\dagger (\psi_a - Hq) = \min_{r \in \Delta_k} \sum_{a \in \mathcal{A}} q(a) (\psi_a - Hr)^\top Q_\delta(p)^\dagger (\psi_a - Hr)\,.$$

*Proof.* Define $F \colon \Delta_k \to \mathbb{R}$ such that $F(r) = \sum_{a \in \mathcal{A}} q(a) (\psi_a - Hr)^\top Q_\delta(p)^\dagger (\psi_a - Hr)$, which is easily seen to be convex ($Q_\delta(p)^\dagger$ is positive semi-definite). Its gradient is given by

$$\nabla F(r) = 2 H^\top Q_\delta(p)^\dagger Hr - 2 H^\top Q_\delta(p)^\dagger Hq\,.$$

This clearly vanishes when $r = q$; hence, via the convexity of $F$, it is minimized at $q$. $\qquad\square$

# D Proofs of Section 3

As a starting point for the analysis, the following lemma states a standard result for exponential weights. A proof can be easily extracted, for example, from the proof of Theorem 1.5 in [26].

**Lemma 15.** *Assuming that at every round $t$, $\eta\widehat{y}_t(a) \geq -1$ holds for every action $a$; then, for any $a^* \in \mathcal{A}^*$, the sequence of predictions $(q_t)_t$ defined by (4) satisfies*

$$\sum_{t=1}^{T} \langle q_t - e_{a^*}, \widehat{y}_t \rangle \leq \frac{\log k}{\eta} + \eta \sum_{t=1}^{T} \langle q_t, \widehat{y}_t^2 \rangle \, .$$

**Lemma 1.** *Fix a learning policy and define $p_t$ as the law of $A_t$ conditioned on $\mathcal{F}_{t-1}$. Let $q_t$ be as given in (4) for some learning rate $\eta > 0$ and sequence of surrogate loss functions $(\widehat{y}_t)_t$ such that $\widehat{y}_t \in \mathbb{R}^k$ is $\mathcal{F}_t-$measurable and satisfies $\max_{a \in \mathcal{A}} |\eta\widehat{y}_t(a)| \leq 1$. Further, let $a^* \in \arg\min_{a \in \mathcal{A}^*} \sum_{t=1}^{T} y_t(a)$. Then, the regret of the policy satisfies*

$$R_T \leq \frac{\log k}{\eta} + \sum_{t=1}^{T} \mathbb{E}\left[ \langle p_t - q_t, y_t \rangle + \eta\langle q_t, \mathbb{E}_t \widehat{y}_t^2 \rangle + \langle q_t - e_{a^*}, y_t - \mathbb{E}_t \widehat{y}_t \rangle \right] \, .$$

*Proof.* Firstly, we have that

$$R_T = \mathbb{E}\left[ \sum_{t=1}^{T} y_t(A_t) - y_t(a^*) \right]$$

$$= \mathbb{E}\left[ \sum_{t=1}^{T} \langle p_t - e_{a^*}, y_t \rangle \right]$$

$$= \mathbb{E}\left[ \sum_{t=1}^{T} \langle p_t - q_t, y_t \rangle + \sum_{t=1}^{T} \langle q_t - e_{a^*}, \widehat{y}_t \rangle + \sum_{t=1}^{T} \langle q_t - e_{a^*}, y_t - \widehat{y}_t \rangle \right] \, ,$$

where the second equality follows from the definition of $p_t$, the linearity of expectation, and the tower rule. Next, we apply Lemma 15 to the middle term ($\eta\widehat{y}_t(a) \geq -1$ holds by assumption) to get that

$$R_T \leq \frac{\log k}{\eta} + \mathbb{E}\left[ \sum_{t=1}^{T} \langle p_t - q_t, y_t \rangle + \eta \sum_{t=1}^{T} \langle q_t, \widehat{y}_t^2 \rangle + \sum_{t=1}^{T} \langle q_t - e_{a^*}, y_t - \widehat{y}_t \rangle \right]$$

$$= \frac{\log k}{\eta} + \sum_{t=1}^{T} \mathbb{E}\left[ \langle p_t - q_t, y_t \rangle + \eta\langle q_t, \mathbb{E}_t \widehat{y}_t^2 \rangle + \langle q_t - e_{a^*}, y_t - \mathbb{E}_t \widehat{y}_t \rangle \right] \, ,$$

where the equality is another application of the linearity of expectation and the tower rule, using the fact that $q_t$ is $\mathcal{F}_{t-1}$ measurable as it only depends on $(\widehat{y}_s)_{s=1}^{t-1}$. $\qquad\square$

**Lemma 2.** *In the same setting as Lemma 1, let the predictions of the policy satisfy $p_t = (1 - \delta)\widetilde{p}_t + \delta\mathbf{1}_k/k$ for some $\delta \in (0, 1)$ and $\widetilde{p}_t \in \Delta_k$, and let the surrogate loss functions satisfy $\widehat{y}_t = g_t(A_t, \phi_t)$ where $g_t(a, \phi) = \left( \mathbf{I}_k - \mathbf{1}_k q_t^\top \right) H^\top Q(p_t)^\dagger M_a \phi$. Then, assuming global observability, it holds that*

$$R_T \leq \frac{\log k}{\eta} + 2\delta T + \sum_{t=1}^{T} \mathbb{E}\left[ \langle \widetilde{p}_t - q_t, H^\top \ell_t \rangle + \eta\omega^2 \sum_{a,b \in \mathcal{A}} q_t(a)q_t(b)(\psi_a - \psi_b)^\top Q_\delta(\widetilde{p}_t)^\dagger (\psi_a - \psi_b) \right]$$

*provided that $\max_{a,b \in \mathcal{A}}(\psi_a - \psi_b)^\top Q_\delta(\widetilde{p}_t)^\dagger (\psi_a - \psi_b) + \max_{c \in \mathcal{A}} \|M_c^\top Q_\delta(\widetilde{p}_t)^\dagger M_c\|_2 \leq \frac{2}{\eta\omega}$, where*

$$\omega := \sup_{a,b,c \in \mathcal{A}, \, p \in \Delta_k, \, \ell \in \mathcal{L}} \frac{|(\psi_b - \psi_c)^\top Q_\delta(p)^\dagger M_a M_a^\top \ell|}{\|M_a^\top Q_\delta(p)^\dagger (\psi_b - \psi_c)\|} \leq \max_{a \in \mathcal{A}, \ell \in \mathcal{L}} \|M_a^\top \ell\| \, .$$

*Proof.* Fix some $a^* \in \mathcal{A}^*$. If $\|g_t(A_t, \phi_t)\|_\infty \leq \eta^{-1}$ holds in all rounds, Lemma 1 gives that $R_T \leq \eta^{-1}\log k + \sum_{t=1}^{T} \mathbb{E}\Gamma_t$, where

$$\Gamma_t := \langle p_t - q_t, H^\top \ell_t \rangle + \eta\left\langle q_t, \sum_a p_t(a)g_t(a, M_a^\top \ell_t)^2 \right\rangle + \left\langle q_t - e_{a^*}, H^\top \ell_t - \sum_a p_t(a)g_t(a, M_a^\top \ell_t) \right\rangle . \tag{9}$$

Concerning the third term in (9), we have that

$$\sum_a p_t(a) g_t(a, M_a^\top \ell_t) = \left(\boldsymbol{I}_k - \boldsymbol{1}_k q_t^\top\right) H^\top Q(p_t)^\dagger \sum_a p_t(a) M_a M_a^\top \ell_t$$
$$= \left(\boldsymbol{I}_k - \boldsymbol{1}_k q_t^\top\right) H^\top Q(p_t)^\dagger Q(p_t)\ell_t$$
$$= \left(\boldsymbol{I}_k - \boldsymbol{1}_k q_t^\top\right) H^\top U B_\delta(\widetilde{p}_t)^{-1} U^\top U B_\delta(\widetilde{p}_t) U^\top \ell_t$$
$$= \left(\boldsymbol{I}_k - \boldsymbol{1}_k q_t^\top\right) H^\top U U^\top \ell_t$$
$$= \left(\boldsymbol{I}_k - \boldsymbol{1}_k q_t^\top\right) H^\top \ell_t\,,$$

where $U$ and $B_\delta(\cdot)$ are defined in Appendix C (see (8)), and the last equality holds since each row in $\left(\boldsymbol{I}_k - \boldsymbol{1}_k q_t^\top\right) H^\top$ is a convex combination of vectors of the form $\psi_a - \psi_b$ with $a, b \in \mathcal{A}$, which, via global observability, belong to $\mathrm{col}(\boldsymbol{M})$, onto which $UU^\top$ projects. Hence, we get that

$$\left\langle q_t - e_{a^*}, H^\top \ell_t - \sum_a p_t(a) g_t(a, M_a^\top \ell_t)\right\rangle = \left\langle q_t - e_{a^*}, H^\top \ell_t - \left(\boldsymbol{I}_k - \boldsymbol{1}_k q_t^\top\right) H^\top \ell_t\right\rangle$$
$$= \left\langle q_t - e_{a^*}, \boldsymbol{1}_k q_t^\top H^\top \ell_t\right\rangle$$
$$= 0$$

using in the last equality that $q_t, e_{a^*} \in \Delta_k$ and all coordinates of $\boldsymbol{1}_k q_t^\top H^\top \ell_t$ are identical. In what follows, we will refer to the individual coordinates of $g_t(\cdot, \cdot)$ as $g_t(b; \cdot, \cdot)$ for $b \in \mathcal{A}$. Shifting to the second term, fix $b \in \mathcal{A}$ and observe that

$$\sum_a p_t(a) g_t(b; a, M_a^\top \ell_t)^2 = \sum_a p_t(a) \left((\psi_b - H q_t)^\top Q_\delta(\widetilde{p}_t)^\dagger M_a M_a^\top \ell_t\right)^2$$
$$= \sum_a p_t(a) \left(q_t^\top (\psi_b \boldsymbol{1}_k^\top - H)^\top Q_\delta(\widetilde{p}_t)^\dagger M_a M_a^\top \ell_t\right)^2$$
$$\leq \sum_a p_t(a) \sum_c q_t(c) \left((\psi_b - \psi_c)^\top Q_\delta(\widetilde{p}_t)^\dagger M_a M_a^\top \ell_t\right)^2$$
$$\leq \omega^2 \sum_a p_t(a) \sum_c q_t(c) \left\|M_a^\top Q_\delta(\widetilde{p}_t)^\dagger (\psi_b - \psi_c)\right\|^2$$
$$= \omega^2 \sum_a p_t(a) \sum_c q_t(c) (\psi_b - \psi_c)^\top Q_\delta(\widetilde{p}_t)^\dagger M_a M_a^\top Q_\delta(\widetilde{p}_t)^\dagger (\psi_b - \psi_c)$$
$$= \omega^2 \sum_c q_t(c) (\psi_b - \psi_c)^\top Q_\delta(\widetilde{p}_t)^\dagger Q_\delta(\widetilde{p}_t) Q_\delta(\widetilde{p}_t)^\dagger (\psi_b - \psi_c)$$
$$= \omega^2 \sum_c q_t(c) (\psi_b - \psi_c)^\top Q_\delta(\widetilde{p}_t)^\dagger (\psi_b - \psi_c)\,,$$

where the first inequality is an application of Jensen's inequality, while the second inequality follows from the definition of $\omega$.

As for the first term in (9), it is immediate that

$$\langle p_t - q_t, H^\top \ell_t\rangle = \delta\langle \boldsymbol{1}_k/k - q_t, H^\top \ell_t\rangle + (1 - \delta)\langle \widetilde{p}_t - q_t, H^\top \ell_t\rangle \leq 2\delta + \langle \widetilde{p}_t - q_t, H^\top \ell_t\rangle\,,$$

where we used the definition of $p_t$ and the assumptions that $\max_{a,b \in \mathcal{A}, \ell \in \mathcal{L}} |(\psi_a - \psi_b)^\top \ell| \leq 2$ and $\delta \in (0, 1)$.

Finally, we examine the boundedness condition. Fixing $a, b \in \mathcal{A}$, we have that

$$|g_t(b; a, M_a^\top \ell_t)| = |(\psi_b - H q_t)^\top Q_\delta(\widetilde{p}_t)^\dagger M_a M_a^\top \ell_t|$$
$$= |q_t^\top (\psi_b \boldsymbol{1}_k^\top - H)^\top Q_\delta(\widetilde{p}_t)^\dagger M_a M_a^\top \ell_t|$$
$$\leq \sum_c q_t(c) |(\psi_b - \psi_c)^\top Q_\delta(\widetilde{p}_t)^\dagger M_a M_a^\top \ell_t|$$
$$\leq \omega \sum_c q_t(c) \left\|M_a^\top \left(Q_\delta(\widetilde{p}_t)^\dagger\right)^{1/2} \left(Q_\delta(\widetilde{p}_t)^\dagger\right)^{1/2} (\psi_b - \psi_c)\right\|$$
$$\leq \omega \sum_c q_t(c) \left\|M_a^\top \left(Q_\delta(\widetilde{p}_t)^\dagger\right)^{1/2}\right\| \left\|\left(Q_\delta(\widetilde{p}_t)^\dagger\right)^{1/2} (\psi_b - \psi_c)\right\|$$
$$\leq \frac{\omega}{2} \sum_c q_t(c) \left(\left\|M_a^\top \left(Q_\delta(\widetilde{p}_t)^\dagger\right)^{1/2}\right\|^2 + \left\|\left(Q_\delta(\widetilde{p}_t)^\dagger\right)^{1/2} (\psi_b - \psi_c)\right\|^2\right)$$
$$\leq \frac{\omega}{2} \left(\left\|M_a^\top Q_\delta(\widetilde{p}_t)^\dagger M_a\right\| + \max_{c \in \mathcal{A}} (\psi_b - \psi_c)^\top Q_\delta(\widetilde{p}_t)^\dagger (\psi_b - \psi_c)\right)\,,$$

where the first and second inequalities again follow from Jensen's inequality and the definition of $\omega$ respectively. The required result now follows from Lemma 1. □

**Lemma 3.** *For any $\eta > 0$ and $q \in \Delta_k$, it holds that*

$$\Lambda_{\eta,q}^* = \max_{\ell \in \mathcal{L}} \min_{p \in \Xi_\eta} \Lambda_{\eta,q}(p, \ell)\,.$$

*Proof.* Recall that

$$\Lambda_{\eta,q}(p,\ell) := \frac{1}{\eta}\langle p - q, H^\top \ell \rangle + L^2 \sum_{a,b\in\mathcal{A}} q(a)q(b)\mathcal{E}(a,b;p)$$

for $p \in \Delta_k$, and $\ell \in \mathcal{L}$; and that for any $a, b \in \mathcal{A}$,

$$\mathcal{E}(a,b;p) := (\psi_a - \psi_b)^\top Q_\delta(p)^\dagger (\psi_a - \psi_b).$$

Also, recall that $\Xi_\eta := \{p \in \Delta_k \colon z(p) \leq {}^2/_{\eta L}\}$, where

$$z(p) := \max_{a,b\in\mathcal{A}} \mathcal{E}(a,b;p) + \max_{c\in\mathcal{A}}\|M_c^\top Q_\delta(p)^\dagger M_c\|_2,$$

and that $\Lambda_{\eta,q}^* := \min_{p\in\Xi_\eta} \max_{\ell\in\mathcal{L}} \Lambda_{\eta,q}(p,\ell)$.

We start by showing that for a fixed pair $(a,b) \in \mathcal{A}^2$, $\mathcal{E}(a,b;p)$ is convex in $p$. Via Lemma 11 in Appendix C, $\mathrm{col}(Q_\delta(p)) = \mathrm{col}(M)$ for any $p \in \Delta_k$. Hence, the convexity of $\mathcal{E}$ in $p$ follows from Lemma 6 in Appendix C and the fact that $Q_\delta(p)$ is affine in $p$. Moreover, for a fixed $c \in \mathcal{A}$, the function $p \mapsto \|M_c^\top Q_\delta(p)^\dagger M_c\|_2$ can also be easily shown to be convex by invoking once again Lemma 6 and using that $Q_\delta(p)$ is affine in $p$ and that the spectral norm is convex and non-decreasing in the Loewner order. Also note that both functions are continuous as the rank of $Q_\delta(p)$ is constant for any $p \in \Delta_k$ (see Corollary 3.5 in [52]), or simply via the continuity of matrix inversion since $Q_\delta(p)^\dagger = U B_\delta(p)^{-1} U^\top$ and $B_\delta(p)$ (defined in (8)) is positive definite for any $p \in \Delta_k$ as asserts Lemma 11 in Appendix C. Hence, the function $z$ is continuous and convex in $p \in \Delta_k$, and its sub-level set $\Xi_\eta := \{p \in \Delta_k \colon z(p) \leq {}^2/_{\eta L}\}$ is convex and compact.

The arguments above imply that for a fixed $\ell \in \mathcal{L}$, $p \mapsto \Lambda_{\eta,q}(p,\ell)$ is continuous and convex over the compact set $\Xi_\eta$. Moreover, for a fixed $p \in \Delta_k$, $\ell \mapsto \Lambda_{\eta,q}(p,\ell)$ is trivially continuous and concave over the compact set $\mathcal{L}$. Therefore, Sion's minimax theorem [51] asserts that

$$\min_{p\in\Xi_\eta} \max_{\ell\in\mathcal{L}} \Lambda_{\eta,q}(p,\ell) = \max_{\ell\in\mathcal{L}} \min_{p\in\Xi_\eta} \Lambda_{\eta,q}(p,\ell).$$

The theorem then follows from the definition of $\Lambda_{\eta,q}^*$. $\qquad\square$

# E Proofs of Section 4

**Lemma 4.** *A game is locally observable if and only if it holds for all $a \in \mathcal{A}^*$, $\ell \in \mathbb{R}^d$, and $a^* \in \arg\min_{a'\in\mathcal{A}} \psi_{a'}^\top\ell$ that $\psi_a - \psi_{a^*} \in \mathrm{span}\{M_b \mid \psi_b^\top\ell \leq \psi_a^\top\ell\}$.*

*Proof.* This follows from Lemma 5 in Appendix B. $\qquad\square$

Before moving on to the proof of Theorem 1, we define an analogous set to $\mathcal{W}^{\mathrm{loc}}$ that does not take the losses into account. Recall that global observability implies that for any pair of actions $(a,b)$, there exists a (not necessarily unique) weight vector $v \in \mathbb{R}^n$ such that $\psi_a - \psi_b = Mv$. The following is the set of all possible action-pair-wise assignments of such weight vectors:

$$\mathcal{W}^{\mathrm{glo}} := \left\{\boldsymbol{\xi} = (\boldsymbol{\xi}_{a,b})_{a,b\in\mathcal{A}} \in \mathbb{R}^{k^2\times n} \mid \forall a,b \in \mathcal{A}, \ \psi_a - \psi_b = M\boldsymbol{\xi}_{a,b}\right\}.$$

Recall that

$$\beta_{\mathrm{glo}} := \max_{a,b\in\mathcal{A}} \min_{v\in\mathbb{R}^n \colon \psi_a - \psi_b = Mv} \sum_{c\in\mathcal{A}}\|v(c)\|.$$

The following lemma provides a slightly different characterization of $\beta_{\mathrm{glo}}$ featuring $\mathcal{W}^{\mathrm{glo}}$.

**Lemma 16.** *It holds that*

$$\beta_{\mathrm{glo}} = \min_{\boldsymbol{\xi}\in\mathcal{W}^{\mathrm{glo}}} \max_{a,b\in\mathcal{A}} \sum_{c\in\mathcal{A}}\|\boldsymbol{\xi}_{a,b}(c)\|.$$

*Proof.* For $a,b \in \mathcal{A}$, let $v_{a,b}^*$ refer to an arbitrary member of $\arg\min_{v\in\mathbb{R}^n \colon \psi_a - \psi_b = Mv} \sum_{c\in\mathcal{A}}\|v(c)\|$. Via the definition of $\mathcal{W}^{\mathrm{glo}}$, $(v_{a,b}^*)_{a,b\in\mathcal{A}} \in \mathcal{W}^{\mathrm{glo}}$; moreover, it holds that $(v_{a,b}^*)_{a,b\in\mathcal{A}} \in \arg\min_{\boldsymbol{\xi}\in\mathcal{W}^{\mathrm{glo}}} \sum_{c\in\mathcal{A}}\|\boldsymbol{\xi}_{a,b}(c)\|$ for all $a,b \in \mathcal{A}$ simultaneously. This implies that $(v_{a,b}^*)_{a,b\in\mathcal{A}} \in \arg\min_{\boldsymbol{\xi}\in\mathcal{W}^{\mathrm{glo}}} \max_{a,b\in\mathcal{A}} \sum_{c\in\mathcal{A}}\|\boldsymbol{\xi}_{a,b}(c)\|$, concluding the proof. $\qquad\square$

The following lemma relates $\beta_{\mathrm{glo}}$ and $\beta_{\mathrm{loc}}$.

**Lemma 17.** *It holds that* $\beta_{\mathrm{glo}} \le \beta_{\mathrm{loc}}$.

*Proof.* Recall that

$$\beta_{\mathrm{loc}} := \max_{\ell \in \mathcal{L}} \min_{\boldsymbol{\lambda} \in \mathcal{W}_\ell^{\mathrm{loc}}} \max_{a \in \mathcal{A}^*} \sum_{c \in \mathcal{A}} \|\boldsymbol{\lambda}_a(c)\| \,,$$

where

$$\mathcal{W}_\ell^{\mathrm{loc}} := \Big\{ \boldsymbol{\lambda} = (\boldsymbol{\lambda}_a)_{a \in \mathcal{A}^*} \in \mathbb{R}^{k^* \times n} \mid \exists a^* \in \arg\min_{c \in \mathcal{A}} \psi_c^\top \ell \; \forall (a,b) \in \mathcal{A}^* \times \mathcal{A} \,,$$
$$\psi_a - \psi_{a^*} = M \boldsymbol{\lambda}_a \text{ and } \boldsymbol{\lambda}_a(b) \ne \mathbf{0} \Rightarrow \psi_b^\top \ell \le \psi_a^\top \ell \Big\} \,.$$

Additionally, define

$$\mathcal{W}_\ell^{\mathrm{glo}} := \Big\{ \boldsymbol{\lambda} = (\boldsymbol{\lambda}_a)_{a \in \mathcal{A}^*} \in \mathbb{R}^{k^* \times n} \mid \exists a^* \in \arg\min_{c \in \mathcal{A}} \psi_c^\top \ell \; \forall a \in \mathcal{A}^* \,, \psi_a - \psi_{a^*} = M \boldsymbol{\lambda}_a \Big\}$$

for $\ell \in \mathcal{L}$ and, with some notation abuse,

$$\mathcal{W}_{a'}^{\mathrm{glo}} := \Big\{ \boldsymbol{\lambda} = (\boldsymbol{\lambda}_a)_{a \in \mathcal{A}^*} \in \mathbb{R}^{k^* \times n} \mid \forall a \in \mathcal{A}^* \,, \psi_a - \psi_{a'} = M \boldsymbol{\lambda}_a \Big\}$$

for $a' \in \mathcal{A}^*$. Then,

$$\beta_{\mathrm{loc}} = \max_{\ell \in \mathcal{L}} \min_{\boldsymbol{\lambda} \in \mathcal{W}_\ell^{\mathrm{loc}}} \max_{a \in \mathcal{A}^*} \sum_{c \in \mathcal{A}} \|\boldsymbol{\lambda}_a(c)\|$$

$$\ge \max_{\ell \in \mathcal{L}} \min_{\boldsymbol{\lambda} \in \mathcal{W}_\ell^{\mathrm{glo}}} \max_{a \in \mathcal{A}^*} \sum_{c \in \mathcal{A}} \|\boldsymbol{\lambda}_a(c)\|$$

$$\ge \max_{b \in \mathcal{A}^*} \min_{\boldsymbol{\lambda} \in \mathcal{W}_b^{\mathrm{glo}}} \max_{a \in \mathcal{A}^*} \sum_{c \in \mathcal{A}} \|\boldsymbol{\lambda}_a(c)\|$$

$$\ge \max_{b \in \mathcal{A}^*} \max_{a \in \mathcal{A}^*} \min_{\boldsymbol{\lambda} \in \mathcal{W}_b^{\mathrm{glo}}} \sum_{c \in \mathcal{A}} \|\boldsymbol{\lambda}_a(c)\|$$

$$= \max_{a,b \in \mathcal{A}^*} \min_{\boldsymbol{v} \in \mathbb{R}^n : \psi_a - \psi_b = M \boldsymbol{v}} \sum_{c \in \mathcal{A}} \|\boldsymbol{v}(c)\| \,,$$

where the first inequality holds since $\mathcal{W}_\ell^{\mathrm{loc}} \subseteq \mathcal{W}_\ell^{\mathrm{glo}}$. Concerning the second inequality, as argued in the proof of Lemma 5, it holds for any Pareto optimal action $b$ that $\{\psi_a\} \cap \mathrm{co}(\mathcal{X} \setminus \{\psi_a\}) = \varnothing$ (since $\psi_b \in \mathrm{ext}(\mathcal{V})$); hence, by the separating hyperplane theorem, there exists a loss vector $\ell_b \in \mathbb{R}^d$ such that $\psi_b^\top \ell_b < x^\top \ell_b$ for all $x \in \mathcal{X} \setminus \{\psi_b\}$, yielding that only $b$ and its duplicates can minimize $c \mapsto \psi_c^\top \ell_b$ over $\mathcal{A}$. The inequality then follows by restricting the maximization to any selection of loss vectors of the form $(\ell_b)_{b \in \mathcal{A}^*}$ and using the fact that $\mathcal{W}_{\ell_b}^{\mathrm{glo}} = \mathcal{W}_b^{\mathrm{glo}}$.

On the other hand, we have that

$$\beta_{\mathrm{glo}} = \max_{a,b \in \mathcal{A}} \min_{\boldsymbol{v} \in \mathbb{R}^n : \psi_a - \psi_b = M \boldsymbol{v}} \sum_{c \in \mathcal{A}} \|\boldsymbol{v}(c)\| \,;$$

hence, showing that the maximum on the right-hand side of the equation above is attained by a pair $a, b \in \mathcal{A}^*$ suffices to conclude the proof. Let $a, b \in \mathcal{A}$ be two arbitrary actions. By the definition of $\mathcal{A}^*$, there exist two probability distributions $p_1, p_2 \in \Delta_{k^*}$ such that $\psi_a = \sum_{h \in \mathcal{A}^*} p_1(h) \psi_h$ and $\psi_b = \sum_{h \in \mathcal{A}^*} p_2(h) \psi_h$. Then,

$$\psi_a - \psi_b = \sum_{h \in \mathcal{A}^*} p_1(h) \psi_h - \sum_{j \in \mathcal{A}^*} p_2(j) \psi_j = \sum_{h,j \in \mathcal{A}^*} p_1(h) p_2(j) (\psi_h - \psi_j) \,,$$

and $\psi_a - \psi_b = M \sum_{h,j \in \mathcal{A}^*} p_1(h) p_2(j) \boldsymbol{v}_{h,j}^*$, with $\boldsymbol{v}_{h,j}^* \in \arg\min_{\boldsymbol{v} \in \mathbb{R}^n \,:\, \psi_h - \psi_j = M\boldsymbol{v}} \sum_{c \in \mathcal{A}} \|\boldsymbol{v}(c)\|$.
Hence,

$$
\begin{aligned}
\min_{\boldsymbol{v} \in \mathbb{R}^n \,:\, \psi_a - \psi_b = M\boldsymbol{v}} \sum_{c \in \mathcal{A}} \|\boldsymbol{v}(c)\| &\leq \sum_{c \in \mathcal{A}} \big\| \textstyle\sum_{h,j \in \mathcal{A}^*} p_1(h) p_2(j) \boldsymbol{v}_{h,j}^*(c) \big\| \\
&\leq \sum_{h,j \in \mathcal{A}^*} p_1(h) p_2(j) \sum_{c \in \mathcal{A}} \|\boldsymbol{v}_{h,j}^*(c)\| \\
&\leq \max_{h,j \in \mathcal{A}^*} \sum_{c \in \mathcal{A}} \|\boldsymbol{v}_{h,j}^*(c)\| \\
&= \max_{h,j \in \mathcal{A}^*} \min_{\boldsymbol{v} \in \mathbb{R}^n \,:\, \psi_h - \psi_j = M\boldsymbol{v}} \sum_{c \in \mathcal{A}} \|\boldsymbol{v}(c)\| \,,
\end{aligned}
$$

where the second inequality is an application of Jensen's inequality. We have thus shown that

$$
\beta_{\mathrm{glo}} = \max_{a,b \in \mathcal{A}^*} \min_{\boldsymbol{v} \in \mathbb{R}^n \,:\, \psi_a - \psi_b = M\boldsymbol{v}} \sum_{c \in \mathcal{A}} \|\boldsymbol{v}(c)\| \leq \beta_{\mathrm{loc}} \,.
$$

$\square$

The proof of Theorem 1 relies on the following Proposition. (The definition of $B(\cdot)$ is given in (8).)

**Proposition 3.** *Suppose that the game is locally observable. Then, for any $\ell \in \mathcal{L}$, $q \in \Delta_k^*$, $\boldsymbol{\lambda} \in \mathcal{W}_\ell^{\mathrm{loc}}$, and $\boldsymbol{\xi} \in \mathcal{W}^{\mathrm{glo}}$, it holds that*

$$
\min_{p \in \Xi_\eta} \Lambda_{\eta,q}(p, \ell) \leq 8L^2 \beta_{\boldsymbol{\lambda}}^2 \sum_{b \in \mathcal{A}} q_{-}^{\boldsymbol{\lambda}}(b) \big\| M_b^\top Q(q^{\boldsymbol{\lambda}})^\dagger M_b \big\|
$$

$$
+ 2L\big(1 + \beta_{\boldsymbol{\xi}}^2\big) \min_{\pi \in \Delta_k} \max_{c \in \mathcal{A}} \big\| M_c^\top U B(\pi)^{-1} U^\top M_c \big\|
$$

*provided that $1/\eta \geq 2L\big(1 + \beta_{\boldsymbol{\xi}}^2\big) \min_{\pi \in \Delta_k} \max_{c \in \mathcal{A}} \big\| M_c^\top U B(\pi)^{-1} U^\top M_c \big\|$, where*

$$
\beta_{\boldsymbol{\lambda}} := \max_{a \in \mathcal{A}^*} \sum_{b \in \mathcal{A}} \|\boldsymbol{\lambda}_a(b)\| \,, \quad \beta_{\boldsymbol{\xi}} := \max_{a,b \in \mathcal{A}} \sum_{c \in \mathcal{A}} \|\boldsymbol{\xi}_{a,b}(c)\| \,, \quad \text{and}
$$

$$
q^{\boldsymbol{\lambda}}(b) := q(b) \, \llbracket b \in \mathcal{A}^*, \boldsymbol{\lambda}_b = \boldsymbol{0} \rrbracket + \underbrace{\sum_{a \in \mathcal{A}^* \,:\, \boldsymbol{\lambda}_a \neq \boldsymbol{0}} q(a) \frac{\|\boldsymbol{\lambda}_a(b)\|}{\sum_{c \in \mathcal{A}} \|\boldsymbol{\lambda}_a(c)\|}}_{=: \, q_{-}^{\boldsymbol{\lambda}}(b)} \,.
$$

*Proof.* We show this by constructing a certain distribution $p \in \Xi_\eta$ and bounding $\Lambda_{\eta,q}(p, \ell)$. Our choice of $p$ will take the form $p = \gamma \pi + (1 - \gamma)\widehat{p}$, with $\gamma \in (0, 1/2]$ and $\pi, \widehat{p} \in \Delta_k$. For brevity, let $\bar{\gamma} = 1 - \gamma$. The role of $\pi$ is to ensure that $\gamma \pi + \bar{\gamma}\widehat{p} \in \Xi_\eta$ and that of $\widehat{p}$ is to control the magnitude of $\Lambda_{\eta,q}(\gamma \pi + \bar{\gamma}\widehat{p}, \ell)$. Let $a^* \in \arg\min_{a' \in \mathcal{A}^*} \langle \psi_{a'}, \ell \rangle$. Now,

$$
\begin{aligned}
\Lambda_{\eta,q}(\gamma \pi + \bar{\gamma}\widehat{p}, \ell) &= \frac{\gamma}{\eta} \langle \pi - q, H^\top \ell \rangle + \frac{\bar{\gamma}}{\eta} \langle \widehat{p} - q, H^\top \ell \rangle + L^2 \sum_{a,b \in \mathcal{A}} q(a) q(b) \mathcal{E}(a, b; \gamma \pi + \bar{\gamma}\widehat{p}) \\
&\leq \frac{2\gamma}{\eta} + \frac{\bar{\gamma}}{\eta} \langle \widehat{p} - q, H^\top \ell \rangle + L^2 \sum_{a,b \in \mathcal{A}} q(a) q(b) \mathcal{E}(a, b; \gamma \pi + \bar{\gamma}\widehat{p}) \\
&\leq \frac{2\gamma}{\eta} + \frac{\bar{\gamma}}{\eta} \langle \widehat{p} - q, H^\top \ell \rangle + 2L^2 \sum_{a \in \mathcal{A}} q(a) \mathcal{E}(a, a^*; \gamma \pi + \bar{\gamma}\widehat{p}) \,, \quad (10)
\end{aligned}
$$

where the first inequality uses the assumption that $\max_{a,b \in \mathcal{A}, \ell \in \mathcal{L}} |(\psi_a - \psi_b)^\top \ell| \leq 2$, and the second inequality follows from the definition of $\mathcal{E}$, Lemma 13, and Lemma 14 (noting that $\mathcal{E}(a, a^*; p) = (\psi_a - He_{a^*})^\top Q_\delta(p)^\dagger (\psi_a - He_{a^*})$). We now proceed with the selection of $\widehat{p}$ with the purpose of minimizing the second and third terms. A convenient choice is to select $\widehat{p}$ so as to perform better than $q$ against the loss vector $\ell$. For that, it suffices to consider $\widehat{p}$ of the form

$$
\widehat{p} = \sum_{a \in \mathcal{A}^*} q(a) \nu_a \quad \text{such that} \quad \forall a \in \mathcal{A}^* \, \langle \nu_a - e_a, H^\top \ell \rangle \leq 0 \,, \quad (11)
$$

where $\nu_a \in \Delta_k$. That is, $\widehat{p}$ is a mixture (with weights proportional to $q$) of $|\mathcal{A}^*|$ distributions, each associated with a Pareto optimal action and is chosen to outperform this action against $\ell$. This choice allows us to simultaneously eliminate the second term in (10) and bound the third term independently of tunable parameters. The former fact is easily demonstrated:

$$\frac{\bar{\gamma}}{\eta}\langle \widehat{p} - q, H^\top \ell \rangle = \frac{\bar{\gamma}}{\eta} \sum_{a \in \mathcal{A}^*} q(a)\langle \nu_a - e_a, H^\top \ell \rangle \leq 0 \,. \tag{12}$$

Moving over to the third term, fix $\boldsymbol{\lambda} \in \mathcal{W}_\ell^{\mathrm{loc}}$ such that $\psi_a - \psi_{a^*} = \boldsymbol{M}\boldsymbol{\lambda}_a$ for all $a \in \mathcal{A}^*$, which exists via local observability. Then, for any $a \in \mathcal{A}^*$ and any $p \in \Delta_k$, we have that

$$\begin{aligned}
\mathcal{E}(a, a^*; p) &= (\psi_a - \psi_{a^*})^\top Q_\delta(p)^\dagger (\psi_a - \psi_{a^*}) \\
&= \left\| \left(Q_\delta(p)^\dagger\right)^{1/2}(\psi_a - \psi_{a^*}) \right\|^2 \\
&= \left\| \left(Q_\delta(p)^\dagger\right)^{1/2}\boldsymbol{M}\boldsymbol{\lambda}_a \right\|^2 \\
&= \left\| \sum_{c \in \mathcal{A}} \left(Q_\delta(p)^\dagger\right)^{1/2} M_c \boldsymbol{\lambda}_a(c) \right\|^2 \\
&\leq \left( \sum_{c \in \mathcal{A}} \left\| \left(Q_\delta(p)^\dagger\right)^{1/2} M_c \boldsymbol{\lambda}_a(c) \right\| \right)^2 \\
&\leq \left( \sum_{c \in \mathcal{A}} \left\| \left(Q_\delta(p)^\dagger\right)^{1/2} M_c \right\| \|\boldsymbol{\lambda}_a(c)\| \right)^2 \\
&\leq \left( \sum_{c \in \mathcal{A}} \|\boldsymbol{\lambda}_a(c)\| \right) \sum_{c \in \mathcal{A}} \|\boldsymbol{\lambda}_a(c)\| \left\| \left(Q_\delta(p)^\dagger\right)^{1/2} M_c \right\|^2 \,, \tag{13}
\end{aligned}$$

where the first inequality is an application of the triangle inequality, the second a consequence of the definition of the spectral norm, and the third an application of the Cauchy-Schwarz inequality. Hence,

$$\begin{aligned}
\sum_{a \in \mathcal{A}} q(a)\mathcal{E}(a, a^*; p) &= \sum_{a \in \mathcal{A}^*} q(a)\mathcal{E}(a, a^*; p) \\
&\leq \sum_{a \in \mathcal{A}^*} q(a)\left(\sum_{c \in \mathcal{A}}\|\boldsymbol{\lambda}_a(c)\|\right) \sum_{c \in \mathcal{A}} \|\boldsymbol{\lambda}_a(c)\| \left\| \left(Q_\delta(p)^\dagger\right)^{1/2} M_c \right\|^2 \tag{14} \\
&\leq \max_{s \in \mathcal{A}^*}\left(\sum_{c \in \mathcal{A}}\|\boldsymbol{\lambda}_s(c)\|\right) \sum_{a \in \mathcal{A}^*} q(a) \sum_{c \in \mathcal{A}} \|\boldsymbol{\lambda}_a(c)\| \left\| \left(Q_\delta(p)^\dagger\right)^{1/2} M_c \right\|^2 \\
&= \max_{s \in \mathcal{A}^*}\left(\sum_{c \in \mathcal{A}}\|\boldsymbol{\lambda}_s(c)\|\right) \sum_{c \in \mathcal{A}} \sum_{a \in \mathcal{A}^*} q(a)\|\boldsymbol{\lambda}_a(c)\| \left\| \left(Q_\delta(p)^\dagger\right)^{1/2} M_c \right\|^2 \\
&= \max_{s \in \mathcal{A}^*}\left(\sum_{c \in \mathcal{A}}\|\boldsymbol{\lambda}_s(c)\|\right)^2 \sum_{c \in \mathcal{A}} \sum_{a \in \mathcal{A}^*} q(a)\frac{\|\boldsymbol{\lambda}_a(c)\|}{\max_{s \in \mathcal{A}^*}\sum_{b \in \mathcal{A}}\|\boldsymbol{\lambda}_s(b)\|} \left\| \left(Q_\delta(p)^\dagger\right)^{1/2} M_c \right\|^2 \\
&\leq \max_{s \in \mathcal{A}^*}\left(\sum_{c \in \mathcal{A}}\|\boldsymbol{\lambda}_s(c)\|\right)^2 \sum_{c \in \mathcal{A}} \sum_{a \in \mathcal{A}^* : \boldsymbol{\lambda}_a \neq \mathbf{0}} q(a)\frac{\|\boldsymbol{\lambda}_a(c)\|}{\sum_{b \in \mathcal{A}}\|\boldsymbol{\lambda}_a(b)\|} \left\| \left(Q_\delta(p)^\dagger\right)^{1/2} M_c \right\|^2 \,, \tag{15}
\end{aligned}$$

where the first equality holds since $q$ is only supported on $\mathcal{A}^*$. Note that the right-hand-side of (15) now offers a natural way to choose $\widehat{p}$ as to respect the constraint in (11) and simultaneously simplify the bound. Simply, let $\nu_a \in \Delta_k$ (as referred to in (11)) be such that $\nu_a(b) \propto \|\boldsymbol{\lambda}_a(b)\|$ for $b \in \mathcal{A}$ if $\boldsymbol{\lambda}_a \neq \mathbf{0}$; otherwise, if $\boldsymbol{\lambda}_a = \mathbf{0}$ (which is the case for $a^*$ and its duplicates), simply set $\nu_a = e_a$. This choice satisfies the condition in (11) since the fact that $\boldsymbol{\lambda} \in \mathcal{W}_\ell^{\mathrm{loc}}$ implies that any action in the support of $\nu_a$ has loss smaller than or equal to that of $a$. To emphasize its dependence on $\boldsymbol{\lambda}$ and $q$, we denote this choice of $\widehat{p}$ by $q^{\boldsymbol{\lambda}}$. For clarity, we state its definition explicitly below: for $b \in \mathcal{A}$, let

$$\begin{aligned}
q^{\boldsymbol{\lambda}}(b) &:= \sum_{a \in \mathcal{A}^*} q(a)\left( \llbracket \boldsymbol{\lambda}_a = \mathbf{0}, a = b \rrbracket + \llbracket \boldsymbol{\lambda}_a \neq \mathbf{0} \rrbracket \frac{\|\boldsymbol{\lambda}_a(b)\|}{\sum_{c \in \mathcal{A}}\|\boldsymbol{\lambda}_a(c)\|} \right) \\
&= q(b)\,\llbracket b \in \mathcal{A}^*, \boldsymbol{\lambda}_b = \mathbf{0} \rrbracket + \underbrace{\sum_{a \in \mathcal{A}^* : \boldsymbol{\lambda}_a \neq \mathbf{0}} q(a)\frac{\|\boldsymbol{\lambda}_a(b)\|}{\sum_{c \in \mathcal{A}}\|\boldsymbol{\lambda}_a(c)\|}}_{=: \, q_-^{\boldsymbol{\lambda}}(b)} \,.
\end{aligned}$$

For convenience, we have also defined above an abridged mixture $q_-^{\boldsymbol{\lambda}}$, which excludes actions $a \in \mathcal{A}^*$ for which $\boldsymbol{\lambda}_a = \mathbf{0}$. Now, with $\beta_{\boldsymbol{\lambda}} := \max_{s \in \mathcal{A}^*} \sum_{c \in \mathcal{A}} \|\boldsymbol{\lambda}_s(c)\|$, (15) implies that

$$
\begin{aligned}
\sum_{a \in \mathcal{A}} q(a) \mathcal{E}(a, a^*; \gamma\pi + \bar{\gamma}q^{\boldsymbol{\lambda}}) &\leq \beta_{\boldsymbol{\lambda}}^2 \sum_{b \in \mathcal{A}} q_-^{\boldsymbol{\lambda}}(b) \left\| \left(Q_\delta(\gamma\pi + \bar{\gamma}q^{\boldsymbol{\lambda}})^\dagger\right)^{1/2} M_b \right\|^2 \\
&= \beta_{\boldsymbol{\lambda}}^2 \sum_{b \in \mathcal{A}} q_-^{\boldsymbol{\lambda}}(b) \left\| M_b^\top Q_\delta(\gamma\pi + \bar{\gamma}q^{\boldsymbol{\lambda}})^\dagger M_b \right\| \\
&\leq \frac{\beta_{\boldsymbol{\lambda}}^2}{(1-\delta)(1-\gamma)} \sum_{b \in \mathcal{A}} q_-^{\boldsymbol{\lambda}}(b) \left\| M_b^\top Q(q^{\boldsymbol{\lambda}})^\dagger M_b \right\| \\
&\leq 4\beta_{\boldsymbol{\lambda}}^2 \sum_{b \in \mathcal{A}} q_-^{\boldsymbol{\lambda}}(b) \left\| M_b^\top Q(q^{\boldsymbol{\lambda}})^\dagger M_b \right\|,
\end{aligned}
$$

where the penultimate step is an application of Lemma 7 using the definition of $Q_\delta$ and the fact that $\mathrm{col}(M_b) \subseteq \mathrm{col}(Q(q^{\boldsymbol{\lambda}}))$ whenever $q^{\boldsymbol{\lambda}}(b) > 0$ (see Lemma 8), and the last step uses that $\delta, \gamma \leq 1/2$. Hence, combined with (10) and (12), this entails that

$$
\Lambda_{\eta,q}(\gamma\pi + \bar{\gamma}q^{\boldsymbol{\lambda}}, \ell) \leq \frac{2\gamma}{\eta} + 8L^2 \beta_{\boldsymbol{\lambda}}^2 \sum_{b \in \mathcal{A}} q_-^{\boldsymbol{\lambda}}(b) \left\| M_b^\top Q(q^{\boldsymbol{\lambda}})^\dagger M_b \right\|. \tag{16}
$$

The smallest admissible value for $\gamma$ depends on the choice of $\pi$, which should be chosen so as to ensure that $(\gamma\pi + \bar{\gamma}q^{\boldsymbol{\lambda}}) \in \Xi_\eta$; that is, $z(\gamma\pi + \bar{\gamma}q^{\boldsymbol{\lambda}}) \leq 2/\eta L$. Now, fix $\boldsymbol{\xi} \in \mathcal{W}^{\mathrm{glo}}$ and define $\beta_{\boldsymbol{\xi}} := \max_{a,b \in \mathcal{A}} \sum_{c \in \mathcal{A}} \|\boldsymbol{\xi}_{a,b}(c)\|$. Analogously to (13), we have that for any $a, b \in \mathcal{A}$,

$$
\begin{aligned}
\mathcal{E}(a, b; \gamma\pi + \bar{\gamma}q^{\boldsymbol{\lambda}}) &\leq \left( \sum_{c \in \mathcal{A}} \|\boldsymbol{\xi}_{a,b}(c)\| \right) \sum_{c \in \mathcal{A}} \|\boldsymbol{\xi}_{a,b}(c)\| \left\| \left(Q_\delta(\gamma\pi + \bar{\gamma}q^{\boldsymbol{\lambda}})^\dagger\right)^{1/2} M_c \right\|^2 \\
&\leq \left( \sum_{c \in \mathcal{A}} \|\boldsymbol{\xi}_{a,b}(c)\| \right)^2 \max_{c \in \mathcal{A}} \left\| \left(Q_\delta(\gamma\pi + \bar{\gamma}q^{\boldsymbol{\lambda}})^\dagger\right)^{1/2} M_c \right\|^2 \\
&\leq \beta_{\boldsymbol{\xi}}^2 \max_{c \in \mathcal{A}} \left\| \left(Q_\delta(\gamma\pi + \bar{\gamma}q^{\boldsymbol{\lambda}})^\dagger\right)^{1/2} M_c \right\|^2. \tag{17}
\end{aligned}
$$

Then, enforcing that $\mathrm{col}(Q(\pi)) = \mathrm{col}(\boldsymbol{M})$, we get that

$$
\begin{aligned}
z(\gamma\pi + \bar{\gamma}q^{\boldsymbol{\lambda}}) &= \max_{a,b \in \mathcal{A}} \mathcal{E}(a, b; \gamma\pi + \bar{\gamma}q^{\boldsymbol{\lambda}}) + \max_{c \in \mathcal{A}} \left\| \left(Q_\delta(\gamma\pi + \bar{\gamma}q^{\boldsymbol{\lambda}})^\dagger\right)^{1/2} M_c \right\|^2 \\
&\leq \left( 1 + \beta_{\boldsymbol{\xi}}^2 \right) \max_{c \in \mathcal{A}} \left\| \left(Q_\delta(\gamma\pi + \bar{\gamma}q^{\boldsymbol{\lambda}})^\dagger\right)^{1/2} M_c \right\|^2 \\
&\leq \frac{1}{(1-\delta)\gamma} \left( 1 + \beta_{\boldsymbol{\xi}}^2 \right) \max_{c \in \mathcal{A}} \left\| \left(Q(\pi)^\dagger\right)^{1/2} M_c \right\|^2 \\
&\leq \frac{2}{\gamma} \left( 1 + \beta_{\boldsymbol{\xi}}^2 \right) \max_{c \in \mathcal{A}} \left\| \left(Q(\pi)^\dagger\right)^{1/2} M_c \right\|^2 \\
&= \frac{2}{\gamma} \left( 1 + \beta_{\boldsymbol{\xi}}^2 \right) \max_{c \in \mathcal{A}} \left\| \left(U B(\pi)^{-1} U^\top\right)^{1/2} M_c \right\|^2, \tag{18}
\end{aligned}
$$

where the second inequality follows from Lemma 7 as $\mathrm{col}(M_b) \subseteq \mathrm{col}(Q(\pi))$ by the assumption on $\pi$, the third inequality holds since $\delta \leq 1/2$, and the last equality also follows from the assumption on $\pi$ (see Lemmas 9 and 10). At this junction, we pick

$$
\pi \in \arg\min_{\pi' \in \Delta_k} \max_{c \in \mathcal{A}} \left\| \left(U B(\pi')^{-1} U^\top\right)^{1/2} M_c \right\|^2,
$$

which satisfies that $\mathrm{col}(Q(\pi)) = \mathrm{col}(\boldsymbol{M})$ as this is equivalent to $B(\pi)$ being non-singular, see Lemma 9. With this choice of $\pi$, equating the right-hand side of (18) to $2/\eta L$ dictates setting

$$
\gamma = \eta L \left( 1 + \beta_{\boldsymbol{\xi}}^2 \right) \min_{\pi' \in \Delta_k} \max_{c \in \mathcal{A}} \left\| \left(U B(\pi')^{-1} U^\top\right)^{1/2} M_c \right\|^2.
$$

To ensure that this choice of $\gamma$ is valid (i.e., $\gamma \leq 1/2$), we enforce that

$$
1/\eta \geq 2L \left( 1 + \beta_{\boldsymbol{\xi}}^2 \right) \min_{\pi' \in \Delta_k} \max_{c \in \mathcal{A}} \left\| \left(U B(\pi')^{-1} U^\top\right)^{1/2} M_c \right\|^2.
$$

Plugging this value of $\gamma$ back in (16) finally yields that

$$\min_{(p,r)\in\Xi_\eta}\Lambda_{\eta,q}(p,r,\ell) \leq 8L^2\beta_{\boldsymbol{\lambda}}^2\sum_{b\in\mathcal{A}}q_-^{\boldsymbol{\lambda}}(b)\big\|M_b^\top Q(q^{\boldsymbol{\lambda}})^\dagger M_b\big\|$$
$$+ 2L\big(1+\beta_{\boldsymbol{\xi}}^2\big)\min_{\pi'\in\Delta_k}\max_{c\in\mathcal{A}}\big\|\big(UB(\pi')^{-1}U^\top\big)^{1/2}M_c\big\|^2.$$

$\square$

**Theorem 1.** *In locally observable games, it holds that*

$$\Lambda_\eta^* \leq \max_{\ell\in\mathcal{L}}\inf_{\boldsymbol{\lambda}\in\mathcal{W}_\ell^{\mathrm{loc}}} 8L^2\beta_{\boldsymbol{\lambda}}^2\min\{\mathrm{rk}(\boldsymbol{M}),|\operatorname{supp}(\boldsymbol{\lambda})|\} + 2L\big(1+\beta_{\mathrm{glo}}^2\big)\min\{\mathrm{rk}(\boldsymbol{M}),w^*\}$$

*provided* $\dfrac{1}{\eta} \geq 2L\big(1+\beta_{\mathrm{glo}}^2\big)\min\{\mathrm{rk}(\boldsymbol{M}),w^*\}$, *where* $\beta_{\mathrm{glo}} := \max_{a,b\in\mathcal{A}}\min_{\boldsymbol{v}\in\mathbb{R}^n:\,\psi_a-\psi_b=\boldsymbol{M}\boldsymbol{v}}\sum_{c\in\mathcal{A}}\|\boldsymbol{v}(c)\|$

$$\text{and}\quad w^* := \min_{S\subseteq\mathcal{A}}|S|\max_{b\in\mathcal{A}}\big\|M_b^\top U\big(\textstyle\sum_{s\in S}U^\top M_s M_s^\top U\big)^{-1}U^\top M_b\big\| \leq k.$$

*Proof.* Fix $\ell\in\mathcal{L}$, $q\in\Delta_k^*$, and $\boldsymbol{\lambda}\in\mathcal{W}_\ell^{\mathrm{loc}}$. Concerning the first term in the bound of Proposition 3, we have that

$$\begin{aligned}
\textstyle\sum_{b\in\mathcal{A}}q_-^{\boldsymbol{\lambda}}(b)\big\|M_b^\top Q(q^{\boldsymbol{\lambda}})^\dagger M_b\big\| &\leq \textstyle\sum_{b\in\mathcal{A}}q^{\boldsymbol{\lambda}}(b)\big\|M_b^\top Q(q^{\boldsymbol{\lambda}})^\dagger M_b\big\| \\
&\leq \textstyle\sum_{b\in\mathcal{A}}q^{\boldsymbol{\lambda}}(b)\operatorname{tr}\big(M_b^\top Q(q^{\boldsymbol{\lambda}})^\dagger M_b\big) \\
&= \textstyle\sum_{b\in\mathcal{A}}q^{\boldsymbol{\lambda}}(b)\operatorname{tr}\big(Q(q^{\boldsymbol{\lambda}})^\dagger M_b M_b^\top\big) \\
&= \operatorname{tr}\big(Q(q^{\boldsymbol{\lambda}})^\dagger\textstyle\sum_{b\in\mathcal{A}}q^{\boldsymbol{\lambda}}(b)M_b M_b^\top\big) \\
&= \operatorname{tr}\big(Q(q^{\boldsymbol{\lambda}})^\dagger Q(q^{\boldsymbol{\lambda}})\big) = \operatorname{rk}(Q(q^{\boldsymbol{\lambda}})) \leq \operatorname{rk}(\boldsymbol{M}),
\end{aligned}$$

where the last inequality follows from Lemma 8. At the same time, whenever $q^{\boldsymbol{\lambda}}(b) > 0$, an application of Lemma 7 gives that

$$\big\|M_b^\top Q(q^{\boldsymbol{\lambda}})^\dagger M_b\big\| \leq \big\|M_b^\top\big(q^{\boldsymbol{\lambda}}(b)M_b M_b^\top\big)^\dagger M_b\big\| = \frac{\big\|M_b^\top\big(M_b M_b^\top\big)^\dagger M_b\big\|}{q^{\boldsymbol{\lambda}}(b)} = \frac{\big\|M_b^\dagger M_b\big\|}{q^{\boldsymbol{\lambda}}(b)} \leq \frac{1}{q^{\boldsymbol{\lambda}}(b)},$$

where the last inequality uses that $M_b^\dagger M_b$ is an orthogonal projector. Then,

$$\sum_{b\in\mathcal{A}}q_-^{\boldsymbol{\lambda}}(b)\big\|M_b^\top Q(q^{\boldsymbol{\lambda}})^\dagger M_b\big\| \leq \sum_{b\in\mathcal{A}:\,q_-^{\boldsymbol{\lambda}}(b)>0}\frac{q_-^{\boldsymbol{\lambda}}(b)}{q^{\boldsymbol{\lambda}}(b)} \leq |\{b\in\mathcal{A}:q_-^{\boldsymbol{\lambda}}(b)>0\}| = |\operatorname{supp}(\boldsymbol{\lambda})|.$$

For the second term, setting $\pi$ as the uniform distribution over some $S\subseteq\mathcal{A}$ (assuming $\sum_{s\in S}U^\top M_s M_s^\top U$ is invertible) gives that

$$\max_{b\in\mathcal{A}}\big\|M_b^\top UB(\pi)^{-1}U^\top M_b\big\| = |S|\max_{b\in\mathcal{A}}\big\|M_b^\top U\big(\textstyle\sum_{s\in S}U^\top M_s M_s^\top U\big)^{-1}U^\top M_b\big\|.$$

Hence,

$$\min_{\pi\in\Delta_k}\max_{b\in\mathcal{A}}\big\|M_b^\top UB(\pi)^{-1}U^\top M_b\big\| \leq \min_{S\subseteq\mathcal{A}}|S|\max_{b\in\mathcal{A}}\big\|M_b^\top U\big(\textstyle\sum_{s\in S}U^\top M_s M_s^\top U\big)^{-1}U^\top M_b\big\| =: w^*$$

Note that for any $b\in\mathcal{A}$, the norm on the right-hand side of the inequality is bounded by 1 whenever $b\in S$ (again via an application of Lemma 7 as above). Hence, choosing $S$ as $\mathcal{A}$ yields $k$ as an upper bound on $w^*$. At the same time,

$$\min_{\pi\in\Delta_k}\max_{b\in\mathcal{A}}\big\|M_b^\top UB(\pi)^{-1}U^\top M_b\big\| \leq \min_{\pi\in\Delta_k}\max_{b\in\mathcal{A}}\operatorname{tr}\big(M_b^\top UB(\pi)^{-1}U^\top M_b\big).$$

The right-hand side is a form of the $G$-optimal design criterion (with matrices as elements of the design space), for which one can show that the minimizer coincides with the maximizer of $\log\det\big(\sum_{a\in\mathcal{A}}\pi(a)U^\top M_a M_a^\top U\big)$ and attains a value of $\operatorname{rk}(\boldsymbol{M})$, see Lemma 12.

Let $\beta_{\mathrm{glo}} := \min_{\boldsymbol{\xi} \in \mathcal{W}^{\mathrm{glo}}} \beta_{\boldsymbol{\xi}}$. Combining the four bounds derived above with Proposition 3 (which holds for any $\boldsymbol{\lambda} \in \mathcal{W}_\ell^{\mathrm{loc}}$ and $\boldsymbol{\xi} \in \mathcal{W}^{\mathrm{glo}}$) and Lemma 16 gives that

$$\min_{p \in \Xi_\eta} \Lambda_{\eta,q}(p,\ell) \leq \inf_{\boldsymbol{\lambda} \in \mathcal{W}_\ell^{\mathrm{loc}}} 8L^2 \beta_{\boldsymbol{\lambda}}^2 \min\left\{\mathrm{rk}(\boldsymbol{M}), |\operatorname{supp}(\boldsymbol{\lambda})|\right\} + 2L\left(1 + \beta_{\mathrm{glo}}^2\right) \min\{\mathrm{rk}(\boldsymbol{M}), w^*\},$$

as long as $1/\eta \geq 2L\left(1 + \beta_{\mathrm{glo}}^2\right) \min\{\mathrm{rk}(\boldsymbol{M}), w^*\}$. Taking the maximum over $\ell \in \mathcal{L}$ in this bound yields an upper bound for $\Lambda_{\eta,q}^*$ via Lemma 3, which is also an upper bound for $\Lambda_\eta^*$ as it does not depend on $q$. Finally, note that the image of the map $\ell \mapsto \mathcal{W}_\ell^{\mathrm{loc}}$ is finite as $\mathcal{W}_\ell^{\mathrm{loc}}$ depends on $\ell$ only via the ordering induced by the latter over the actions; hence, the maximum value over $\ell \in \mathcal{L}$ of the bound above is attainable.

$\square$

## E.1 Alternative exploration distributions

We revisit here the key argument in the proof of Proposition 3; that of choosing $\widehat{p}$ to control $\sum_{a \in \mathcal{A}} q(a)\mathcal{E}(a, a^*; \widehat{p})$ while keeping $\langle \widehat{p} - q, H^\top \ell \rangle$ non-positive, having fixed $\ell \in \mathcal{L}$ and $a^* \in \arg\min_{a \in \mathcal{A}} \psi_a^\top \ell$. As argued in that proof, for any choice of $\boldsymbol{\lambda} \in \mathcal{W}_\ell^{\mathrm{loc}}$ (such that $\psi_a - \psi_{a^*} = \boldsymbol{M}\boldsymbol{\lambda}_a$ for all $a \in \mathcal{A}$), choosing $\widehat{p} = \sum_{a \in \mathcal{A}^*} q(a)\nu_a$ with $\nu_a$ supported on $\operatorname{supp}(\boldsymbol{\lambda}, a)$ suffices to ensure that $\langle \widehat{p} - q, H^\top \ell \rangle \leq 0$. On the other hand, establishing an upper bound on $\mathcal{E}(a, a^*; \widehat{p})$ that is proportional to $\sum_{c \in \operatorname{supp}(\boldsymbol{\lambda}, a)} \nu_a(c)\|M_c^\top Q_\delta(\widehat{p})^\dagger M_c\|$ allows using the fact that $\sum_{a \in \mathcal{A}} r(a)\|M_a^\top Q_\delta(r)^\dagger M_a\|$ is at most $2\,\mathrm{rk}(\boldsymbol{M})$ for any $r \in \Delta_k$ (also, at most twice the size of the support of $r$) to bound $\sum_{a \in \mathcal{A}} q(a)\mathcal{E}(a, a^*; \widehat{p})$ as such (see the proof of Theorem 1 above). A slight generalization of an argument in the proof of Proposition 3 gives that for any $\alpha \in \mathbb{R}$ and $a \in \mathcal{A}^*$ (with $\boldsymbol{\lambda}_a \neq \boldsymbol{0}$)

$$
\begin{aligned}
\mathcal{E}(a, a^*; \widehat{p}) &= (\psi_a - \psi_{a^*})^\top Q_\delta(\widehat{p})^\dagger (\psi_a - \psi_{a^*}) \\
&= \left\|\left(Q_\delta(\widehat{p})^\dagger\right)^{1/2}(\psi_a - \psi_{a^*})\right\|^2 \\
&= \left\|\left(Q_\delta(\widehat{p})^\dagger\right)^{1/2}\boldsymbol{M}\boldsymbol{\lambda}_a\right\|^2 \\
&= \left\|\textstyle\sum_{c \in \operatorname{supp}(\boldsymbol{\lambda}, a)}\left(Q_\delta(\widehat{p})^\dagger\right)^{1/2} M_c \boldsymbol{\lambda}_a(c)\right\|^2 \\
&\leq \left(\textstyle\sum_{c \in \operatorname{supp}(\boldsymbol{\lambda}, a)}\left\|\left(Q_\delta(\widehat{p})^\dagger\right)^{1/2} M_c \boldsymbol{\lambda}_a(c)\right\|\right)^2 \\
&\leq \left(\textstyle\sum_{c \in \operatorname{supp}(\boldsymbol{\lambda}, a)}\left\|\left(Q_\delta(\widehat{p})^\dagger\right)^{1/2} M_c\right\|\|\boldsymbol{\lambda}_a(c)\|\right)^2 \\
&\leq \left(\textstyle\sum_{c \in \operatorname{supp}(\boldsymbol{\lambda}, a)}\|\boldsymbol{\lambda}_a(c)\|^{2-2\alpha}\right)\left(\textstyle\sum_{c \in \operatorname{supp}(\boldsymbol{\lambda}, a)}\|\boldsymbol{\lambda}_a(c)\|^{2\alpha}\|M_c^\top Q_\delta(\widehat{p})^\dagger M_c\|\right) \\
&= \left(\textstyle\sum_{c \in \operatorname{supp}(\boldsymbol{\lambda}, a)}\|\boldsymbol{\lambda}_a(c)\|^{2-2\alpha}\right)\left(\textstyle\sum_{c \in \operatorname{supp}(\boldsymbol{\lambda}, a)}\|\boldsymbol{\lambda}_a(c)\|^{2\alpha}\right) \\
&\qquad \cdot \left(\textstyle\sum_{c \in \operatorname{supp}(\boldsymbol{\lambda}, a)} \frac{\|\boldsymbol{\lambda}_a(c)\|^{2\alpha}}{\sum_{b \in \operatorname{supp}(\boldsymbol{\lambda}, a)}\|\boldsymbol{\lambda}_a(b)\|^{2\alpha}}\|M_c^\top Q_\delta(\widehat{p})^\dagger M_c\|\right),
\end{aligned}
$$

where the last inequality is an application of the Cauchy-Schwarz inequality. One can then set $\nu_a(c) \propto \|\boldsymbol{\lambda}_a(c)\|^{2\alpha}$, where $\alpha = 1/2$ corresponds to the choice of $\nu_a$ used in the proof of Proposition 3 and $\alpha = 0$ yields $\nu_a(c) \propto [\![c \in \operatorname{supp}(\boldsymbol{\lambda}, a)]\!]$, which is essentially the analogous choice of $\widehat{p}$ to that used in the analysis of Lattimore and Szepesvári [40] in finite partial monitoring. In any case, the resulting bound is

$$
\begin{aligned}
\sum_{a \in \mathcal{A}} q(a)\mathcal{E}(a, a^*; \widehat{p}) &\leq \sum_{a \in \mathcal{A}} q(a)\left(\textstyle\sum_{c \in \operatorname{supp}(\boldsymbol{\lambda}, a)}\|\boldsymbol{\lambda}_a(c)\|^{2-2\alpha}\right)\left(\textstyle\sum_{c \in \operatorname{supp}(\boldsymbol{\lambda}, a)}\|\boldsymbol{\lambda}_a(c)\|^{2\alpha}\right) \\
&\qquad\qquad\qquad\qquad\qquad \cdot \left(\textstyle\sum_{c \in \mathcal{A}} \nu_a(c)\|M_c^\top Q_\delta(\widehat{p})^\dagger M_c\|\right) \\
&\leq \max_{a \in \mathcal{A}^*}\left(\textstyle\sum_{c \in \operatorname{supp}(\boldsymbol{\lambda}, a)}\|\boldsymbol{\lambda}_a(c)\|^{2-2\alpha}\right)\left(\textstyle\sum_{c \in \operatorname{supp}(\boldsymbol{\lambda}, a)}\|\boldsymbol{\lambda}_a(c)\|^{2\alpha}\right) \\
&\qquad\qquad\qquad\qquad\qquad \cdot \sum_{b \in \mathcal{A}} \widehat{p}(b)\|M_b^\top Q_\delta(\widehat{p})^\dagger M_b\|.
\end{aligned}
$$

Since for any $\alpha \in \mathbb{R}$ the Cauchy-Schwarz inequality gives that

$$\left(\textstyle\sum_{c \in \operatorname{supp}(\boldsymbol{\lambda}, a)}\|\boldsymbol{\lambda}_a(c)\|^{2-2\alpha}\right)\left(\textstyle\sum_{c \in \operatorname{supp}(\boldsymbol{\lambda}, a)}\|\boldsymbol{\lambda}_a(c)\|^{2\alpha}\right) \geq \left(\textstyle\sum_{c \in \operatorname{supp}(\boldsymbol{\lambda}, a)}\|\boldsymbol{\lambda}_a(c)\|\right)^2,$$

choosing $\alpha = 1/2$ minimizes the leading factor in the bound above. To be clear, it is not claimed here that this choice of $\widehat{p}$ minimizes $\sum_{a \in \mathcal{A}} q(a) \mathcal{E}(a, a^*; \widehat{p})$, neither under the structural constraint that $\widehat{p} = \sum_{a \in \mathcal{A}^*} q(a) \nu_a$ with $\nu_a$ supported on $\mathrm{supp}(\boldsymbol{\lambda}, a)$ nor the general constraint that $\langle \widehat{p} - q, H^\top \ell \rangle \le 0$, neither for a fixed $q$ nor after passing to the sup over $q \in \Delta_k^*$. It is not generally clear if such an optimal distribution can be expressed in closed form. The aim, rather, was to show that our choice of $\widehat{p}$ is fairly natural and justified in pursuit of a simple and interpretable bound (one that isolates in a standalone leading factor the 'cost' of using the weights $(\boldsymbol{\lambda}_a)_a$) through our linear-bandit-inspired analysis.

To expand on this discussion, we consider now a special case where a tighter analysis leads to a different value of $\alpha$ being the most convenient. Assume that $\mathrm{col}(M_b)$ and $\mathrm{col}(M_c)$ are orthogonal for any $b \ne c$. In such a case, it holds that

$$
\begin{aligned}
\mathcal{E}(a, a^*; \widehat{p}) &= (\psi_a - \psi_{a^*})^\top Q_\delta(\widehat{p})^\dagger (\psi_a - \psi_{a^*}) \\
&= (\boldsymbol{M}\boldsymbol{\lambda}_a)^\top Q_\delta(\widehat{p})^\dagger (\boldsymbol{M}\boldsymbol{\lambda}_a) \\
&= \big(\textstyle\sum_{c \in \mathrm{supp}(\boldsymbol{\lambda}, a)} M_c \boldsymbol{\lambda}_a(c)\big)^\top Q_\delta(\widehat{p})^\dagger \big(\textstyle\sum_{c' \in \mathrm{supp}(\boldsymbol{\lambda}, a)} M_{c'} \boldsymbol{\lambda}_a(c')\big) \\
&= \textstyle\sum_{c \in \mathrm{supp}(\boldsymbol{\lambda}, a)} (M_c \boldsymbol{\lambda}_a(c))^\top Q_\delta(\widehat{p})^\dagger (M_c \boldsymbol{\lambda}_a(c)) \\
&\le \textstyle\sum_{c \in \mathrm{supp}(\boldsymbol{\lambda}, a)} \|\boldsymbol{\lambda}_a(c)\|^2 \|M_c^\top Q_\delta(\widehat{p})^\dagger M_c\| \\
&= \big(\textstyle\sum_{c' \in \mathrm{supp}(\boldsymbol{\lambda}, a)} \|\boldsymbol{\lambda}_a(c')\|^2\big) \textstyle\sum_{c \in \mathrm{supp}(\boldsymbol{\lambda}, a)} \frac{\|\boldsymbol{\lambda}_a(c)\|^2}{\sum_{b \in \mathrm{supp}(\boldsymbol{\lambda}, a)} \|\boldsymbol{\lambda}_a(b)\|^2} \|M_c^\top Q_\delta(\widehat{p})^\dagger M_c\|
\end{aligned}
$$

where the fourth equality follows from the assumed structure of the observation matrices. This immediately suggests setting $\nu_a(c) \propto \|\boldsymbol{\lambda}_a(c)\|^2$ (or picking $\alpha = 1$), obtaining as result that

$$
\sum_{a \in \mathcal{A}} q(a) \mathcal{E}(a, a^*; \widehat{p}) \le \max_{a \in \mathcal{A}^*} \big(\textstyle\sum_{c \in \mathrm{supp}(\boldsymbol{\lambda}, a)} \|\boldsymbol{\lambda}_a(c)\|^2\big) \sum_{b \in \mathcal{A}} \widehat{p}(b) \|M_b^\top Q_\delta(\widehat{p})^\dagger M_b\|,
$$

which features a better leading factor than what we obtained with $\alpha = 1/2$ in the general case. Again, this choice is not necessarily optimal in any precise sense, but it allows exploiting the structure of this special case in a simple and natural manner. Devising a general choice of $\widehat{p}$ that can recover improved bounds in special cases as such is a worthy direction for refining the bound of Theorem 1.

# F  Proofs of Section 5

The proof of Theorem 2 relies on the following proposition. (The definition of $B(\cdot)$ is given in (8).)

**Proposition 4.** *In globally observable games, it holds that*

$$
\Lambda_\eta^* \le 4 \sqrt{\frac{1}{\eta}} L \min \begin{cases} \sqrt{(1 + \beta_{\mathrm{glo}}^2) \min_{\pi \in \Delta_k} \max_{b \in \mathcal{A}} \|M_b^\top U B(\pi)^{-1} U^\top M_b\|} \\ \sqrt{(1 + \beta_{2,\mathrm{glo}}^2) \min_{\pi \in \Delta_k} \|\boldsymbol{M}^\top U B(\pi)^{-1} U^\top \boldsymbol{M}\|} \end{cases}
$$

*provided that*

$$
1/\eta \ge (1 + L^2) \min \begin{cases} (1 + \beta_{\mathrm{glo}}^2) \min_{\pi \in \Delta_k} \max_{b \in \mathcal{A}} \|M_b^\top U B(\pi)^{-1} U^\top M_b\| \\ (1 + \beta_{2,\mathrm{glo}}^2) \min_{\pi \in \Delta_k} \|\boldsymbol{M}^\top U B(\pi)^{-1} U^\top \boldsymbol{M}\| \end{cases}
$$

*where*

$$
\beta_{2,\mathrm{glo}} := \min_{\boldsymbol{\xi} \in \mathcal{W}^{\mathrm{glo}}} \max_{a,b \in \mathcal{A}} \|\boldsymbol{\xi}_{a,b}\| = \max_{a,b \in \mathcal{A}} \min_{\boldsymbol{v} \in \mathbb{R}^n : \psi_a - \psi_b = \boldsymbol{M}\boldsymbol{v}} \|\boldsymbol{v}\| = \max_{a,b \in \mathcal{A}} \|\boldsymbol{M}^\dagger (\psi_a - \psi_b)\|.
$$

*Proof.* Fix $\ell \in \mathcal{L}$ and $q \in \Delta_k^*$. As in the proof of Proposition 3, we choose a certain distribution $p \in \Xi_\eta$ and bound $\Lambda_{\eta,q}(p, \ell)$. Here, we simply set $p = \gamma \pi + (1 - \gamma) q$, with $\gamma \in (0, 1]$ and $\pi \in \Delta_k$ such that $\mathrm{col}(Q(\pi)) = \mathrm{col}(\boldsymbol{M})$. For brevity, let $\bar{\gamma} = 1 - \gamma$. With these choices we obtain that

$$
\begin{aligned}
\Lambda_{\eta,q}(\gamma \pi + \bar{\gamma} q, \ell) &= \frac{\gamma}{\eta} \langle \pi - q, H^\top \ell \rangle + \frac{\bar{\gamma}}{\eta} \langle q - q, H^\top \ell \rangle + L^2 \sum_{a,b \in \mathcal{A}} q(a) q(b) \mathcal{E}(a, b; \gamma \pi + \bar{\gamma} q) \\
&\le \frac{2\gamma}{\eta} + L^2 \sum_{a,b \in \mathcal{A}} q(a) q(b) \mathcal{E}(a, b; \gamma \pi + \bar{\gamma} q), \quad\quad\quad (19)
\end{aligned}
$$

where the inequality uses the assumption that $\max_{a,b\in\mathcal{A},\ell\in\mathcal{L}}|(\psi_a-\psi_b)^\top\ell|\leq 2$. Fix some $\boldsymbol{\xi}\in\mathcal{W}^{\text{glo}}$, which exists via global observability. Then, similarly to (17) in the proof of Proposition 3, for any $a,b\in\mathcal{A}$,

$$\mathcal{E}(a,b;\gamma\pi+\bar{\gamma}q)\leq\beta_{\boldsymbol{\xi}}^2\max_{c\in\mathcal{A}}\big\|\big(Q_\delta(\gamma\pi+\bar{\gamma}q)^\dagger\big)^{1/2}M_c\big\|^2$$

$$\leq\frac{1}{(1-\delta)\gamma}\beta_{\boldsymbol{\xi}}^2\max_{c\in\mathcal{A}}\big\|M_c^\top Q(\pi)^\dagger M_c\big\|$$

$$\leq\frac{2}{\gamma}\beta_{\boldsymbol{\xi}}^2\max_{c\in\mathcal{A}}\big\|M_c^\top Q(\pi)^\dagger M_c\big\|,$$

where the second step follows from Lemma 7 using the assumption that $\text{col}(Q(\pi))=\text{col}(\boldsymbol{M})$, and the last step uses that $\delta\leq 1/2$. Then, choosing $\boldsymbol{\xi}\in\arg\min_{\boldsymbol{\xi}'\in\mathcal{W}^{\text{glo}}}\beta_{\boldsymbol{\xi}'}$, we get via Lemma 16 that

$$\sum_{a,b\in\mathcal{A}}q(a)q(b)\mathcal{E}(a,b;\gamma\pi+\bar{\gamma}q)\leq\frac{2}{\gamma}\beta_{\text{glo}}^2\max_{b\in\mathcal{A}}\big\|M_b^\top Q(\pi)^\dagger M_b\big\|$$

$$\leq\frac{2}{\gamma}\big(1+\beta_{\text{glo}}^2\big)\max_{b\in\mathcal{A}}\big\|M_b^\top Q(\pi)^\dagger M_b\big\|.$$

Alternatively, for any $\boldsymbol{\xi}\in\mathcal{W}^{\text{glo}}$, we also have that

$$\mathcal{E}(a,b;\gamma\pi+\bar{\gamma}q)=(\psi_a-\psi_b)^\top Q_\delta(\gamma\pi+\bar{\gamma}q)^\dagger(\psi_a-\psi_b)$$

$$=\big\|\big(Q_\delta(\gamma\pi+\bar{\gamma}q)^\dagger\big)^{1/2}(\psi_a-\psi_b)\big\|^2$$

$$=\big\|\big(Q_\delta(\gamma\pi+\bar{\gamma}q)^\dagger\big)^{1/2}\boldsymbol{M}\boldsymbol{\xi}_{a,b}\big\|^2$$

$$\leq\|\boldsymbol{\xi}_{a,b}\|^2\big\|\big(Q_\delta(\gamma\pi+\bar{\gamma}q)^\dagger\big)^{1/2}\boldsymbol{M}\big\|^2$$

$$\leq\frac{2}{\gamma}\|\boldsymbol{\xi}_{a,b}\|^2\big\|\boldsymbol{M}^\top Q(\pi)^\dagger\boldsymbol{M}\big\|,$$

again using the assumption that $\text{col}(Q(\pi))=\text{col}(\boldsymbol{M})$. With $\boldsymbol{\xi}\in\arg\min_{\boldsymbol{\xi}'\in\mathcal{W}^{\text{glo}}}\max_{a,b\in\mathcal{A}}\|\boldsymbol{\xi}_{a,b}'\|$ we obtain that

$$\sum_{a,b\in\mathcal{A}}q(a)q(b)\mathcal{E}(a,b;\gamma\pi+\bar{\gamma}q)\leq\frac{2}{\gamma}\beta_{2,\text{glo}}^2\big\|\boldsymbol{M}^\top Q(\pi)^\dagger\boldsymbol{M}\big\|\leq\frac{2}{\gamma}\big(1+\beta_{2,\text{glo}}^2\big)\big\|\boldsymbol{M}^\top Q(\pi)^\dagger\boldsymbol{M}\big\|,$$

where

$$\beta_{2,\text{glo}}:=\max_{a,b\in\mathcal{A}}\big\|\boldsymbol{M}^\dagger(\psi_a-\psi_b)\big\|=\max_{a,b\in\mathcal{A}}\min_{\boldsymbol{v}\in\mathbb{R}^n\,:\,\psi_a-\psi_b=\boldsymbol{M}\boldsymbol{v}}\|v\|=\min_{\boldsymbol{\xi}\in\mathcal{W}^{\text{glo}}}\max_{a,b\in\mathcal{A}}\|\boldsymbol{\xi}_{a,b}\|.$$

The first equality follows via the minimum norm property of the Moore-Penrose inverse, and the second is due to the structure of $\mathcal{W}^{\text{glo}}$ (see also the proof of Lemma 16 in Appendix E). Let

$$v_1(\pi):=(1+\beta_{\text{glo}}^2)\max_{b\in\mathcal{A}}\big\|M_b^\top U B(\pi)^{-1}U^\top M_b\big\|$$

and

$$v_2(\pi):=\big(1+\beta_{2,\text{glo}}^2\big)\big\|\boldsymbol{M}^\top U B(\pi)^{-1}U^\top\boldsymbol{M}\big\|.$$

Returning back to (19), we have shown that

$$\Lambda_{\eta,q}(\gamma\pi+\bar{\gamma}q,\ell)\leq\frac{2\gamma}{\eta}+\frac{2}{\gamma}L^2\min\{v_1(\pi),v_2(\pi)\}.$$

This expression is minimized at $\gamma=\sqrt{\eta}L\min\{\sqrt{v_1(\pi)},\sqrt{v_2(\pi)}\}$, yielding that

$$\Lambda_{\eta,q}(\gamma\pi+\bar{\gamma}q,\ell)\leq 4\sqrt{\frac{1}{\eta}}L\min\{\sqrt{v_1(\pi)},\sqrt{v_2(\pi)}\}.$$

For $\gamma\leq 1$ to hold, it must hold that

$$1/\eta\geq L^2\min\{v_1(\pi),v_2(\pi)\}.$$

Consider also that

$$z(\gamma\pi + \bar\gamma q) = \max_{a,b\in\mathcal{A}} \mathcal{E}(a,b;\gamma\pi + \bar\gamma q) + \max_{c\in\mathcal{A}}\|M_c^\top Q_\delta(\gamma\pi + \bar\gamma q)^\dagger M_c\|$$

$$\leq \frac{2}{\gamma}\left(1 + \beta_{\mathrm{glo}}^2\right)\max_{b\in\mathcal{A}}\|M_b^\top Q(\pi)^\dagger M_b\|.$$

And that at the same time,

$$z(\gamma\pi + \bar\gamma q) = \max_{a,b\in\mathcal{A}} \mathcal{E}(a,b;\gamma\pi + \bar\gamma q) + \max_{c\in\mathcal{A}}\|M_c^\top Q_\delta(\gamma\pi + \bar\gamma q)^\dagger M_c\|$$

$$\leq \frac{2}{\gamma}\beta_{2,\mathrm{glo}}^2\big\|\boldsymbol{M}^\top Q(\pi)^\dagger \boldsymbol{M}\big\| + \frac{2}{\gamma}\max_{b\in\mathcal{A}}\|M_b^\top Q(\pi)^\dagger M_b\|$$

$$\leq \frac{2}{\gamma}\left(1 + \beta_{2,\mathrm{glo}}^2\right)\big\|\boldsymbol{M}^\top Q(\pi)^\dagger \boldsymbol{M}\big\|,$$

where the second inequality holds since the columns of every $M_b$ are also columns of $\boldsymbol{M}$ via its definition; hence, we can construct a matrix $C_b$ with $\|C_b\| \leq 1$ such that $\boldsymbol{M} = M_b C_b$. We have then shown that,

$$z(\gamma\pi + \bar\gamma q) \leq \frac{2}{\gamma}\min\{v_1(\pi), v_2(\pi)\}$$

$$= \frac{2}{\sqrt{\eta}L}\min\{\sqrt{v_1(\pi)}, \sqrt{v_2(\pi)}\}.$$

Therefore, for $\gamma\pi + \bar\gamma q \in \Xi_\eta$ to hold, it suffices that the last expression is no larger than $2/\eta L$; meaning that $\eta$ must also satisfy that

$$1/\eta \geq \min\{v_1(\pi), v_2(\pi)\}.$$

Thus, if we pick $\pi \in \arg\min_{\pi'\in\Delta_k}\min\{v_1(\pi'), v_2(\pi')\}$,[5] then enforcing that

$$1/\eta \geq (1 + L^2)\min_{\pi\in\Delta_k}\min\{v_1(\pi), v_2(\pi)\} = (1 + L^2)\min\Big\{\min_{\pi\in\Delta_k} v_1(\pi), \min_{\pi\in\Delta_k} v_2(\pi)\Big\}$$

suffices to have

$$\min_{p\in\Xi_\eta}\Lambda_{\eta,q}(p,\ell) \leq 4\sqrt{\frac{1}{\eta}}L\min_{\pi\in\Delta_k}\min\{\sqrt{v_1(\pi)}, \sqrt{v_2(\pi)}\}$$

$$= 4\sqrt{\frac{1}{\eta}}L\min\Big\{\min_{\pi\in\Delta_k}\sqrt{v_1(\pi)}, \min_{\pi\in\Delta_k}\sqrt{v_2(\pi)}\Big\}$$

$$= 4\sqrt{\frac{1}{\eta}}L\min\Big\{\sqrt{\min_{\pi\in\Delta_k} v_1(\pi)}, \sqrt{\min_{\pi\in\Delta_k} v_2(\pi)}\Big\}.$$

Since this bound does not depend on $\ell$ or $q$, it is also an upper bound for $\Lambda_{\eta,q}^*$ (via Lemma 3) and $\Lambda_\eta^*$. $\qquad\square$

**Theorem 2.** *In globally observable games, it holds that*

$$\Lambda_\eta^* \leq 4\sqrt{1/\eta}L\min\Big\{(1 + \beta_{\mathrm{glo}})\sqrt{\min\{\mathrm{rk}(\boldsymbol{M}), w^*\}}, (1 + \beta_{2,\mathrm{glo}})\sqrt{u^*}\Big\}$$

*provided that* $\dfrac{1}{\eta} \geq (1 + L^2)\min\Big\{(1 + \beta_{\mathrm{glo}}^2)\min\{\mathrm{rk}(\boldsymbol{M}), w^*\}, (1 + \beta_{2,\mathrm{glo}}^2)u^*\Big\}$, *where*

$$\beta_{2,\mathrm{glo}} := \max_{a,b\in\mathcal{A}}\big\|\boldsymbol{M}^\dagger(\psi_a - \psi_b)\big\|, \text{ and } u^* := \min_{S\subseteq\mathcal{A}}|S|\big\|\boldsymbol{M}^\top U\big(\textstyle\sum_{s\in S}U^\top M_s M_s^\top U\big)^{-1}U^\top \boldsymbol{M}\big\| \leq k.$$

*Proof.* It follows from the proof of Theorem 1 that

$$\sqrt{\min_{\pi\in\Delta_k}\max_{b\in\mathcal{A}}\big\|M_b^\top U B(\pi)^{-1}U^\top M_b\big\|} \leq \sqrt{\min\{\mathrm{rk}(\boldsymbol{M}), w^*\}}.$$

Similarly, restricting $\pi$ to be uniform over a subset of actions gives that

$$\min_{\pi\in\Delta_k}\big\|\boldsymbol{M}^\top U B(\pi)^{-1}U^\top \boldsymbol{M}\big\| \leq \min_{S\subseteq\mathcal{A}}|S|\big\|\boldsymbol{M}^\top U\big(\textstyle\sum_{s\in S}U^\top M_s M_s^\top U\big)^{-1}U^\top \boldsymbol{M}\big\| =: u^*.$$

This is again bounded by $k$, which is attained with $S = \mathcal{A}$. The theorem then follows from Proposition 4. $\qquad\square$

---

[5]Note that the functions $v_1(\pi)$ and $v_2(\pi)$ do attain their minimum in $\pi$ over $\Delta_k$. A minimizer $\pi^*$ of either function must satisfy our requirement that $\mathrm{col}(Q(\pi^*)) = \mathrm{col}(\boldsymbol{M})$ as this is equivalent to $B(\pi)$ being positive definite, see Lemma 9, which is necessary for $B(\pi)$ to be non-singular.

# G Implementation

Define

$$\Phi_{\eta,q}(p) := \max_{\ell \in \mathcal{L}} \Lambda_{\eta,q}(p,\ell)$$

$$= \frac{1}{\eta} \underbrace{\max_{\ell \in \mathcal{L}} \langle p - q,\, H^\top \ell \rangle}_{=:\sigma_{\mathcal{L}}(H(p-q))} + L^2 \sum_{a,b \in \mathcal{A}} q(a)q(b) \underbrace{(\psi_a - \psi_b)^\top Q_\delta(p)^\dagger (\psi_a - \psi_b)}_{=:\mathcal{E}(a,b;p)},$$

where $\sigma_{\mathcal{L}}$ is the support function of $\mathcal{L}$, which we assumed to be compact. In Algorithm 1, we are asked to solve an optimization problem of the following form at every round:

$$\min_{p \in \Xi_\eta} \Phi_{\eta,q}(p), \qquad \Xi_\eta := \{p \in \Delta_k \colon z(p) \le \tfrac{2}{\eta L}\},$$

where $\eta > 0$ and $q \in \Delta_k$ are given and $z(p) := \max_{a,b \in \mathcal{A}} \mathcal{E}(a,b;p) + \max_{c \in \mathcal{A}} \|M_c^\top Q_\delta(p)^\dagger M_c\|$. This was shown to be a convex problem in the proof of Lemma 3. Going further, we point out here that it can in many interesting cases be represented as a semidefinite program, a well-structured and widely-studied class of convex programs that can be solved in polynomial time via interiorpoint methods [9]. In terms of semidefinite representability, the crucial components to study here are the functions $\mathcal{E}(a,b;\cdot)$, $\|M_c^\top Q_\delta(\cdot)^\dagger M_c\|$, and $\sigma_{\mathcal{L}}(\cdot)$; that the program is semidefinite representable (SDR) would then follow via the combination rules outlined in [9, Chapters 3 and 4], which include taking the intersection of SDR sets, taking the maximum of finitely many SDR functions, and summing (with non-negative weights) SDR functions.

Fix some positive integer $r$ and a $d \times r$ matrix $C$ that satisfies $\operatorname{col}(C) \subseteq \operatorname{col}(M)$, and consider the mapping $p \mapsto \|C^\top Q_\delta(p)^\dagger C\|$ for $p \in \Delta_k$. This mapping, assuming global observability, subsumes the first two functions specified above; namely, $p \mapsto \mathcal{E}(a,b;p)$ and $p \mapsto \|M_c^\top Q_\delta(p)^\dagger M_c\|$ for any $a,b,c \in \mathcal{A}$. We show now that the mapping in question is SDR by showing that the set

$$\Gamma := \left\{(p,G,t) \in \Delta_k \times \mathcal{S}^r \times \mathbb{R} \mid C^\top Q_\delta(p)^\dagger C \preceq G \text{ and } \|G\| \le t\right\}$$

coincides with the solution set of a number of linear matrix inequalities, where $\mathcal{S}^r$ is the set of symmetric $r \times r$ matrices. This is sufficient since the epigraph of the mapping in question coincides with the projection of $\Gamma$ onto the $(p,t)$ space. We firstly note that the semidefinite representability $\Delta_k$ is immediate; it can be characterized by a set of simple linear inequalities. Consider now the following symmetric matrix:

$$J(p,G) := \begin{bmatrix} Q_\delta(p) & C \\ C^\top & G \end{bmatrix}.$$

By the properties of the (generalized) Schur complement (see, e.g., [59, Theorem 1.20]), we have that

$$J(p,G) \succeq 0 \iff Q_\delta(p) \succeq 0,\ \operatorname{col}(C) \subseteq \operatorname{col}(Q_\delta(p)),\ \text{and}\ C^\top Q_\delta(p)^\dagger C \preceq G.$$

For any $p \in \Delta_k$, the first two conditions are always satisfied thanks to the structure of $Q_\delta(p)$ (convex combination of positive semidefinite matrices), the assumption that $\operatorname{col}(C) \subseteq \operatorname{col}(M)$, and the fact that $\operatorname{col}(Q_\delta(p)) = \operatorname{col}(M)$ as asserts Lemma 11. Moreover, it holds that

$$\|G\| \le t \iff G \preceq tI_r \text{ and } G \succeq -tI_r.$$

Hence, for $p \in \Delta_k$ and $G \in \mathcal{S}^r$,

$$(p,G,t) \in \Gamma \iff J(p,G) \succeq 0,\ G \preceq tI_r,\ \text{and}\ G \succeq -tI_r;$$

that is, membership in $\Gamma$ can be characterized by linear matrix inequalities (recalling that $Q_\delta(p)$ is affine in $p$), implying it is SDR.

As for $\sigma_{\mathcal{L}}$, consider now the common scenario when $\mathcal{L} = \mathcal{B}_p(c) := \{\ell \in \mathbb{R}^d \mid \|\ell\|_p \le c\}$ for some $p \ge 1$ and $c > 0$; that is, $\mathcal{L}$ is a centered $L_p$ ball. The support function then becomes $\sigma_{\mathcal{L}}(x) = c\|x\|_q$ with $1/p + 1/q = 1$. For rational $q$, the $L_q$ norm is cone quadratic representable [9, Chapter 3], hence SDR (note that the input to $\sigma_{\mathcal{L}}$ is $H(p-q)$, which is affine in our variable $p$ and thus preserves semidefinite representability). This also holds when $p = 1$ ($q = \infty$) and $p = \infty$ ($q = 1$).

Another common case is when $\mathcal{L}$ is $\mathcal{X}^\circ$, the absolute polar set of $\mathcal{X}$.[6] Here, $\sigma_{\mathcal{L}}(H(p-q))$ is the optimal value of the following program:

$$\max_{\ell \in \mathbb{R}^d} \sum_{a \in \mathcal{A}} (p(a) - q(a))\psi_a^\top \ell \quad \text{s.t.} \quad |\psi_a^\top \ell| \le 1 \ \forall a \in \mathcal{A}.$$

Via strong duality, it is straightforward to show that

$$\sigma_{\mathcal{L}}(H(p-q)) = \min_{\boldsymbol{w} \in \mathbb{R}^k : \ H(p-q)=H\boldsymbol{w}} \|\boldsymbol{w}\|_1 = \min_{\boldsymbol{u}_1, \boldsymbol{u}_2 \in \mathbb{R}^k_{\ge 0} : \ H(p-q)=H(\boldsymbol{u}_1-\boldsymbol{u}_2)} (\boldsymbol{u}_1 + \boldsymbol{u}_2)^\top \mathbf{1}_k.$$

This is the minimum value of a linear function under linear constraints, which can be easily incorporated into the full program adding $\boldsymbol{u}_1$ and $\boldsymbol{u}_2$ as auxiliary variables.

## H Adaptive Learning Rate

---

**Algorithm 2** Adaptive Anchored Exploration-by-Optimization

---

1: **input:** stability parameter $\delta \in (0, 1/2]$, sub-optimality tolerance $\varepsilon \ge 0$, scale parameter $L \ge \omega$, $B > 0$
2: **initialize:** $\eta_1 = \min\{B^{-1}, \sqrt{\log k}\}$, $\widehat{y}_0(a) = 0 \ \forall a \in \mathcal{A}$
3: **for** $t = 1, \ldots, T$ **do**
4:     $\forall a \in \mathcal{A}$, set $q_t(a) \propto [\![a \in \mathcal{A}^*]\!] \exp\big(-\eta_t \sum_{s=1}^{t-1} \widehat{y}_s(a)\big)$
5:     choose $\widetilde{p}_t \in \Delta_k$ such that $\max_{\ell \in \mathcal{L}} \Lambda_{\eta_t, q_t}(\widetilde{p}_t, \ell) \le \Lambda_{\eta_t}^* + \varepsilon$ and $z(\widetilde{p}_t) \le \frac{2}{\eta_t L}$
6:     set $p_t = (1-\delta)\widetilde{p}_t + \delta \mathbf{1}_k/k$
7:     execute $A_t \sim p_t$ and observe $\phi_t = M_t^\top \ell_t$
8:     $\forall a \in \mathcal{A}$, set $\widehat{y}_t(a) = (\psi_a - Hq_t)^\top Q(p_t)^\dagger M_{A_t} \phi_t$
9:     set $V_t = \max\big\{0, \max_{\ell \in \mathcal{L}} \Lambda_{\eta_t, q_t}(\widetilde{p}_t, \ell)\big\}$ and $\eta_{t+1} = \min\Big\{\frac{1}{B}, \sqrt{\frac{\log k}{1+\sum_{s=1}^t V_s}}\Big\}$
10: **end for**

---

Following Lattimore and Szepesvári [40], we briefly describe here an 'adaptive' variant of Algorithm 1 that tunes the learning rate in an online fashion, sidestepping (for the most part) the need to have prior knowledge concerning the value of $\Lambda_\eta^*$. This variant is described in Algorithm 2, with the main difference being the use of a learning rate schedule $(\eta_t)_t$, where $\eta_t$ is set based on the attained values at the optimization problems of past rounds. Note that the algorithm is not completely parameter-free, besides the stability ($\delta$) and tolerance ($\varepsilon$) parameters (which can in principle be reduced at will), it requires a sufficiently large constant $B > 0$ as input to guarantee that the learning rates are sufficiently small to meet the requirements of Theorems 1 and 2. However, $B$ can be picked conservatively as it only affects the regret through an additive term as shown below.

Adapting our analysis along the same lines as the proof of Theorem 6 in [40], one can show (analogously to Proposition 2) that Algorithm 2 satisfies (assuming global observability)

$$R_T \le 2\delta T + \frac{\log k}{\eta_T} + \mathbb{E}\left[\sum_{t=1}^T \eta_t V_t\right]$$

$$\le 2\delta T + 5\mathbb{E}\left[\sqrt{\left(1 + \sum_{t=1}^T V_t\right)\log k}\right] + \mathbb{E}\left[\max_{t \in [T]} V_t\right]\sqrt{\log k} + B\log k.$$

**Locally observable games.** Via Theorem 1, there exist constants $\alpha, B'$ such that $\Lambda_\eta^* \le \alpha$ given that $\eta \le 1/B'$. Hence, as long as $B \ge B'$, we have that $V_t \le \Lambda_{\eta_t}^* + \varepsilon \le \alpha + \varepsilon$. Thus,

$$R_T \le 2\delta T + 5\sqrt{(1 + (\alpha + \varepsilon)T)\log k} + (\alpha + \varepsilon)\sqrt{\log k} + B\log k,$$

where, assuming $\delta$ and $\varepsilon$ are picked sufficiently small (e.g., $o(T^{-1/2})$), the dominant term is of order $\sqrt{\alpha T \log k}$.

---

[6]Note that for $\mathcal{X}^\circ$ to be compact, $\mathcal{X}$ must span $\mathbb{R}^d$.

**Globally observable games.** Theorem 2 asserts that there exist constants $\alpha, B'$ such that $\Lambda_\eta^* \leq \alpha/\sqrt{\eta}$ given that $\eta \leq 1/B'$. Hence, assuming that $B \geq \max\{1, B'\}$, we get that

$$V_t \leq \Lambda_{\eta_t}^* + \varepsilon \leq \alpha/\sqrt{\eta_t} + \varepsilon \leq \alpha/\sqrt{\eta_t} + \varepsilon/\sqrt{\eta_t}.$$

Then, from the proof of Proposition 7 in [40] we obtain that

$$R_T \leq 2\delta T + 5 \left( \frac{\alpha + \varepsilon}{(\log k)^{1/4}} T + \max\{1, B^2 \log k\}^{3/4} \right)^{2/3} \sqrt{\log k}$$

$$+ \alpha \left( \frac{\alpha + \varepsilon}{(\log k)^{1/4}} T + \max\{1, B^2 \log k\}^{3/4} \right)^{1/3} (\log k)^{1/4} + \varepsilon + B \log k,$$

where, assuming $\delta$ and $\varepsilon$ are picked sufficiently small (e.g., $o(T^{-1/3})$), the dominant term is of order $(\alpha T)^{2/3} (\log k)^{1/3}$.

As a final remark, note that $\varepsilon$ is our error tolerance when minimizing $\max_{\ell \in \mathcal{L}} \Lambda_{\eta_t, q_t}(p, \ell)$ in $p \in \Xi_{\eta_t}$, where the optimal value is $\Lambda_{\eta_t, q_t}^*$. The properties of this optimization program are discussed in Appendix G. On the other hand, neither algorithm undertakes the (generally) more challenging task of computing $\Lambda_{\eta_t}^*$ up to some tolerance. Still, when tuning a fixed learning rate, this task is somewhat unavoidable if tight regret guarantees are sought, and therein lies the utility of the online learning rate schedule described here.

# I Examples - Continued

## I.1 Linear bandits

Since $\psi_a = M_a$ for all $a \in \mathcal{A}$ and $\mathcal{L} = \mathcal{X}^\circ$, $\max_{a \in \mathcal{A}, \ell \in \mathcal{L}} |M_a^\top \ell| \leq 1$. Hence, $\omega <= 1$ (see its definition in Lemma 2) and $L$ can be chosen as 1. Moreover, trivially, $\psi_a - \psi_b = M_a - M_b = \boldsymbol{M}(e_a - e_b)$, which shows that the game is locally observable. Finally, it is immediate that for any $\ell \in \mathcal{L}$ and $a^* \in \arg\min_{a \in \mathcal{A}} \psi_a^\top \ell$, $(e_a - e_{a^*})_{a \in \mathcal{A}^*} \in \mathcal{W}_\ell^{\text{loc}}$, and consequently, $\beta_{\text{loc}} := \max_{\ell \in \mathcal{L}} \min_{\boldsymbol{\lambda} \in \mathcal{W}_\ell^{\text{loc}}} \max_{a \in \mathcal{A}^*} \|\boldsymbol{\lambda}_a\|_1 \leq 2$. We then obtain via Theorem 1 (and Lemma 17) that $\Lambda_\eta^* \lesssim d$ (i.e., $\Lambda_\eta^*$ is no larger than $d$ up to a universal constant).

## I.2 Linear dueling bandits

Recall that here, the action set has the form $\mathcal{A} = [m] \times [m]$ for some positive integer $m$. With a fixed feature mapping $a \mapsto \psi_a$ from $[m]$ to $\mathbb{R}^d$, we have that for $(a, b) \in \mathcal{A}$, $\psi_{a,b} := \psi_a + \psi_b$ and $M_{a,b} = \psi_a - \psi_b$. If $\mathcal{L} = ((\psi_a)_{a \in \mathcal{A}})^\circ$, then $\omega \leq 2$ and $L$ can be taken as 2. As pointed out in [32], the Pareto optimal actions are those of the form $(a, a)$ for $a \in [m]$. Fix $\ell \in \mathcal{L}$, and let $(a, b) \in \arg\min_{(a', b') \in \mathcal{A}} \langle \psi_{a', b'}, \ell \rangle$. (Here, $a$ and $b$ are not necessarily distinct and, in any case, satisfy $\psi_a = \psi_b = \min_{c \in [m]} \langle \psi_c, \ell \rangle$.) Now, for any $(c, c) \in \mathcal{A}$, we can write

$$\psi_{c,c} - \psi_{a,b} = \psi_c + \psi_c - \psi_a - \psi_b = M_{c,a} + M_{c,b}.$$

At the same time, $\langle \psi_{c,a} - \psi_{c,c}, \ell \rangle = \langle \psi_a - \psi_c, \ell \rangle \leq 0$ and $\langle \psi_{c,b} - \psi_{c,c}, \ell \rangle = \langle \psi_b - \psi_c, \ell \rangle \leq 0$. This shows that the game is locally observable (see Lemma 4) and that $\beta_{\text{loc}} \leq 2$ as in the linear bandit case. Hence, it also holds here that $\Lambda_\eta^* \lesssim d$.

## I.3 Learning with graph feedback

The game here, to recall, is characterized by an undirected graph $\mathcal{G} = (\mathcal{A}, E)$ and we have that $d = k$, $\psi_a = e_a$, and $M_a \in \mathbb{R}^{k \times |N_\mathcal{G}(a)|}$ has the vectors $(e_b)_{b \in N_\mathcal{G}(a)}$ as columns, where $N_\mathcal{G}(a)$ is the neighborhood of $a$ in $\mathcal{G}$ (which need not include $a$). In the terminology of [1], a graph is said to be weakly observable if $\forall a \in \mathcal{A}, \exists b \in \mathcal{A} : a \in N_\mathcal{G}(b)$, while in a strongly observable graph, it holds that $\forall a \in \mathcal{A}$, either $a \in N_\mathcal{G}(a)$, $a \in N_\mathcal{G}(b) \; \forall b \in \mathcal{A}$, or both. Here, and in all the examples to follow in this section, all actions are Pareto optimal. We consider $\mathcal{L}$ in this example to be $\mathcal{B}_\infty(1)$, which coincides with $((e_a)_{a \in [d]})^\circ$. Note that $\max_{a \in \mathcal{A}, \ell \in \mathcal{L}} \|M_a^\top \ell\| = \max_{a \in \mathcal{A}} \sqrt{|N_\mathcal{G}(a)|}$, which can be as large as $\sqrt{k}$, so we derive in the following a tighter bound on $\omega$.

Recall from Lemma 2 that

$$\omega := \sup_{a,b,c\in\mathcal{A},\, p\in\Delta_k,\, \ell\in\mathcal{L}} \frac{|(\psi_b - \psi_c)^\top Q_\delta(p)^\dagger M_a M_a^\top \ell|}{\left\|M_a^\top Q_\delta(p)^\dagger(\psi_b - \psi_c)\right\|}\,.$$

Now fix $a, b, c \in \mathcal{A}$, $p \in \Delta_k$, and $\ell \in \mathcal{L}$. Firstly, the Cauchy-Schwarz inequality gives that

$$|(\psi_b - \psi_c)^\top Q_\delta(p)^\dagger M_a M_a^\top \ell| \leq \left\|M_a^\top Q_\delta(p)^\dagger(\psi_b - \psi_c)\right\|_1 \|M_a^\top \ell\|_\infty\,.$$

Notice now that $\|M_a^\top \ell\|_\infty \leq \|\ell\|_\infty \leq 1$. By the structure of the observation matrices, $Q_\delta(p)$ is a diagonal matrix with $(P(a))_{a\in\mathcal{A}}$ as the diagonal entries, where

$$P(a) := \delta + (1-\delta) \sum_{b\in N_\mathcal{G}(a)} p(b)\,.$$

Hence, $Q_\delta(p)^\dagger(\psi_b - \psi_c) = P(b)^{-1}e_b - P(c)^{-1}e_c$, and consequently, $M_a^\top Q_\delta(p)^\dagger(\psi_b - \psi_c)$ has at most two possibly non-zero entries:

$$\alpha := P(b)^{-1}[\![b \in N_\mathcal{G}(a)]\!] \qquad \text{and} \qquad \beta := -P(c)^{-1}[\![c \in N_\mathcal{G}(a)]\!]\,.$$

Then, using that $|\alpha| + |\beta| = \sqrt{(|\alpha| + |\beta|)^2} \leq \sqrt{2}\sqrt{\alpha^2 + \beta^2}$, we get that

$$\left\|M_a^\top Q_\delta(p)^\dagger(\psi_b - \psi_c)\right\|_1 \leq \sqrt{2}\left\|M_a^\top Q_\delta(p)^\dagger(\psi_b - \psi_c)\right\|\,,$$

which implies via the arguments above that

$$|(\psi_b - \psi_c)^\top Q_\delta(p)^\dagger M_a M_a^\top \ell| \leq \sqrt{2}\left\|M_a^\top Q_\delta(p)^\dagger(\psi_b - \psi_c)\right\|\,,$$

and consequently, $\omega \leq \sqrt{2}$. Alternatively, if $\mathcal{L} = [0,1]^k$, the non-negativity of the losses and the structure of $M_a^\top Q_\delta(p)^\dagger(\psi_b - \psi_c)$ described above yield that

$$\begin{aligned}
|(\psi_b - \psi_c)^\top Q_\delta(p)^\dagger M_a M_a^\top \ell| &\leq \max\{|\alpha|, |\beta|\} \\
&= \left\|M_a^\top Q_\delta(p)^\dagger(\psi_b - \psi_c)\right\|_\infty \leq \left\|M_a^\top Q_\delta(p)^\dagger(\psi_b - \psi_c)\right\|\,.
\end{aligned}$$

Hence, it simply holds in this variant that $\omega \leq 1$.

Next, we study the problem-dependent terms appearing in the bounds of Theorem 1 and Theorem 2 in weakly and strongly observable observable graphs respectively, leading to a bound of order $\alpha(\mathcal{G})$ on $\Lambda_\eta^*$ in the former and a bound of order $\sqrt{\delta(\mathcal{G})/\eta}$ in the latter.

**Strongly observable graphs.**  Fix $\ell \in \mathcal{L}$, as mentioned in Section 4, we aim to find $\boldsymbol{\lambda} \in \mathcal{W}_\ell^{\mathrm{loc}}$ that is (essentially) supported on an independent set and satisfying $\beta_{\boldsymbol{\lambda}} \leq 2$. Let $a^* \in \arg\min_{a\in\mathcal{A}}\langle\psi_a, \ell\rangle$. Notice here that $\mathcal{A}^* = \mathcal{A}$ and $\langle\psi_a, \ell\rangle = \ell(a)$. We construct such a $\boldsymbol{\lambda}$ by iteratively constructing two functions $g_1, g_2 \colon \mathcal{A} \setminus \{a^*\} \to \mathcal{A}$ such that for all $a \in \mathcal{A} \setminus \{a^*\}$,

i) $a \in N_\mathcal{G}(g_1(a))$
ii) $a^* \in N_\mathcal{G}(g_2(a))$
iii) $\ell(g_1(a)) \leq \ell(a)$
iv) $\ell(g_2(a)) \leq \ell(a)$
v) $|\operatorname{img}(g_1) \cup \operatorname{img}(g_2)| \leq \alpha(\mathcal{G}) + 1\,.$

Fix $a \in \mathcal{A} \setminus \{a^*\}$. Note that for $\boldsymbol{\lambda} \in \mathcal{W}_\ell^{\mathrm{loc}}$ and $b \in \mathcal{A}$, $\boldsymbol{\lambda}_a(b) \in \mathbb{R}^{|N_\mathcal{G}(b)|}$ with each coordinate corresponding to a neighbor of $b$. Since, by assumption, $a \in N_\mathcal{G}(g_1(a))$, we set $\boldsymbol{\lambda}_a(g_1(a)) = e_a$ where we abuse notation and use $e_a$ in this expression as the indicator vector in $\mathbb{R}^{|N_\mathcal{G}(b)|}$ for the coordinate corresponding to $a$. Similarly, set $\boldsymbol{\lambda}_a(g_2(a)) = -e_{a^*}$ using that $a^* \in N_\mathcal{G}(g_2(a))$. While for $b \notin \{g_1(a), g_2(a)\}$, set $\boldsymbol{\lambda}_a(b) = \mathbf{0}$. Finally, we naturally set $\boldsymbol{\lambda}_{a^*}(b) = \mathbf{0}$ for all $b \in \mathcal{A}$. Using the assumed properties above, we get that indeed $\psi_a - \psi_{a^*} = M\boldsymbol{\lambda}_a$, $\boldsymbol{\lambda} \in \mathcal{W}_\ell^{\mathrm{loc}}$ (since $\ell(g_1(a)), \ell(g_2(a)) \leq \ell(a)$), $\beta_{\boldsymbol{\lambda}} \leq 2$, and $\operatorname{supp}(\boldsymbol{\lambda}) \leq \alpha(\mathcal{G}) + 1$, as desired.

The two functions can be constructed using the iterative node elimination procedure detailed in Algorithm 3. This construction satisfies the required properties for $g_1$ and $g_2$ as listed above. In particular, $\operatorname{img}(g_1)$ is an independent set, and $|\operatorname{img}(g_2) \setminus \operatorname{img}(g_1)| \leq 1$. The arguments laid so far

illustrate that the game is locally observable (see Lemma 4), and, moreover, that the first addend in the bound of Theorem 1 is of order $\alpha(\mathcal{G})$. What remains now is bounding $w^*$ (note that $\beta_{\mathrm{glo}} \leq 2$ using Lemma 17 and the fact that $\beta_{\mathrm{loc}} \leq 2$). Let $S^* = \mathrm{img}(g_1) \cup \mathrm{img}(g_2)$, which is a total dominating set (that is, every node in the graph, including the ones in $S^*$, is connected to a node in $S^*$). Then,

$$w^* := \min_{S \subseteq \mathcal{A}} |S| \max_{b \in \mathcal{A}} \left\| M_b^\top \left( \textstyle\sum_{s \in S} M_s M_s^\top \right)^{-1} M_b \right\|$$

$$\leq |S^*| \max_{b \in \mathcal{A}} \left\| M_b^\top \left( \textstyle\sum_{s \in S^*} M_s M_s^\top \right)^{-1} M_b \right\| \leq |S^*| \max_{b \in \mathcal{A}} \left\| M_b^\top M_b \right\| \leq |S^*| \leq \alpha(\mathcal{G}) + 1 \,,$$

where we have used that $\sum_{s \in S^*} M_s M_s^\top$ is a diagonal matrix where every entry on the diagonal is lower bounded by 1 (thanks to $S^*$ being a total dominating set).

---

**Algorithm 3** Constructing $g_1$ and $g_2$

> **input:** undirected graph $\mathcal{G} = (\mathcal{A}, E)$, $\ell \in \mathbb{R}^d$, $a^* \in \arg\min_{a \in \mathcal{A}} \ell(a)$
> **output:** $g_1, g_2 \colon \mathcal{A} \setminus \{a^*\} \to \mathcal{A}$
> **initialize:** $\mathcal{G}' \leftarrow \mathcal{G}$
> **while** $\mathcal{G}'$ contains at least one vertex **do**
>      **if** $V_{\mathcal{G}'} = \mathcal{A}$ **then** $b \leftarrow a^*$ **else** select $b \in \arg\min_{a \in V_{\mathcal{G}'}} \ell(a)$
>                                                 ▷ $V_{\mathcal{G}'}$ denotes the set of all vertices in $\mathcal{G}'$
>      set $g_1(a) = b$ for all $a \in N_{\mathcal{G}'}(b) \setminus \{a^*\}$
>    ▷ note that $b \in N_{\mathcal{G}'}(b)$ if $b \neq a^*$, otherwise $b \in N_{\mathcal{G}}(a^*)$ implying it has already been eliminated
>      $\mathcal{G}' \leftarrow \mathcal{G}'[V_{\mathcal{G}'} \setminus (N_{\mathcal{G}'}(b) \cup \{b\})]$      ▷ for $S \subseteq V_{\mathcal{G}'}$, $\mathcal{G}'[S]$ denotes the sub-graph induced by $S$
> **end while**
> **if** $a^* \in N_{\mathcal{G}}(a^*)$ **then**
>      set $g_2(a) = a^*$ for all $a \in \mathcal{A} \setminus a^*$
> **else**                            ▷ in this case, $N_{\mathcal{G}}(a^*) = \mathcal{A} \setminus \{a^*\}$ and $g_1(a) = a^*$ for all $a \in \mathcal{A} \setminus \{a^*\}$
>      select $c \in \arg\min_{a \in \mathcal{A} \setminus a^*} \ell(a)$
>      set $g_2(a) = c$ for all $a \in \mathcal{A} \setminus a^*$
> **end if**
> **return** $g_1$ and $g_2$

---

**Weakly observable graphs.** It is immediate to verify that weakly observable graphs are globally observable with $\beta_{\mathrm{glo}} \leq 2$. Moreover, $w^*$ is easily seen to be bounded by the total domination number $\delta(\mathcal{G})$ (size of a minimal total dominating set) by following the same argument laid above in the strongly observable case.

## I.4 Full information

This is a special case of strongly observable graphs, with the graph being a clique (fully connected graph) with self loops. For this graph, $\alpha(\mathcal{G}) = 1$; hence, the arguments laid out in the previous example yield that $\Lambda_\eta^*$ is simply upper bounded by a universal constant (independent of the number of actions). In fact, since $M_a$ is the identity matrix for all $a$, it is easily seen that $w^* = 1$. Moreover, the construction we described for strongly observable graphs (Algorithm 3) trivially yields, for every $\ell \in \mathcal{L}$, weights $\boldsymbol{\lambda} \in \mathcal{W}_\ell^{\mathrm{loc}}$ that satisfy $\mathrm{supp}(\boldsymbol{\lambda}) = \{a^*\}$ (with $a^* \in \arg\min_{a \in \mathcal{A}} \psi_a^\top \ell$) and $\beta_{\boldsymbol{\lambda}} = \max_{a \in \mathcal{A}} \|\boldsymbol{\lambda}_a(a^*)\| = \max_{a \in \mathcal{A}} \|e_a - e_{a^*}\| = \sqrt{2}$.

## I.5 Bandits with ill-conditioned observers

In this game: $d = k$, $\psi_a = e_a$, and $M_a = (1 - \varepsilon)\mathbf{1}_k / k + \varepsilon e_a$ for some $\varepsilon \in (0, 1]$. Moreover, we take $\mathcal{L} = \mathcal{B}_\infty(1)$. Since $M_a \in \Delta_k$ for all $a$, we have that $\|M_a\|_1 = 1$. Hence, the Cauchy-Schwarz inequality gives that $\max_{a \in \mathcal{A}, \ell \in \mathcal{L}} |M_a^\top \ell| \leq 1$, implying that $\omega \leq 1$. As noted in Section 4, $e_a - e_b = 1/\varepsilon(M_a - M_b)$ for any $a, b \in \mathcal{A}$. Hence, it holds for any $\ell \in \mathcal{L}$ and $a^* \in \arg\min_{a \in \mathcal{A}} \psi_a^\top \ell$ that $(e_a/\varepsilon - e_{a^*}/\varepsilon)_{a \in \mathcal{A}} \in \mathcal{W}_\ell^{\mathrm{loc}}$, and consequently, that $\beta_{\mathrm{loc}} \leq \max_{a, b \in \mathcal{A}} \|e_a/\varepsilon - e_b/\varepsilon\|_1 = 2/\varepsilon$. Theorem 1 then gives that $\Lambda_\eta^* \lesssim d/\varepsilon^2$.

## I.6 Learning with composite graph feedback

This game is a variant of the graph feedback problem (on an undirected graph $G = (\mathcal{A}, E)$ *with all self-loops*) where, again, $d = k$ and $\psi_a = e_a$; however, the feedback vector $M_a$ is the $a$-th row of the $k \times k$ matrix $D^{-1}A$; hence $\boldsymbol{M} = (D^{-1}A)^\top = AD^{-1}$. Here, $D$ is the degree matrix and $A$ is the adjacency matrix. Hence, after playing $A_t$, the learner observes the average of the losses $\{\ell_t(i): i \in N_\mathcal{G}(A_t)\}$, which include $\ell_t(A_t)$. As mentioned in Section 2, this feedback is less informative than standard graph feedback, and draws inspiration from problems studied in [57, 25] motivated by applications in signal processing (SNETs [19]). Once again we take $\mathcal{L} = \mathcal{B}_\infty(1)$. Note that $\|M_a\|_1 = 1$; hence, $\max_{a \in \mathcal{A}, \ell \in \mathcal{L}} |M_a^\top \ell| \leq 1$, implying again that $\omega \leq 1$. Depending on the structure of the graph, the game could be hopeless, globally or locally observable (or even trivial if there is just one action).

If the graph is empty of edges except self-loops (i.e., $\boldsymbol{M} = \boldsymbol{I}_k$), we recover the standard bandit game, which is locally observable. In fact, this is a necessary condition for locally observability. To see this, fix $\ell \in \mathcal{L}$ such that $\ell(a) < \ell(b) < \min_{c \in \mathcal{A} \setminus \{a,b\}} \ell(c)$. Then, $e_b - e_a \notin \operatorname{span}\{M_a, M_b\}$ if $a$ and $b$ are connected in the graph, which rules out local observability via Lemma 4 (note that all actions are Pareto optimal). As for global observability, one needs that $e_b - e_a \in \operatorname{span}\{M_c: c \in \mathcal{A}\}$ for all $a, b \in \mathcal{A}$. If we define $\mathcal{S} := \operatorname{span}\{e_a - e_b: \text{ for all } a, b \in \mathcal{A}\}$, global observability then translates to $\mathcal{S} \subseteq \operatorname{col}(\boldsymbol{M})$. Now, suppose that $\det(\boldsymbol{M}) = 0$. This would imply that $\dim(\operatorname{col}(\boldsymbol{M}))$ is at most $k - 1$. Then, since $\dim(\mathcal{S}) = k - 1$, global observability would require that $\dim(\operatorname{col}(\boldsymbol{M})) = k - 1$ and $\mathcal{S} = \operatorname{col}(\boldsymbol{M})$. This can never hold, however, because the vector $\mathbf{1}_k$ is orthogonal to $\mathcal{S}$, while any column of $\boldsymbol{M}$ has a non-zero component along $\mathbf{1}_k$, implying hence that $\operatorname{col}(\boldsymbol{M}) \not\subset \mathcal{S}$. Therefore, in order to be globally observable, the game must satisfy that $\det(\boldsymbol{M}) \neq 0$, which is also clearly a sufficient condition.

Assume in what follows that the game is globally observable, meaning that $\boldsymbol{M}$ has full rank. As mentioned in Section 5, it holds here that $u^* = k$. To see this, note that Lemma 9 implies in this case that for any orthonormal basis $U$, $\sum_{s \in S} U^\top M_s M_s^\top U$ is invertible only when $S = \mathcal{A}$. Hence,

$$u^* = k\big\|\boldsymbol{M}^\top U \big(U^\top \boldsymbol{M} \boldsymbol{M}^\top U\big)^{-1} U^\top \boldsymbol{M}\big\| = k\,.$$

As for $w^*$, we know from Theorem 1 that $w^* \leq k$. Moreover, recalling that $(M_a)_{a \in \mathcal{A}}$ are vectors in this example, we have that

$$w^* \geq \min_{\pi \in \Delta_k} \max_{b \in \mathcal{A}} M_b^\top U B(\pi)^{-1} U^\top M_b = k\,,$$

where the equality follows from Lemma 12 using that $\operatorname{rk}(\boldsymbol{M}) = k$ by assumption. Hence, it holds too that $w^* = k$. Theorem 2 then yields a bound of order $\beta_{2,\text{glo}} \sqrt{k/\eta}$ on $\Lambda_\eta^*$.

We now examine the two quantities $\beta_{\text{glo}}$ and $\beta_{2,\text{glo}}$ in this problem. The assumed Invertibility of $\boldsymbol{M}$ implies that

$$\beta_{\text{glo}} = \max_{a,b \in \mathcal{A}} \big\|\boldsymbol{M}^{-1}(e_a - e_b)\big\|_1 \qquad \text{and} \qquad \beta_{2,\text{glo}} = \max_{a,b \in \mathcal{A}} \big\|\boldsymbol{M}^{-1}(e_a - e_b)\big\|_2\,.$$

It holds here, as it does in general, that $\beta_{2,\text{glo}} \leq \beta_{\text{glo}} \leq \sqrt{k}\beta_{2,\text{glo}}$. Recall that $\boldsymbol{M} = AD^{-1}$ and let $S := D^{-1/2}AD^{-1/2}$. Note that $S$ is symmetric and $\boldsymbol{M}$ could be rewritten in terms of $S$ as $\boldsymbol{M} = D^{1/2}SD^{-1/2}$. Further, $\boldsymbol{M}$ and $S$ are similar, thus they have the same eigenvalues. For a square matrix $V$, let $\sigma(V)$ denote its smallest singular value, and let $\{w_i(V)\}_{i=1,\dots,k}$ denote its eigenvalues. Lastly, let $\deg(a)$ denote the degree of action $a$, and let $\deg_{\min}$ and $\deg_{\max}$ denote, respectively, the smallest and the largest degree of any action in the graph. Now,

$$\beta_{2,\text{glo}} = \max_{a,b \in \mathcal{A}} \big\|\boldsymbol{M}^{-1}(e_a - e_b)\big\|_2 = \max_{a,b \in \mathcal{A}} \|D^{1/2}S^{-1}D^{-1/2}(e_a - e_b)\|_2$$

$$\leq \sqrt{\frac{\deg_{\max}}{\deg(a)} + \frac{\deg_{\max}}{\deg(b)}} \|S^{-1}\|_2 \leq \frac{\sqrt{2\deg_{\max}/\deg_{\min}}}{\sigma(S)}$$

$$= \frac{\sqrt{2\deg_{\max}/\deg_{\min}}}{\min_{i=1,\dots,k} |w_i(S)|} = \frac{\sqrt{2\deg_{\max}/\deg_{\min}}}{\min_{i=1,\dots,k} |w_i(\boldsymbol{M})|}\,,$$

where the penultimate step follows from the symmetry of $S$, and the last step uses that $\boldsymbol{M}$ and $S$ are similar. Note also that $w_i(S) = 1 - w_i(L^{\text{sym}})$ where $L^{\text{sym}} = \boldsymbol{I}_k - S$ is the symmetrically

normalized Laplacian of the graph $\mathcal{G}$. Hence, it also holds that

$$\beta_{2,\text{glo}} \leq \frac{\sqrt{2\deg_{\max}/\deg_{\min}}}{\min_{i=1,\dots,k}|1 - w_i(L^{\text{sym}})|} .$$

### I.6.1  Cycle graphs

As an example, take $\mathcal{G}$ to be a $k$-nodes cycle graph (with self-loops), where $k \geq 3$. For convenience, we index actions starting from $0$ instead of $1$ in the following discussion; that is we take $\mathcal{A} = \{0, \dots, k-1\}$. We also extend this notation to vectors and matrices indexed by the actions. For this family of graphs, we have that

$$A_{a,b} = [\![ b \in \{(a-1) \bmod k, \ a, \ (a+1) \bmod k\}]\!] \qquad \text{and} \qquad M = \frac{1}{3}A .$$

If $k \bmod 3 = 0$, then $M$ is singular, which is synonymous with a hopeless (i.e., not globally observable) game in this problem as argued before. A simple non-zero vector $v$ that belongs to $\ker(M)$ in this case is given by

$$v(a) = \begin{cases} 2, & \text{if } a \bmod 3 = 0 \\ -1, & \text{otherwise} \end{cases}$$

for $a \in \mathcal{A}$, see Graph (i) in Figure 1 for a schematic representation on a 9-node cycle. When $k \bmod 3 \in \{1, 2\}$, $M$ is invertible and the game is globally observable. In particular, for $a, b \in \mathcal{A}$, one can verify that

$$M_{a,b}^{-1} = \frac{2}{\sqrt{3}} \sin\left(\frac{2\pi k}{3}\right) \cos\left(\frac{2\pi h(a,b)}{3}\right) + \sqrt{3}\sin\left(\frac{2\pi h(a,b)}{3}\right) ,$$

where $h(a,b) = (a-b) \bmod k$. We now characterize $\beta_{\text{glo}}$ and $\beta_{2,\text{glo}}$ up to small constants, focusing on the case when $k \bmod 3 = 1$. (The remaining case, when $k \bmod 3 = 2$, can be treated in a similar manner.) Firstly, via the triangle's inequality,

$$\beta_{2,\text{glo}} = \max_{a,b\in\mathcal{A}}\big\|M^{-1}(e_a - e_b)\big\|_2 \leq 2\max_{a\in\mathcal{A}}\big\|M^{-1}e_a\big\|_2 .$$

And similarly, $\beta_{\text{glo}} \leq 2\max_{a\in\mathcal{A}}\big\|M^{-1}e_a\big\|_1$. As the entries of $M^{-1}$ are determined by $h(a,b)$, all its columns are identical up to cyclical shifts. Hence, the bound above is attained by any action. From the formula stated above for $M^{-1}$, one can check that each of its columns contains $1 + 2(k-1)/3$ entries with value 1 and $(k-1)/3$ entries with value $-2$, Graph (ii) in Figure 1 provides a schematic representation of one column of $M^{-1}$ on a 10-node cycle. Evaluating the $L_1$ and $L_2$ norms we obtain

$$\beta_{2,\text{glo}} \leq 2\sqrt{2k-1} \qquad \text{and} \qquad \beta_{\text{glo}} \leq 2 + 8(k-1)/3 .$$

On the other hand, we can obtain a lower bound by taking $a$ and $b$ as neighbors. In which case, $M^{-1}(e_a - e_b)$ contains $(k-1)/3$ entries with value 3, $(k-1)/3$ entries with value $-3$, and $1 + (k-1)/3$ entries with value 0, a visual representation is provided by Graph (iii) in Figure 1. Then, again evaluating the $L_1$ and $L_2$ norms gives that

$$\beta_{2,\text{glo}} \geq \sqrt{6(k-1)} \qquad \text{and} \qquad \beta_{\text{glo}} \geq 2(k-1) .$$

This shows that $\beta_{\text{glo}}$ is $\Theta(k)$, while $\beta_{2,\text{glo}}$ is $\Theta(\sqrt{k})$, providing thus an appreciable improvement relative to the former in the bound of Theorem 2 (recalling that here, $w^* = u^* = k$).

### I.6.2  Complete bipartite graphs

As a second example, we consider a complete balanced bipartite graph $G_{k/2,k/2}$ (with self loops), assuming $k$ is even. The vertices of this graph are partitioned into two subsets of of size $k$; $L = \{1, ..., k/2\}$ and $R = \{k/2 + 1, ..., k\}$ such that for $a, b \in \mathcal{A}$ with $a < b$,

$$A_{a,b} = A_{b,a} = [\![ a \in L \text{ and } b \in R]\!] ,$$

while $A_{a,a} = 1$ for all $a \in \mathcal{A}$. Hence, all nodes in this graph have degree $k/2 + 1$ and

$$M = \frac{1}{k/2+1}A = \frac{1}{k/2+1}\begin{pmatrix} I_{k/2} & J_{k/2} \\ J_{k/2} & I_{k/2} \end{pmatrix} .$$

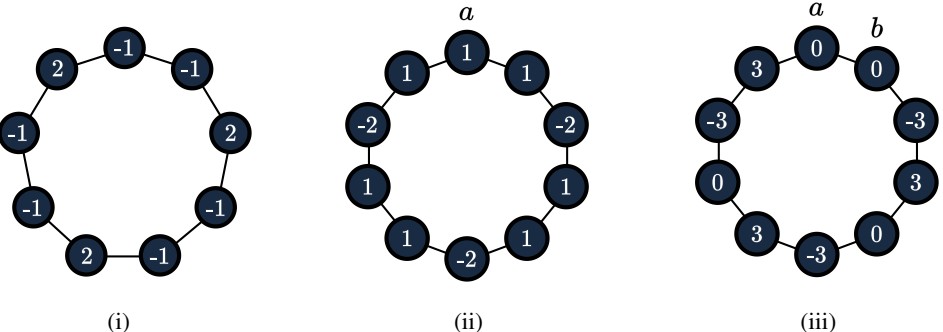

Figure 1: The first figure represents a (non-zero) assignment of weights on a 9-node cycle such that the sum at any three consecutive nodes is $0$, illustrating that $M$ is singular. On a 10-node cycle, the second figure provides a representation of $M^{-1}e_a$; i.e., the $a$-th column of $M^{-1}$, where $a$ is an arbitrary action in $\mathcal{A}$. The last figure is a representation of $M^{-1}(e_a - e_b)$ on the same graph, where $b$ is a neighbor of $a$.

where $I_{k/2}$ is the $(k/2) \times (k/2)$ identity matrix and $J_{k/2}$ is the $(k/2) \times (k/2)$ matrix with all entries equal to $1$. In a similar manner, we can write any vector $x \in \mathbb{R}^k$ in the block form

$$x = \begin{pmatrix} x_L \\ x_R \end{pmatrix}$$

with $x_L, x_R \in \mathbb{R}^{k/2}$. If $k = 2$, then $M$ is singular and the game is hopeless. Otherwise, when $k \geq 4$, $M$ is full rank and its inverse is given by

$$M^{-1} = \frac{1}{k/2 - 1} \begin{pmatrix} (k^2/4 - 1)I_{k/2} - (k/2)J_{k/2} & J_{k/2} \\ J_{k/2} & (k^2/4 - 1)I_{k/2} - (k/2)J_{k/2} \end{pmatrix}.$$

If two actions $a$ and $b$ are on the same side; i.e., $a, b \in L$ or $a, b \in R$, then

$$M^{-1}(e_a - e_b) = (k/2 + 1)(e_a - e_b).$$

Otherwise, if $a \in L$ and $b \in R$, then

$$M^{-1}(e_a - e_b) = (k/2 + 1) \left( e_a - e_b + \frac{1}{k/2 - 1} \begin{pmatrix} -\mathbf{1}_{k/2} \\ \mathbf{1}_{k/2} \end{pmatrix} \right).$$

In either case, both $\left\| M^{-1}(e_a - e_b) \right\|_2$ and $\left\| M^{-1}(e_a - e_b) \right\|_1$ are $\Theta(k)$. Hence, both $\beta_{2,\mathrm{glo}}$ and $\beta_{\mathrm{glo}}$ are $\Theta(k)$. Compared to the previous example, $\beta_{2,\mathrm{glo}}$ grows faster with the number of actions, while $\beta_{\mathrm{glo}}$ remains of the same order.

## J   Lower Bounds

In this section, we provide a proof for Proposition 1. This is an extension of the lower bounds in [31, Appendix G], with a construction that is compatible with the adversarial setting. The proof still relies on designing hard instances in a stochastic linear partial monitoring framework. However, the noise is added to the loss vector itself. This noisy loss vector, moreover, is designed to respect a boundedness condition. The latter is achieved via truncation, following [16], at the cost of a factor logarithmic in the time horizon. We also provide a tailored lower bound for the bandit game with ill-conditioned observers that matches the upper bound we achieved.

### J.1   Stochastic Linear Partial Monitoring with Parameter Noise

We will assume henceforth that, for some positive integer $m$, $n(a) = m$ for all $a \in \mathcal{A}$; this can be taken as $\max_a n(a)$ while padding with zeros the matrices with fewer columns. Moreover, as

mentioned in Section 2, we assume that $\mathcal{B}_2(r) \subseteq \mathcal{L}$ for some $r > 0$. Let $\theta \in \mathbb{R}^d$. We will use $\theta$ to parameterize a stochastic linear partial monitoring problem as follows. At round $t$, the player select a (possibly random) action $A_t \in \mathcal{A}$ and observes the signal $\phi_t = M_{A_t}^\top \ell_t \in \mathbb{R}^m$, where $\ell_t \in \mathbb{R}^d$ is given by

$$\ell_t = \theta + X_t \,,$$

with $X_t$ drawn i.i.d. across rounds from a Gaussian distribution $\mathcal{N}(\mathbf{0}, \Sigma)$, where $\Sigma \in \mathbb{R}^{d \times d}$ is some covariance matrix. Extending [16] (see also [18, Theorem 9]), we describe in the following a truncation scheme that allows adapting to the adversarial setting lower bound constructions that employ unbounded noise. Let $\mathrm{clip} \colon \mathbb{R}^d \to \mathcal{L}$ denote some clipping operation. Consider the following two notions of (random) regret:

$$\widehat{R}_T(\theta) := \max_{a \in \mathcal{A}} \sum_{t=1}^T (\psi_{A_t} - \psi_a)^\top \ell_t$$

$$\widetilde{R}_T(\theta) := \max_{a \in \mathcal{A}} \sum_{t=1}^T (\psi_{A_t} - \psi_a)^\top \mathrm{clip}(\ell_t) \,,$$

where $(\ell_t)_t$ are drawn as described above and the actions $(A_t)_t$ are chosen via some policy $(\pi_t)_t$. Moreover, define the stochastic regret $\overline{R}_T(\theta)$ as:

$$\overline{R}_T(\theta) := \max_{a \in \mathcal{A}} \mathbb{E}_\theta \sum_{t=1}^T (\psi_{A_t} - \psi_a)^\top \ell_t = \max_{a \in \mathcal{A}} \mathbb{E}_\theta \sum_{t=1}^T (\psi_{A_t} - \psi_a)^\top \theta \,,$$

where the expectation is taken over the aforementioned sequences $(\ell_t)_t$ and $(A_t)_t$. (The dependence on $(\pi_t)_t$ is not explicit in the notation since it will be fixed throughout.) The equality in the above display follows from the fact that $X_t$ is centered and independent from $A_t$.

**Lemma 18.** *Let $\Theta$ be a finite subset of $\mathbb{R}^d$ with $|\Theta| = N$, and let $\alpha^* := \max_{a,b \in \mathcal{A}} \|\psi_a - \psi_b\|$. Then,*

$$R_T^* \geq \frac{1}{N} \sum_{\theta \in \Theta} \overline{R}_T(\theta) - \frac{\sqrt{2}\alpha^* T^{3/2}}{N} \sum_{\theta \in \Theta} \left( \|\theta\| + \sqrt{\mathbb{E}\|X_1\|^2} \right) \sqrt{\mathbb{P}_\theta \big( \mathrm{clip}(\ell_1) \neq \ell_1 \big)} \,.$$

*While if there exists some $\mathcal{R} > 0$ such that $\max_{\theta \in \Theta} \widehat{R}_T(\theta) \leq \mathcal{R}$ uniformly, then*

$$R_T^* \geq \frac{1}{N} \sum_{\theta \in \Theta} \overline{R}_T(\theta) - \frac{\mathcal{R}T}{N} \sum_{\theta \in \Theta} \mathbb{P}_\theta \big( \mathrm{clip}(\ell_1) \neq \ell_1 \big) \,.$$

*Proof.* Firstly, note that

$$R_T^* \geq \sup_\theta \mathbb{E}_\theta \widetilde{R}_T(\theta) \geq \frac{1}{N} \sum_{\theta \in \Theta} \mathbb{E}_\theta \widetilde{R}_T(\theta) \quad \text{and} \quad \mathbb{E}\widehat{R}_T(\theta) \geq \overline{R}_T(\theta) \,,$$

the latter holding via Jensen's inequality. Define the event $B := \{\forall t \in [T], \; \mathrm{clip}(\ell_t) = \ell_t\}$. Clearly, $\widetilde{R}_T(\theta) = \widehat{R}_T(\theta)$ whenever $B$ occurs. Hence,

$$\mathbb{E}_\theta \widehat{R}_T(\theta) = \mathbb{E}_\theta \big[ [\![ B ]\!] \widehat{R}_T(\theta) \big] + \mathbb{E}_\theta \big[ [\![ \overline{B} ]\!] \widehat{R}_T(\theta) \big] \leq \mathbb{E}_\theta \widetilde{R}_T(\theta) + \mathbb{E}_\theta \big[ [\![ \overline{B} ]\!] \widehat{R}_T(\theta) \big] \,.$$

Rather crudely, we have that

$$\widehat{R}_T(\theta) = \max_{a \in \mathcal{A}} \sum_{t=1}^T (\psi_{A_t} - \psi_a)^\top (X_t + \theta)$$

$$\leq \max_{a \in \mathcal{A}} \sum_{t=1}^T \|\psi_{A_t} - \psi_a\| \|X_t + \theta\|$$

$$\leq \max_{a,b \in \mathcal{A}} \|\psi_a - \psi_b\| \sum_{t=1}^T \|X_t + \theta\|$$

$$\leq \max_{a,b \in \mathcal{A}} \|\psi_a - \psi_b\| \left( \|\theta\| T + \sum_{t=1}^T \|X_t\| \right) \,.$$

And,

$$\widehat{R}_T^2(\theta) \leq \max_{a,b \in \mathcal{A}} \|\psi_a - \psi_b\|^2 \left( \|\theta\|T + \sum_{t=1}^{T} \|X_t\| \right)^2$$

$$\leq 2 \max_{a,b \in \mathcal{A}} \|\psi_a - \psi_b\|^2 \left( \|\theta\|^2 T^2 + \left( \sum_{t=1}^{T} \|X_t\| \right)^2 \right)$$

$$\leq 2 \max_{a,b \in \mathcal{A}} \|\psi_a - \psi_b\|^2 \left( \|\theta\|^2 T^2 + T \sum_{t=1}^{T} \|X_t\|^2 \right).$$

Now, via the Cauchy-Schwarz inequality,

$$\mathbb{E}_\theta \big[ [\![\overline{B}]\!] \widehat{R}_T(\theta) \big] \leq \sqrt{\mathbb{E}_\theta \big[ [\![\overline{B}]\!]^2 \big] \mathbb{E}_\theta \big[ \widehat{R}_T^2(\theta) \big]}$$

$$= \sqrt{\mathbb{E}_\theta \big[ [\![\overline{B}]\!] \big] \mathbb{E}_\theta \big[ \widehat{R}_T^2(\theta) \big]}$$

$$= \sqrt{\mathbb{P}_\theta(\overline{B}) \mathbb{E}_\theta \big[ \widehat{R}_T^2(\theta) \big]}$$

$$\leq \sqrt{2} \max_{a,b \in \mathcal{A}} \|\psi_a - \psi_b\| \sqrt{\mathbb{P}_\theta(\overline{B})} \left( \|\theta\|T + \sqrt{T} \sqrt{\sum_{t=1}^{T} \mathbb{E}\|X_t\|^2} \right)$$

$$\leq \sqrt{2} \max_{a,b \in \mathcal{A}} \|\psi_a - \psi_b\| \big( \|\theta\| + \sqrt{\mathbb{E}\|X_1\|^2} \big) \sqrt{\mathbb{P}_\theta(\overline{B})} T,$$

Where we have used the fact that $(X_t)_t$ is an i.i.d. sequence. Moreover, a union bound gives that

$$\mathbb{P}_\theta(\overline{B}) \leq \sum_{t=1}^{T} \mathbb{P}_\theta \big( \mathrm{clip}(\ell_t) \neq \ell_t \big) = T \mathbb{P}_\theta \big( \mathrm{clip}(\ell_1) \neq \ell_1 \big).$$

Let $\alpha^* := \max_{a,b \in \mathcal{A}} \|\psi_a - \psi_b\|$. We have shown that

$$\mathbb{E}_\theta \big[ [\![\overline{B}]\!] \widehat{R}_T(\theta) \big] \leq \sqrt{2} \alpha^* \big( \|\theta\| + \sqrt{\mathbb{E}\|X_1\|^2} \big) \sqrt{\mathbb{P}_\theta \big( \mathrm{clip}(\ell_1) \neq \ell_1 \big)} T^{3/2}.$$

In the simpler case when $\max_{\theta \in \Theta} \widehat{R}_T(\theta) \leq \mathcal{R}$ uniformly, we can similarly show that

$$\mathbb{E}_\theta \big[ [\![\overline{B}]\!] \widehat{R}_T(\theta) \big] \leq \mathbb{P}_\theta \big( \mathrm{clip}(\ell_1) \neq \ell_1 \big) \mathcal{R} T.$$

The theorem then follows using that

$$\mathbb{E}_\theta \widetilde{R}_T(\theta) \geq \mathbb{E}_\theta \widehat{R}_T(\theta) - \mathbb{E}_\theta \big[ [\![\overline{B}]\!] \widehat{R}_T(\theta) \big] \geq \overline{R}_T(\theta) - \mathbb{E}_\theta \big[ [\![\overline{B}]\!] \widehat{R}_T(\theta) \big].$$

$\qquad \square$

In the following, we will generally use $\mathbb{P}_\theta$ to denote the probability measure over the history of interaction $\mathcal{H}_T := (A_s, \phi_s)_{s=1}^{T}$ induced by our fixed policy $(\pi_t)_t$ and the environment parameterized by $\theta$. Moreover, $\mathbb{P}_{\theta,a}$ will denote the distribution over observations when action $a$ is played. This is stationary (does not depend on $t$) since $(\ell_t)_t$ is an i.i.d. sequence. Let $D_{\mathrm{KL}}(P \| Q)$ denote the KL-divergence between two measure $P$ and $Q$. The following is a standard result based on the chain rule for the KL-divergence; see, e.g., [35, Exercise 15.8].

**Lemma 19.** *For any $\theta, \theta' \in \mathbb{R}^d$, it holds that*

$$D_{\mathrm{KL}}(\mathbb{P}_\theta \| \mathbb{P}_{\theta'}) = \sum_{t=1}^{T} \mathbb{E}_\theta \big[ D_{\mathrm{KL}}(\mathbb{P}_{\theta,A_t} \| \mathbb{P}_{\theta',A_t}) \big] = \sum_{a \in \mathcal{A}} T_\theta(a) \, D_{\mathrm{KL}}(\mathbb{P}_{\theta,A_t} \| \mathbb{P}_{\theta',A_t}),$$

*where $T_\theta(a) := \mathbb{E}_\theta \sum_{t=1}^{T} [\![A_t = a]\!]$.*

As an extra piece of helpful notation, for $S \subseteq \mathcal{A}$, let $T_\theta(S) := \mathbb{E}_\theta \sum_{t=1}^{T} [\![A_t \in S]\!]$.

## J.2 Regime-based lower bounds

Besides the use of truncation and parameter noise, the constructions used in the following three lower bounds are adapted from [31, Appendix G] and [35, Chapter 37]. Analogous constructions appear also in earlier works on finite partial monitoring like [3, 6].

**Theorem 3.** *Assume that $\mathcal{B}_2(r) \subseteq \mathcal{L}$ for some $r > 0$. If the game is globally but not locally observable then $R_T^*$ is $\widetilde{\Omega}(T^{2/3})$.*

*Proof.* By Definition 1 and Lemma 5, there exist at least three non-duplicate Pareto optimal actions, two of which, say $a$ and $b$, are neighbors and satisfy

$$\psi_a - \psi_b \notin \text{span}\{M_c : c \in \mathcal{N}_{a,b}\}.$$

Hence, we can write $\psi_a - \psi_b = u + v$ with $u$ belonging to $\text{span}\{M_c : c \in \mathcal{N}_{a,b}\}$ and $v \neq 0$ belonging to its orthogonal complement. Let $\theta \in \text{relint}(C_a \cap C_b)$. Then $\mathcal{P}(\theta)$ coincides with $\mathcal{N}_{a,b}$; in particular, there exists a game dependent $\varepsilon > 0$ such that any $c \notin \mathcal{N}_{a,b}$ satisfies $\langle \psi_c - \psi_a, \theta \rangle \geq \varepsilon$. Moreover, we can assume that $\|\theta\| = r/4$ ($C_a \cap C_b$ is a cone). Let $0 < \Delta \leq 1$ be a constant to be chosen to later, and let $q := \frac{r}{4\|v\|} v$. Now, define

$$\theta_a = \theta - \Delta q \qquad \text{and} \qquad \theta_b = \theta + \Delta q,$$

and notice that

$$\langle \psi_a - \psi_b, \theta - \Delta q \rangle = -\frac{\Delta r}{4\|v\|} \langle \psi_a - \psi_b, v \rangle = -\frac{\Delta r \|v\|^2}{4\|v\|} = -\frac{\Delta r \|v\|}{4} < 0.$$

Similarly, $\langle \psi_b - \psi_a, \theta + \Delta q \rangle = -\frac{\Delta r\|v\|}{4} < 0$. For brevity, let $\widetilde{\Delta} := \frac{\Delta r\|v\|}{4}$. Note that any $c \in \mathcal{N}_{a,b}$ satisfies that $c \in \mathcal{P}(\theta)$. Hence,

$$\langle \psi_c - \psi_a, \theta - \Delta q \rangle + \langle \psi_c - \psi_b, \theta + \Delta q \rangle = \langle \psi_c - \psi_b - \psi_c + \psi_a, \Delta q \rangle = \widetilde{\Delta}.$$

Therefore, $\max\{\langle \psi_c - \psi_a, \theta_a \rangle, \langle \psi_c - \psi_b, \theta_b \rangle\} \geq \widetilde{\Delta}/2$. In particular, define

$$S_a := \{c \in \mathcal{N}_{a,b} \mid \langle \psi_c - \psi_a, \theta_a \rangle \geq \widetilde{\Delta}/2\}, \quad S_b := \{c \in \mathcal{N}_{a,b} \mid \langle \psi_c - \psi_a, \theta_a \rangle < \widetilde{\Delta}/2\} = \mathcal{N}_{a,b} \setminus S_a.$$

Also note that $\min\{\langle \psi_c - \psi_a, \theta_a \rangle, \langle \psi_c - \psi_b, \theta_b \rangle\} \geq 0$ for any $c \in \mathcal{N}_{a,b}$ since $\psi_c \in [\psi_a, \psi_b]$. Moving over to actions $c \notin \mathcal{N}_{a,b}$, we have that

$$\langle \psi_c - \psi_a, \theta - \Delta q \rangle \geq \varepsilon - \langle \psi_c - \psi_a, \Delta q \rangle \geq \varepsilon - \Delta \|\psi_c - \psi_a\| \|q\| \geq \varepsilon - \frac{r \Delta \alpha^*}{4},$$

where $\alpha^* := \max_{c,d \in \mathcal{A}} \|\psi_c - \psi_d\| > 0$. Assume that $\Delta \leq \frac{2\varepsilon}{r\alpha^*}$; thus, $\langle \psi_c - \psi_a, \theta_a \rangle \geq \varepsilon/2$, and similarly, $\langle \psi_c - \psi_b, \theta_b \rangle \geq \varepsilon/2$. Moreover, since $\|v\| \leq \alpha^*$, we have that $\Delta \leq \frac{2\varepsilon}{r\|v\|}$, implying that $\widetilde{\Delta} \leq \varepsilon/2$. For brevity, let $\overline{\mathcal{N}}_{a,b} = \mathcal{A} \setminus \mathcal{N}_{a,b}$. Overall, we have that $\theta_a \in C_a$ and $\theta_b \in C_b$. In particular,

$$\overline{R}_T(\theta_a) = \mathbb{E}_{\theta_a} \sum_{t=1}^{T} (\psi_{A_t} - \psi_a)^\top \theta_a \geq \frac{\varepsilon}{2} T_{\theta_a}(\overline{\mathcal{N}}_{a,b}) + \frac{\widetilde{\Delta}}{2} T_{\theta_a}(S_a),$$

and,

$$\overline{R}_T(\theta_b) = \mathbb{E}_{\theta_b} \sum_{t=1}^{T} (\psi_{A_t} - \psi_b)^\top \theta_b \geq \frac{\varepsilon}{2} T_{\theta_b}(\overline{\mathcal{N}}_{a,b}) + \frac{\widetilde{\Delta}}{2} T_{\theta_b}(S_b)$$

$$\geq \frac{\widetilde{\Delta}}{2} T_{\theta_b}(\overline{\mathcal{N}}_{a,b} \cup S_b)$$

$$= \frac{\widetilde{\Delta}}{2} \left(T - T_{\theta_b}(S_a)\right)$$

$$= \frac{\widetilde{\Delta}}{2} \left(T - T_{\theta_a}(S_a)\right) - \frac{\widetilde{\Delta}}{2} \left(T_{\theta_b}(S_a) - T_{\theta_a}(S_a)\right)$$

$$\geq \frac{\widetilde{\Delta}}{2} \left(T - T_{\theta_a}(S_a)\right) - \frac{\widetilde{\Delta}}{2} T \sqrt{(1/2) D_{\text{KL}}(\mathbb{P}_{\theta_a} \| \mathbb{P}_{\theta_b})},$$

where the last inequality follows from [35, Exercise 14.4] and Pinsker's inequality. Hence,

$$\overline{R}_T(\theta_a) + \overline{R}_T(\theta_b) \geq \frac{\varepsilon}{2} T_{\theta_a}(\overline{\mathcal{N}}_{a,b}) + \frac{\widetilde{\Delta}}{2} T\big(1 - \sqrt{(1/2)\, D_{\mathrm{KL}}(\mathbb{P}_{\theta_a} \,\|\, \mathbb{P}_{\theta_b})}\big)\,.$$

Now, let $X_t \sim \mathcal{N}(\mathbf{0}, \sigma^2 \mathbf{I}_d)$. Recall that, $\ell_t = \theta_a + X_t$ under environment $\theta_a$, and $\ell_t = \theta_b + X_t$ under environment $\theta_b$. For any action $c$, it then holds that $\mathbb{P}_{\theta_a,c}$ is $\mathcal{N}(M_c^\top \theta_a, \sigma^2 M_c^\top M_c)$, and that

$$D_{\mathrm{KL}}(\mathbb{P}_{\theta_a,c} \,\|\, \mathbb{P}_{\theta_b,c}) = D_{\mathrm{KL}}\Big(\mathcal{N}(M_c^\top \theta_a, \sigma^2 M_c^\top M_c) \,\Big\|\, \mathcal{N}(M_c^\top \theta_b, \sigma^2 M_c^\top M_c)\Big)\,.$$

Note that the first distribution is supported on $M_c^\top \theta_a + \mathrm{col}(M_c^\top M_c) = \mathrm{col}(M_c^\top)$, which coincides with the support of the second. Hence, the KL-divergence is finite and can be written as (using, e.g., Equation 3.1 by Holbrook [29]):

$$\begin{aligned}
D_{\mathrm{KL}}(\mathbb{P}_{\theta_a,c} \,\|\, \mathbb{P}_{\theta_b,c}) &= \frac{1}{2\sigma^2} \big(M_c^\top \theta_a - M_c^\top \theta_b\big)^\top \big(M_c^\top M_c\big)^\dagger \big(M_c^\top \theta_a - M_c^\top \theta_b\big) \\
&= \frac{1}{2\sigma^2} (\theta_a - \theta_b)^\top M_c \big(M_c^\top M_c\big)^\dagger M_c^\top (\theta_a - \theta_b) \\
&= \frac{1}{2\sigma^2} (\theta_a - \theta_b)^\top M_c M_c^\dagger (\theta_a - \theta_b) \\
&= \frac{1}{2\sigma^2} (-2\Delta q)^\top M_c M_c^\dagger (-2\Delta q) \\
&= \frac{2\Delta^2}{\sigma^2} q^\top M_c M_c^\dagger q = \frac{2\Delta^2}{\sigma^2} \big\| M_c M_c^\dagger q \big\|^2\,,
\end{aligned}$$

where we have used that $M_c M_c^\dagger$ is the orthogonal projection onto $\mathrm{col}(M_c)$. This also means that if $c \in \mathcal{N}_{a,b}$, then $M_c M_c^\dagger q = 0$ by the construction of $q$. Otherwise, we simply have that $\big\| M_c M_c^\dagger q \big\| \leq \|q\| = r/4$. Hence, via Lemma 19, we have that

$$D_{\mathrm{KL}}(\mathbb{P}_{\theta_a} \,\|\, \mathbb{P}_{\theta_b}) \leq \frac{r^2 \Delta^2}{8\sigma^2} T_{\theta_a}(\overline{\mathcal{N}}_{a,b}) = \frac{2\widetilde{\Delta}^2}{\sigma^2 \|v\|^2} T_{\theta_a}(\overline{\mathcal{N}}_{a,b})\,.$$

Which gives that

$$\overline{R}_T(\theta_a) + \overline{R}_T(\theta_b) \geq \frac{\varepsilon}{2} T_{\theta_a}(\overline{\mathcal{N}}_{a,b}) + \frac{\widetilde{\Delta}}{2} T\left(1 - \frac{\widetilde{\Delta}}{\sigma \|v\|} \sqrt{T_{\theta_a}(\overline{\mathcal{N}}_{a,b})}\right)\,.$$

Now, set $\Delta = 2\sigma T^{-1/3}/r$, assuming $T$ is large enough so that $\Delta \leq \min\{1, \frac{2\varepsilon}{r\alpha^*}\}$. Precisely, assuming that $T^{1/3} \geq \sigma \max\{2/r, \alpha^*/\varepsilon\}$. We then have that $\widetilde{\Delta} = \sigma \|v\| T^{-1/3}/2$. Now, if $T_{\theta_a}(\overline{\mathcal{N}}_{a,b}) \leq T^{2/3}$, then,

$$\overline{R}_T(\theta_a) + \overline{R}_T(\theta_b) \geq \frac{\widetilde{\Delta}}{2} T\left(1 - \frac{\widetilde{\Delta}}{\sigma \|v\|} T^{1/3}\right) = \frac{\widetilde{\Delta}}{4} T = \frac{\sigma \|v\|}{8} T^{2/3}\,.$$

Otherwise, we directly get that $\overline{R}_T(\theta_a) + \overline{R}_T(\theta_b) \geq \frac{\varepsilon}{2} T^{2/3}$. Hence, in total, we have that

$$\frac{1}{2}\big(\overline{R}_T(\theta_a) + \overline{R}_T(\theta_b)\big) \geq \min\left\{\frac{\sigma \|v\|}{16}, \frac{\varepsilon}{4}\right\} T^{2/3}\,.$$

We now prepare for an application of Lemma 18. First, notice that $\mathbb{E}\|X_1\|^2 = \sigma^2 d$. Second, pick the clipping operator as

$$\mathrm{clip}(\ell_t) := [\![\|\ell_t\| \leq r]\!] \ell_t + [\![\|\ell_t\| > r]\!] \frac{r \ell_t}{\|\ell_t\|}\,.$$

Then, for $\theta' \in \{\theta_a, \theta_b\}$,

$$\begin{aligned}
\mathbb{P}_{\theta'}\big(\mathrm{clip}(\ell_1) \neq \ell_1\big) = \mathbb{P}_{\theta'}\big(\|\ell_t\| > r\big) &\leq \mathbb{P}_{\theta'}\big(\|\theta'\| + \|X_1\| > r\big) \\
&\leq \mathbb{P}_{\theta'}\big(\|X_1\| > r/2\big) \\
&= \mathbb{P}_{\theta'}\big(\|X_1\|^2 > r^2/4\big)\,,
\end{aligned}$$

which uses that $\|\theta_a\| \leq \|\theta\| + \Delta\|q\| \leq r/2$, with the same holding for $\theta_b$. Lemma 1 by Laurent and Massart [41] gives that for $x > 0$,

$$\mathbb{P}\big(\|X_1\|^2 > \sigma^2 d + 2\sigma^2\sqrt{dx} + 2\sigma^2 x\big) \leq e^{-x}.$$

Assuming $\sigma^2 \leq r^2/(8d)$, setting $x = r^2/(80\sigma^2)$ gives that

$$\mathbb{P}\big(\|X_1\|^2 > r^2/4\big) \leq e^{-r^2/(80\sigma^2)}.$$

Then, Lemma 18 yields that

$$R_T^* \geq \frac{1}{2}\sum_{\theta'\in\{\theta_a,\theta_b\}}\overline{R}_T(\theta') - \frac{\sqrt{2}\alpha^*T^{3/2}}{2}\sum_{\theta'\in\{\theta_a,\theta_b\}}\big(\|\theta'\| + \sqrt{\mathbb{E}\|X_1\|^2}\big)\sqrt{\mathbb{P}_{\theta'}\big(\mathrm{clip}(\ell_1)\neq\ell_1\big)}$$

$$\geq \min\bigg\{\frac{\sigma\|v\|}{16},\frac{\varepsilon}{4}\bigg\}T^{2/3} - \sqrt{2}\alpha^*T^{3/2}\big(r/2 + \sigma\sqrt{d}\big)e^{-r^2/(160\sigma^2)}$$

$$\geq \min\bigg\{\frac{\sigma\|v\|}{16},\frac{\varepsilon}{4}\bigg\}T^{2/3} - \sqrt{2}\alpha^*rT^{3/2}e^{-r^2/(160\sigma^2)},$$

where the last step uses that $\sigma \leq r/\sqrt{8d}$. Now, we can pick $\sigma = \frac{r}{\sqrt{240\log(T)}}$, which is a valid choice when $T$ is large enough so that $\sigma^2 \leq r^2/(8d)$, meaning that $T \geq e^{d/30}$. With this choice of $\sigma$ we obtain that

$$R_T^* \geq \min\bigg\{\frac{r\|v\|}{32\sqrt{60\log(T)}},\frac{\varepsilon}{4}\bigg\}T^{2/3} - \sqrt{2}\alpha^*r,$$

which concludes the proof. $\qquad\square$

**Theorem 4.** *Assume that $\mathcal{B}_2(r) \subseteq \mathcal{L}$ for some $r > 0$. If the game is locally observable and there is at least one pair of non-duplicate Pareto optimal actions, then $R_T^*$ is $\widetilde{\Omega}(\sqrt{T})$.*

*Proof.* By the connectedness of the neighborhood graph, there must be at least two pair of neighboring actions, which we again denote by $a$ and $b$. Most of the proof proceeds almost identically to that of Theorem 3. The main difference is that $\psi_a - \psi_b \in \mathrm{span}\{M_c : c \in \mathcal{N}_{a,b}\}$ by local observability. Moreover, $\overline{\mathcal{N}}_{a,b}$ could be empty. Nevertheless, we can directly take

$$v := \psi_a - \psi_b,$$

which is strictly positive, and again define $q := \frac{r}{4\|v\|}v$. Then, for some $\theta \in \mathrm{relint}(C_a \cap C_b)$, we again choose

$$\theta_a = \theta - \Delta q \qquad \text{and} \qquad \theta_b = \theta + \Delta q.$$

Following the proof of Theorem 3 verbatim, we reach that

$$\overline{R}_T(\theta_a) + \overline{R}_T(\theta_b) \geq \frac{\varepsilon}{2}T_{\theta_a}(\mathcal{N}_{a,b}) + \frac{\widetilde{\Delta}}{2}T\big(1 - \sqrt{(1/2)D_{\mathrm{KL}}(\mathbb{P}_{\theta_a}\,\|\,\mathbb{P}_{\theta_b})}\big)$$

$$\geq \frac{\widetilde{\Delta}}{2}T\big(1 - \sqrt{(1/2)D_{\mathrm{KL}}(\mathbb{P}_{\theta_a}\,\|\,\mathbb{P}_{\theta_b})}\big).$$

We choose, again, that $X_t \sim \mathcal{N}(\mathbf{0},\sigma^2\boldsymbol{I}_d)$. Now, however, we use the bound $D_{\mathrm{KL}}(\mathbb{P}_{\theta_a,c}\,\|\,\mathbb{P}_{\theta_b,c}) \leq \frac{2\widetilde{\Delta}^2}{\sigma^2\|v\|^2}$ for all actions $c$, even the ones in $\mathcal{N}_{a,b}$. This yields via Lemma 19 that

$$\overline{R}_T(\theta_a) + \overline{R}_T(\theta_b) \geq \frac{\widetilde{\Delta}}{2}T\bigg(1 - \frac{\widetilde{\Delta}}{\sigma\|v\|}\sqrt{T}\bigg).$$

Taking $\Delta = 2\sigma T^{-1/2}/r$, (assuming again that $T$ is large enough to satisfy the condition set on $\Delta$ in the proof of Theorem 3),[7] we have that $\widetilde{\Delta} = \sigma\|v\|T^{-1/2}/2$, which implies that

$$\overline{R}_T(\theta_a) + \overline{R}_T(\theta_b) \geq \frac{\sigma\|v\|}{8}\sqrt{T}.$$

---

[7]If $\overline{\mathcal{N}}_{a,b}$ is empty (also meaning that $T_\theta(\overline{\mathcal{N}}_{a,b}) = 0$), one can satisfy the conditions on $\Delta$ in the proof of Theorem 3 taking $\varepsilon$ as an arbitrarily large number.

The rest of the proof again follows that of Theorem 3, with an identical use of Lemma 18 and tuning of $\sigma$, finally yielding that

$$R_T^* \geq \frac{r\|v\|}{32\sqrt{60\log(T)}}\sqrt{T} - \sqrt{2}\alpha^* r$$

for sufficiently large $T$. $\qquad\qquad\square$

**Theorem 5.** *Assume that $\mathcal{B}_2(r) \subseteq \mathcal{L}$ for some $r > 0$. If the game is not globally observable, then $R_T^*$ is $\Omega(T)$.*

*Proof.* The game not being globally observable means that there exists $a, b \in \mathcal{A}$ such that

$$\psi_a - \psi_b \notin \text{span}\{M_c : c \in \mathcal{A}\}.$$

This implies that they have distinct features, which in turn implies that there are at least two non-duplicate Pareto optimal actions (vertices). Now, if the property above holds for some pair of actions $(a, b)$, then it must hold for at least one pair of non-duplicate Pareto optimal actions. This is because the features of any action can be written as a convex combination (with positive weights) of the feature of one or more Pareto optimal actions. Hence, $\psi_a - \psi_b$ can be written as a linear combination of feature differences for Pareto optimal actions. This implies that we can take $a$ and $b$ to be (non-duplicate) Pareto optimal actions. In fact, we can take them to be neighbors too; since the neighborhood graph is connected, we can find a path $(a_i)_{i=1}^s$ where $a_1 = a$, $a_s = b$, and $(a_i, a_{i+1})$ are neighbors (endpoint of an edge in the polytope $\mathcal{V} = \text{co}(\mathcal{X})$). And since $\psi_a - \psi_b = \sum_{i=1}^{s-1}(\psi_{a_i} - \psi_{a_{i+1}})$, at least one pair of neighbors must satisfy the non-observability condition above. Thus, without loss of generality, $a$ and $b$ are assumed to be neighbors.

Now, define $v$ as the component of $\psi_a - \psi_b$ in the orthogonal complement of $\text{span}\{M_c : c \in \mathcal{A}\}$, and set $q := \frac{r}{4\|v\|}v$. Neither of which is zero by assumption. With this altered choice of $v$ (and the corresponding one for $q$), we can follow the proof of Theorem 3 verbatim till we reach that

$$\overline{R}_T(\theta_a) + \overline{R}_T(\theta_b) \geq \frac{\varepsilon}{2}T_{\theta_a}(\overline{\mathcal{N}}_{a,b}) + \frac{\widetilde{\Delta}}{2}T\big(1 - \sqrt{(1/2)\,D_{\text{KL}}(\mathbb{P}_{\theta_a}\,\|\,\mathbb{P}_{\theta_b})}\big)$$

$$\geq \frac{\widetilde{\Delta}}{2}T\big(1 - \sqrt{(1/2)\,D_{\text{KL}}(\mathbb{P}_{\theta_a}\,\|\,\mathbb{P}_{\theta_b})}\big).$$

Observe now that $q$ belongs to the orthogonal complement of $\text{col}(M_c)$ for all $c \in \mathcal{A}$, meaning that, via the derivation in the proof of Theorem 3 (again with $X_t \sim \mathcal{N}(\mathbf{0}, \sigma^2 \boldsymbol{I}_d)$), $D_{\text{KL}}(\mathbb{P}_{\theta_a,c}\,\|\,\mathbb{P}_{\theta_b,c}) = 0$. Hence, $D_{\text{KL}}(\mathbb{P}_{\theta_a}\,\|\,\mathbb{P}_{\theta_b}) = 0$ via Lemma 19. Then, choosing $\Delta = \min\{1, \frac{2\varepsilon}{r\alpha^*}\}$ yields that[8]

$$\overline{R}_T(\theta_a) + \overline{R}_T(\theta_b) \geq \min\Big\{\frac{r\|v\|}{8}, \frac{\varepsilon\|v\|}{4\alpha^*}\Big\}T.$$

Next, a similar application of Lemma 18 to that in the proof of Theorem 3 gives (with a sufficiently small $\sigma$) that

$$R_T^* \geq \min\Big\{\frac{r\|v\|}{16}, \frac{\varepsilon\|v\|}{8\alpha^*}\Big\}T - \sqrt{2}\alpha^* r T^{3/2}e^{-r^2/(160\sigma^2)}.$$

Since $\sigma$ does not appear in the first term (in fact, adding noise here is redundant), it can be chosen sufficiently small so that

$$R_T^* \geq \min\Big\{\frac{r\|v\|}{32}, \frac{\varepsilon\|v\|}{16\alpha^*}\Big\}T,$$

proving the theorem. $\qquad\qquad\square$

### J.3 A lower bound for bandits with ill-conditioned observers

As defined in Section 2, we have in this game that $d = k$, $\psi_a = e_a$, and $M_a = (1 - \varepsilon)\mathbf{1}_k/k + \varepsilon e_a$ for some $\varepsilon \in (0, 1]$. Here, we take $\mathcal{L} = \mathcal{B}_\infty(1)$ (see Appendix I). In fact, we clip the losses in this lower bound construction to belong to $[0, 1]^k$. This example is adapted from [18, Section 6], where they consider the case when $\psi_a$ too is $(1 - \varepsilon)\mathbf{1}_k/k + \varepsilon e_a$, and prove a $\widetilde{\Omega}(\sqrt{kT})$ lower bound in their Theorem 9. The game we consider here is harder as asserts the following theorem.

---

[8]As remarked in the proof of Theorem 4, we can take $\varepsilon$ to be an arbitrarily large number if $\overline{\mathcal{N}}_{a,b}$ is empty.

**Theorem 6.** *In the bandit game with ill-conditioned observers, it holds that $R_T^*$ is $\widetilde{\Omega}(1/\varepsilon\sqrt{kT})$.*

*Proof.* Besides the altered definition of $\psi_a$, the proof is essentially identical to that of [18, Proposition 8]. Nonetheless, we report here the full proof in our notation for completeness. Consider $k$ environments $\{\theta_a\}_{a\in\mathcal{A}}$ such that

$$\theta_a := \left(\frac{1}{2} + \Delta\frac{1-\varepsilon}{k}\right)\mathbf{1}_k - \Delta e_a\,,$$

besides an auxiliary environment $\theta_0 := (1/2)\mathbf{1}_k$, where $0 < \Delta \leq 1/4$. Notice then that for any $b \in \mathcal{A} \setminus \{a\}$,

$$\langle \psi_b - \psi_a, \theta_a \rangle = \langle e_b - e_a, \theta_a \rangle = \Delta\,.$$

Hence, $\theta_a \in C_a$, and in a similar fashion to the proof of Theorem 3, we have that

$$\overline{R}_T(\theta_a) = \mathbb{E}_{\theta_a} \sum_{t=1}^{T} \langle \psi_{A_t} - \psi_a, \theta_a \rangle = \Delta(T - T_{\theta_a}(a))$$

$$= \Delta(T - T_{\theta_0}(a)) - \Delta(T_{\theta_a}(a) - T_{\theta_0}(a))$$

$$\geq \Delta(T - T_{\theta_0}(a)) - \Delta T\sqrt{(1/2)\,D_{\mathrm{KL}}(\mathbb{P}_{\theta_0}\,\|\,\mathbb{P}_{\theta_a})}\,.$$

In this game, we will take $X_t$ to be distributed according to $\mathcal{N}(\mathbf{0}, \sigma^2 \mathbf{1}_d \mathbf{1}_d^\top)$, for some $\sigma > 0$. This means that in environment $\theta_a$, $\ell_t = \theta_a + Z_t \mathbf{1}_k$, where $(Z_t)_t$ is an i.i.d. sequence of random variables, each sampled from $\mathcal{N}(0, \sigma^2)$. Notice that, for all $c \in \mathcal{A}$, $M_c \in \Delta_k$; hence, $M_c^\top \ell_t = M_c^\top \theta_a + Z_t$. So we get that

$$D_{\mathrm{KL}}(\mathbb{P}_{\theta_0, c}\,\|\,\mathbb{P}_{\theta_a, c}) = D_{\mathrm{KL}}\Big(\mathcal{N}\big(M_c^\top \theta_0, \sigma^2\big)\,\Big\|\,\mathcal{N}\big(M_c^\top \theta_a, \sigma^2\big)\Big)$$

$$= \frac{1}{2\sigma^2}(\theta_0 - \theta_a)^\top M_c M_c^\top (\theta_0 - \theta_a)\,.$$

Notice, moreover that

$$(\theta_0 - \theta_a)^\top M_c = \Delta\Big(e_a - \frac{1-\varepsilon}{k}\mathbf{1}_k\Big)^\top\Big(\frac{1-\varepsilon}{k}\mathbf{1}_k + \varepsilon e_c\Big) = \varepsilon\Delta[\![a = c]\!]\,.$$

Thus, $D_{\mathrm{KL}}(\mathbb{P}_{\theta_0, c}\,\|\,\mathbb{P}_{\theta_a, c}) = \frac{\varepsilon^2 \Delta^2}{2\sigma^2}[\![a = c]\!]$ Lemma 19 then implies that

$$\overline{R}_T(\theta_a) \geq \Delta(T - T_{\theta_0}(a)) - \Delta T\sqrt{\frac{\varepsilon^2 \Delta^2}{4\sigma^2}T_{\theta_0}(a)} = \Delta(T - T_{\theta_0}(a)) - \frac{\varepsilon\Delta^2}{2\sigma}T\sqrt{T_{\theta_0}(a)}\,.$$

Hence, via Jensen's inequality,

$$\frac{1}{k}\sum_{a\in\mathcal{A}}\overline{R}_T(\theta_a) \geq \Delta(T - T/K) - \frac{\varepsilon\Delta^2}{2\sigma}T\sqrt{T/k}$$

$$\geq \frac{\Delta}{2}T\Big(1 - \frac{\varepsilon\Delta}{\sigma}\sqrt{T/k}\Big)$$

$$= \frac{\sigma}{8\varepsilon}\sqrt{kT}$$

where in the last step we have picked $\Delta = \frac{\sigma\sqrt{k}}{2\varepsilon\sqrt{T}}$, assuming that $T \geq 4\sigma^2 k/\varepsilon^2$ so that $\Delta \leq 1/4$. To apply Lemma 18, we choose the clipping operator as follows (where $\ell_t(c) = e_c^\top \ell_t$):

$$[\mathrm{clip}(\ell_t)](c) := \max\{\min\{1, \ell_t(c)\}, 0\}\,.$$

Note that under any environment $\theta_a$, $1/4 \leq \theta_a(c) \leq 3/4$ for all $c \in \mathcal{A}$ by the definition of $\theta_a$ and the fact that $0 \leq \Delta \leq 1/4$. Hence, since $\ell_t(a) = \theta_a(c) + Z_t$, if $\mathrm{clip}(\ell_t) \neq \ell_t$, then it must be that $|Z_t| > 1/4$. And since $Z_t \sim \mathcal{N}(0, \sigma^2)$, we have that $\mathbb{P}(|Z_t| > 1/4) \leq 2\exp(-1/(32\sigma^2))$. Moreover, under environment $\theta_a$, it holds for any pair of actions $c$ and $d$ that

$$\langle \psi_c - \psi_d, \ell_t \rangle = \langle e_c - e_d, \theta_a + Z_t \mathbf{1}_k \rangle = \langle e_c - e_d, \theta_a \rangle \leq \Delta,$$

which implies that $\widehat{R}_T(\theta_a) \leq \Delta T$ uniformly. Therefore, we can apply the second part of Lemma 18 to obtain that

$$R_T^* \geq \frac{\sigma}{8\varepsilon}\sqrt{kT} - 2\Delta T^2 e^{-1/32\sigma^2} = \frac{\sigma}{\varepsilon}\sqrt{kT}\Big(\frac{1}{8} - Te^{-1/32\sigma^2}\Big) = \frac{\varepsilon^{-1}}{64\sqrt{2\log(16T)}}\sqrt{kT}\,,$$

where the first equality uses our choice of $\Delta$, and to obtain the second we set $\sigma = \frac{1}{4\sqrt{2\log(16T)}}$. $\quad\square$

