# OpenReview forum: "Instance-Dependent Regret Bounds for Nonstochastic Linear Partial Monitoring"
_NeurIPS.cc/2025/Conference — NeurIPS 2025 poster_

### Official Review · Reviewer_aKH4 · 2025-06-26

**Clarity:** 2
**Significance:** 3
**Originality:** 3
**Rating:** 5
**Confidence:** 2

**Summary:**

This paper studies the problem of adversarial linear partial monitoring with finitely many actions. This setting differs from both stochastic linear partial monitoring and finite partial monitoring, which have been studied previously. The authors show that the minimax regret rates are $O(\sqrt{T})$ and $O(T^{2/3})$ (up to logarithmic factors) for locally observable and globally observable games, respectively. The upper bounds are achieved by combining the "exploration by optimization" framework of Lattimore and Szepesvári with the linear structure of the problem, resulting in a computationally efficient algorithm. The regret bounds are also specialized to various settings, such as the full-information setting and the linear bandit, recovering existing optimal results.

**Questions:**

No further questions.

**Ethical Concerns:**

["NO or VERY MINOR ethics concerns only"]

**Final Justification:**

My concern regarding the finite-action assumption is resolved. Please see my response to the authors below. Since this is a technically solid paper, I will recommend acceptance.

**Limitations:**

Yes.

**Paper Formatting Concerns:**

No issues.

**Quality:**

3

**Strengths And Weaknesses:**

# Strengths

I have read the main paper and skimmed through Appendices A, C, and D. The paper is clearly written, and the related works are adequately addressed. The theoretical results I checked are rigorously supported.

---
### Novelty and Significance

The adversarial linear partial monitoring is a fundamental problem in online learning, yet it has not been studied thoroughly in the literature. This paper provides a comprehensive characterization of the minimax regret rate when the action set is finite, which is a valuable contribution.

As far as I understand, the main technical contribution of the work is a computationally efficient algorithm (Algorithm 1), that cleverly leverages the linear structure within the exploration by optimization (ExO) framework. In contrast, the general ExO framework requires solving an infinite-dimensional optimization problem, which may be intractable.

---

# Weaknesses

### The finite-action assumption

The paper assumes that there are only finitely many actions (Line 67 and 94). However, as far as I understand, the linear structure is typically used to address large or even continuous action spaces. Therefore, I am confused about the motivation and advantages of studying the finite-action linear setting. In particular,

1. What is the motivation for studying the linear partial monitoring problem with *finite* actions? Are there any important applications that can be modeled by this problem?
2. Besides computational efficiency, are there other key differences between the finite partial monitoring problem and the finite linear partial monitoring problem?
3. What are the main challenges in extending the current results to the infinite-action case?

I will raise my score if the above concerns are resolved.

---

The following comments are minor.

### Clarity

Generally speaking, this paper is clearly written. However, due to the complexity of the notation, it took me some time to understand the results. Below are some comments that could help improve clarity.

1. In Lemma 5, the loss estimate is defined as $\hat{y}_t = (H^\top-\mathbf{1}_dx_t^\top)Q(p_t)^\dagger M\_{A\_t}\phi_t$. From my understanding, this is an unbiased estimate of the true loss $y_t$, i.e., $\mathbb{E}_t[\hat{y}_t]=y_t$. The definition can be derived by using least squares similar to [1, Chapter 27]. To improve clarity, an explanation of how this estimate is constructed would be helpful.
2. Due to the complex notation, it may be difficult for readers to specialize the results (Theorem 10 and Theorem 11) to different settings. I would suggest including derivations of some of the special cases, such as full information and linear bandits, in the appendix.
3. It is stated in Line 343-347 that "Overall, our bounds seem more versatile [...]." I couldn't see the reason for this claim. Could you elaborate more on the advantages of your bounds?

---
### Typos
1. In Line 299, under the summation in the definition of $\hat{p}$, should it be $a\in\mathcal{A}^\ast$?
2. In Line 354, I believe it should refer to Theorem 10 rather than Theorem 11.
3. In Line 349-351, it is mentioned that in Section 4, we choose $\hat{p}=q$. However, this does not seem to be the case (see Line 299). I assume there are some typos here.

---

References
1. Lattimore and Szepesvári. Bandit Algorithms. 2020.

---

> ### Author Rebuttal · Authors · 2025-07-31
>
> We thank the reviewer for their comments and suggestions. We address the raised questions below.
>
> > What is the motivation for studying the linear partial monitoring problem with finite actions? Are there any important applications that can be modeled by this problem? What are the main challenges in extending the current results to the infinite-action case?  Besides computational efficiency, are there other key differences between the finite partial monitoring problem and the finite linear partial monitoring problem?
>
> Firstly, considering finite actions in problems with linear structure is commonplace in the literature, especially in the adversarial setting; see, e.g., Audibert et al. (2011), Neu and Olkhovskaya (2020), and Lee et al. (2021). In such settings, one strives to achieve regret bounds that depend on the dimension $d$ of the space where the actions live, as opposed to the number of actions $k$ (or at least only logarithmically so) so that one can handle very large actions sets while incurring small regret. This mild dependence on the number of actions also allows obtaining  guarantees for compact action sets via discretization arguments, see Dani et al. (2007) and Bubeck et al. (2012).
>
> In our case, considering a finite actions setting in partial monitoring (as is the usual case in prior works on adversarial partial monitoring, e.g., [29,33]) still allows modeling interesting problems like the composite graph feedback problem and the linear dueling bandits problem that we discuss in the manuscript. Moreover, as we mention in our third response to Reviewer bC8o, it is important to note that passing to the continuous action set case is challenging and requires new ideas in the analysis. In fact, already in the stochastic setting, obtaining a full classification theorem for the regret in this case is still an open problem [24,25]; the example provided in [24, Section 2.4] shows that in determining the achievable rate in $T$, the geometry of the action set also matters, besides the observability conditions.
>
> Compared to finite partial monitoring (i.e., finite actions \emph{and} outcomes), the linear setting allows the outcome set to be infinite, which lends more modeling power to the framework. Moreover, the linear setting provides a more intuitive observation structure, free from the combinatorial complications of the finite model. Thanks to this structure, we are able to obtain bounds that do not depend on the number of outcomes and depend only logarithmically on the number of actions. Instead, the problem complexity is captured in the bounds by the features' dimension $d$ (or more refined quantities) and interpretable game-dependent constants like $\beta_\mathrm{loc}$ and $\beta_\mathrm{glo}$ that provide a measure of the alignment between observations and losses.
>
> References:
>
> Audibert, J. Y., Bubeck, S., \& Lugosi, G. Minimax policies for combinatorial prediction games. COLT 2011.
>
> Neu, G., \& Olkhovskaya, J. Efficient and robust algorithms for adversarial linear contextual bandits. COLT 2020.
>
> Lee, C. W., Luo, H., Wei, C. Y., Zhang, M., \& Zhang, X. Achieving near instance-optimality and minimax-optimality in stochastic and adversarial linear bandits simultaneously. ICML 2021.
>
> Dani, V., Kakade, S. M., \& Hayes, T. The price of bandit information for online optimization. NeurIPS 2007.
>
> Bubeck, S., Cesa-Bianchi, N., \& Kakade, S. M. Towards minimax policies for online linear optimization with bandit feedback. COLT 2012.
>
> > It is stated in Line 343-347 that "Overall, our bounds seem more versatile [...]." I couldn't see the reason for this claim. Could you elaborate more on the advantages of your bounds?
>
> This is in reference to the bounds of [24] and [25] in the stochastic setting. (We also discuss their results further in our first response to Reviewer bC8o.) Our remark about versatility was meant to highlight that our bounds can recover near-optimal results in feedback-rich problems like full information and feedback graphs. In contrast, the bounds in [24] and [25] always depend polynomially on the dimensionality of the loss space or the space spanned by the columns of the observation matrices, which results in a polynomial (and sub-optimal) dependence on the number of actions in the aforementioned problems. Nonetheless, a more refined analysis of their information directed sampling approach can likely recover the optimal bounds for these problems in the stochastic setting.
>
> > Typo 1: In Line 299, under the summation in the definition of $\hat{p}$, should it be $a\in\mathcal{A}^*$?
>
> It can be written either way. Indeed, $q$, via its definition in (4), gives zero weight to actions outside of $\mathcal{A}^*$.
>
> > Typo 2: In Line 354, I believe it should refer to Theorem 10 rather than Theorem 11.
>
> Yes, thanks for pointing this out.
>
> > Typo 3: In Line 349-351, it is mentioned that in Section 4, we choose $\hat{p}=q$. However, this does not seem to be the case (see Line 299). I assume there are some typos here.
>
> The choice of $\hat{p}=q$ (in Lines 349-351) is made in the globally observable case, whereas line 299 refers to the choice of $\hat{p}$ in the locally observable case. This key component is chosen differently in the analysis of locally and globally observable games.

---

> > ### Comment · Reviewer_aKH4 · 2025-08-03
> >
> > I appreciate the authors’ detailed response.
> >
> > Regarding the finite-action assumption, I overlooked its power to model problems with infinite outcomes. I have also read the response to Reviewer bC8o and understood that extending the results to continuous actions are challenging. Thank you for clarifying these points. I have raised my score from 4 to 5 and will continue to recommend acceptance.
> >
> > Regarding Typo 3, I am still a bit confused. I understand that $\hat{p}=q$ is used for the globally observable case and that Line 299 applies to the locally observable case. However, from my understanding, the sentence on Lines 349-351 suggests that $\hat{p}=q$ is used in the locally observable case (Section 4).

---

> > > ### Author Response · Authors · 2025-08-03
> > >
> > > Thank you for your response and for amending your evaluation.
> > >
> > > Regarding the sentence on Lines 349-351, its phrasing is indeed potentially confusing; the intention was to mention that the analysis in the globally observable case picks up from the initial decomposition performed in the previous section, but departs by choosing $\hat{p}$ in a different manner and augmenting the role of $\pi$ as described. We will clarify this in the revised version, thank you for pointing this out.

---

### Official Review · Reviewer_Tk4m · 2025-07-03

**Clarity:** 2
**Significance:** 3
**Originality:** 2
**Rating:** 4
**Confidence:** 4

**Summary:**

This paper proves an instance-dependent upper bound for the regret for non-stochastic adversarial linear partial monitoring which covers possibly infinite outcome spaces. The problem-independent lower regret bound and upper regret bound is characterized by the global or local observability.

**Questions:**

1. Could authors explain the definition of $\Lambda, H$ and $B$?

2. Could authors provide an overview of the analysis on locally observable and globally observable cases, before going into the proof, respectively?

3. What is the novel technique developed in this paper? What are the main challenges to derive the results in locally and globally observable cases?

**Ethical Concerns:**

["NO or VERY MINOR ethics concerns only"]

**Final Justification:**

I appreciate the author's detailed responses.
The responses helped me understand the work better with clear novelty.
I will keep my score as is with higher confidence.

**Limitations:**

Yes.

**Paper Formatting Concerns:**

I could not find any formatting concerns in this paper.

**Quality:**

3

**Strengths And Weaknesses:**

**Strengths**

1. The proposed non-stochastic linear partial monitoring model covers a wide range of settings including linear bandits, graph bandits and ill-conditioned bandits, which contributes in many ways to the literature. The provided examples are helpful to understand the coverage of the partial monitoring model.

2. The proposed algorithm is simple yet principled to obtain the proven instance-dependent regret bound. Explanations on the terms that appear in the regret bound provides understanding of the procedure.

**Weaknesses**

1. The paper is quite focused on technical description and the novelty or significance of the results are not highlighted well. I suggest including comparison with other works especially for Lemma 4,5 and Theorem 11. Including the literal meaning of definitions of $\Lambda, H$ and $B$.

---

> ### Author Rebuttal · Authors · 2025-07-31
>
> We thank the reviewer for their comments and questions. We address below the raised concerns.
>
> > The paper is quite focused on technical description...Could authors provide an overview of the analysis on locally observable and globally observable cases, before going into the proof, respectively?...What are the main challenges to derive the results in locally and globally observable cases?
>
> We kindly refer the reviewer to our first response to Reviewer 5qSe as it touches a similar comment. In short, we remark that most of the main body is devoted to providing an overview of the key aspects in designing the algorithm and analyzing it. In particular, Sections 4 and 5 are indeed mostly dedicated to laying down the main ideas of the analysis for globally and locally observable games, highlighting the challenges posed in each of the two observation regimes and pointing out the new techniques we employed in compared to finite partial monitoring works, all without detailing full formal proofs.
>
> > [T]he novelty or significance of the results are not highlighted well. I suggest including comparison with other works especially for Lemma 4,5 and Theorem 11...What is the novel technique developed in this paper?
>
> Regarding novelty and significance, we kindly refer the reviewer to our second response to Reviewer 5qSe. There, we point to a main novel technique discussed in Section 4 (regarding the choice of exploration distributions in the analysis of locally observable games), and discuss further the significance of our bounds in references to known results in the literature. This latter aspect is expanded upon further still in our first response to Reviewer bC8o, to which we kindly refer the reviewer for more details.
>
> Regarding the subset of the results cited by the reviewer, the results in Section 3 (which include Lemmas 4 and 5) motivate and explain the EXO approach in our particular setting. The general exposition in that section is naturally inspired by previous works on EXO like [30] and [33], as highlighted in the text, with the distinguishing aspect being our choice of estimator and how it affects the structure of the optimization problem characterizing the algorithm. In particular, Lemma 4 is a generic regret decomposition and, Lemma 5 further instantiates this result in light of our choice of estimator; both results motivate the definition of $\Lambda$ and the design of Algorithm 1. Theorem 11 is the analogue of Theorem 10 in the globally observable case. It provides a general bound on $\Lambda^*$, which leads to a $T^{2/3}$ bound that depends on a game-dependent constant. This provides a clear dependence on the game structure compared to [30] and rivals the bounds obtained in [24,25] in the stochastic setting.
>
> > Could authors explain the definition of $\Lambda$, $H$ and $B$?
>
> The function $\Lambda_{\eta, q}$ is defined in Line 242. This is the function to be minimized at every round by the learner in our EXO approach; its definition is directly dictated by the bound in Lemma 5, as mentioned in the preceding paragraph. $H$ is defined in Line 215; it is the $d \times k$ matrix with the feature vectors of the actions as columns. $B$ is defined in Line 224; this is a different representation of the design matrix $Q$ that can be made invertible and is used to simplify some expressions.

---

### Official Review · Reviewer_bC8o · 2025-07-03

**Clarity:** 3
**Significance:** 3
**Originality:** 3
**Rating:** 5
**Confidence:** 3

**Summary:**

This paper studies the setting of adversarial linear partial monitoring. They provide an implementable algorithm based on the “Exploration-by-optimization” method that obtains optimal regret bounds for locally and globally observable games respectively. Moreover, their regret bounds scale with instance-dependent quantities comparable to the one obtained in the stochastic version of this problem.

**Questions:**

- It is stated that the setting studied in [30] is more general than the one of the paper because the observation is taken as arbitrary. In particular, the setting of the paper does not subsume the setting of finite partial monitoring. Could you explain why the restriction is necessary for the methods of this paper to work ?
- The concurrent work of IDS for Linear Partial Monitoring doesn’t require a finite action space for the regret bounds to hold. Is it possible to extend the methods of this paper beyond the finite action space case ?

**Ethical Concerns:**

["NO or VERY MINOR ethics concerns only"]

**Final Justification:**

My questions have been properly adressed by the authors. I remain confident in my good evaluation of the paper.

**Limitations:**

yes

**Paper Formatting Concerns:**

Nothing to report.

**Quality:**

3

**Strengths And Weaknesses:**

The paper is well written and easy to follow. The main ideas behind the algorithm are clearly explained and the mathematical derivations are well presented and shed some more lights on the results. The dependencies in the regret bounds are clear and the instantiation of the regret bound on different setting of interest shows that this method recovers optimal or almost optimal rates in a variety of settings.

As I’m not as familiar as the authors with the litterature on that topic, I found the comparison with existing work a bit hard to follow, in particular, what are the known upper and lower bounds for the regret in the stochastic and adversarial setting, are the proposed algorithms implementable.
One of the main properties of the proposed algorithm is that it is implementable, it would have been good to either have some experiments to illustrate that or at least a clear analysis of the computational complexity of the algorithm in one of the settings of interest.

---

> ### Author Rebuttal · Authors · 2025-07-31
>
> We thank the reviewer for their feedback and positive comments. We provide below the requested clarifications.
>
> > As I’m not as familiar as the authors with the litterature on that topic, I found the comparison with existing work a bit hard to follow, in particular, what are the known upper and lower bounds for the regret in the stochastic and adversarial setting, are the proposed algorithms implementable. One of the main properties of the proposed algorithm is that it is implementable, it would have been good to either have some experiments to illustrate that or at least a clear analysis of the computational complexity of the algorithm in one of the settings of interest.
>
> To expand upon the discussion in the introduction, the state of the art results in the stochastic setting are those provided in [24,25], which adapt the classification theorem from the finite partial monitoring setting. This provides the optimal rates in $T$ (at least for finite actions) in locally and globally observable games; moreover, their upper bounds also feature constants that depend on the finer structure of the game, though their optimality is not studied. In general, their bounds are not comparable with ours (in the adversarial setting) since neither problem directly subsumes the other (see Paragraphs 147-153 and 343-347). Nevertheless, our bounds exhibit a degree of similarity with theirs; for instance, roughly speaking, Theorem 8 in [25] reports a rate of order $r \sqrt{\alpha T}$, where $r$ is analogous to $\mathrm{rank}(\mathbf{M})$ in our setting (but also takes into account the structure of the loss space) and $\alpha$ is a game-dependent constant that can be expressed in a similar form to our $\beta^2_{\mathrm{loc}}$ via their Lemma 5 as we mention at the end of Section 4. An analogous comparison can also be made in globally observable games.
>
> In the adversarial setting, the problem studied in [30] subsumes ours, but their bounds are very generic; they are stated either in terms of $\Lambda^*$ (analogously to Proposition 6) or in terms of the information ratio, both of which remain opaque as to their dependence on the structure of the game. Moreover, thanks to our more structured setting, our EXO algorithm is amenable to efficient implementation, as discussed in Appendix F, where it is illustrated how the optimization problem that we need to solve simplifies in many common scenarios.
>
> > It is stated that the setting studied in [30] is more general than the one of the paper because the observation is taken as arbitrary. In particular, the setting of the paper does not subsume the setting of finite partial monitoring. Could you explain why the restriction is necessary for the methods of this paper to work ?
>
> Despite our linear observation structure, finite partial monitoring can still be modeled as a special case following Example 7 in [25], where $d$ is taken as the number of outcomes. However, this requires enforcing a certain structure on the loss space; namely, it would no longer be full-dimensional, and hence the observability conditions we consider in this work (also used in [24]) are no longer adequate, see Example 8 in [25] for a simple instance illustrating this. Properly addressing finite partial monitoring within the linear framework requires exploiting the possible low dimensionality of the loss space (as done by [25] in the stochastic setting), which we did not focus on in this work.
>
> > The concurrent work of IDS for Linear Partial Monitoring doesn’t require a finite action space for the regret bounds to hold. Is it possible to extend the methods of this paper beyond the finite action space case ?
>
> A direct extension of our tools (through discretization, or through a direct generalization of our EXO algorithm and its analysis) would likely be inadequate for continuous actions spaces in general; it is shown in [24] (in the stochastic setting) that the observability conditions alone are no longer enough to characterize the optimal rates as the shape of the action set matters as well. In fact, aside from special cases, the general case of compact action sets is not well understood even in the stochastic setting (i.e., a full classification theorem is lacking), and remains thus an interesting problem.

---

> > ### Comment · Reviewer_bC8o · 2025-08-07
> >
> > Thanks for the clarification on the state of the art results and clarifying that some of the results are not comparable given the different settings.
> > Thanks for the clarification on why the observability conditions considered in this work are not compatible direct application to finite partial monitoring.
> > Indeed the compact action case seems to be an interesting case for future work.
> > The authors have answered my questions and adressed my concerns and I maintain my positive view of the paper.

---

### Official Review · Reviewer_5qSe · 2025-07-05

**Clarity:** 2
**Significance:** 3
**Originality:** 2
**Rating:** 4
**Confidence:** 3

**Summary:**

The paper considers the adversarial linear partial monitoring setting. The authors provide an efficient algorithm for this setting, based on the principle of exploration-by-optimization (EXO). The authors show that the regret bounds of the proposed algorithm are near-optimal in the globally and locally observable games.

Prior works, e.g. Lattimore and Szepesvari [2020], only provided efficient and optimal algorithms for the finite partial monitoring setting (the current paper extends this to the linear setting). Existing results on linear partial monitoring by Lattimore and Gyorgy [2021] involved an inefficient version of EXO.

**Questions:**

See questions in the strengths and weaknesses.

**Ethical Concerns:**

["NO or VERY MINOR ethics concerns only"]

**Limitations:**

yes

**Paper Formatting Concerns:**

No concerns.

**Quality:**

3

**Strengths And Weaknesses:**

Strenghts:
- The paper provides a unified algorithm for the linear partial monitoring setting, and shows that this generalizes many existing setting actively researched.
- The paper is written in a mathematically rigorous way.

Weaknesses:
- The main concern I have with paper is with the writting. The paper is rather dense with definitions, and authors do not always provide sufficient intuition. If I were to rewrite the paper, I would perhaps include less results in the main body, but then provide more intuition and proof sketch in the main.
- The authors also do not spend enough time highighting the deltas to prior work. Sure, the paper provides the first efficient algorithm with near-optimal regret for the linear partial monitoring setting, and this setting encopasses previous settings such as full information, linear bandit, graph feedback, etc. But are there existing efficient algorithm for these setting not based on EXO? Also, the authors should also highlight any novelties in the analysis of the algorithm, and where they were able to overcome prior hurdles.

Some other comments:
- The notation in the first paragraph is not consistent with the notation in the rest of the paper.
- Line 20 says: "All of these components are known to the learner,", which suggestes L is know, but then L is unobserved.
- Line 136 claims that (3) implies (2) for $C = \mathbb{R}^d$. Is this true? From its definition, the set $\mathcal{P}(\mathbb{R}^d)$ should only contain points in $\mathrm{ext}(\mathrm{co}(\mathcal{X}))$, whereas (2) invloves points for all $a,b\in \mathcal{A}$.
- Does the set of neighboors $N_{\mathcal{G}}(a)$ include the set itself?

---

> ### Author Rebuttal · Authors · 2025-07-31
>
> We thank the reviewer for their comments and suggestions. We address below the raised concerns.
>
> > The main concern I have with paper is with the writting. The paper is rather dense with definitions, and authors do not always provide sufficient intuition. If I were to rewrite the paper, I would perhaps include less results in the main body, but then provide more intuition and proof sketch in the main
>
> We recognize that the presentation is dense; however, we do devote a significant portion of the main body to developing intuition regarding our adopted approach and its analysis. In particular, the progression in Section 3 is meant to motivate and explain the EXO approach, justify our specific choice of (anchored) loss estimator (compared to the more general machinery of [30]), and illustrate how it ultimately gives rise to the optimization problem featured in Algorithm 1 and analyzed in the rest of the paper. Further, the bulk of Sections 4 and 5 is devoted to outlining the key steps and hurdles in the analysis, highlighting how the linear structure of the problem can be exploited subject to the observability conditions. Finally, the example problems that we discuss motivate the general setting and help interpret the results. Nonetheless, with more space available, we would be able to expand the explanations further in the revised version.
>
> > The authors also do not spend enough time highighting the deltas to prior work. Sure, the paper provides the first efficient algorithm with near-optimal regret for the linear partial monitoring setting, and this setting encopasses previous settings such as full information, linear bandit, graph feedback, etc. But are there existing efficient algorithm for these setting not based on EXO? Also, the authors should also highlight any novelties in the analysis of the algorithm, and where they were able to overcome prior hurdles.
>
> Efficient, specialized algorithms do indeed exist for the examples we discuss; however, our aim in this work is not improving upon these algorithms in their respective problems. Our aim, rather, is to present a unified and efficient algorithm and to show that it enjoys interpretable (in terms of the dependence on the problem structure) regret guarantees in our general setting. As a testament to the significance of these guarantees, we show that they recover tight bounds across a wide spectrum of well-known special cases; in particular, we show that our generic game-dependent factors reduce to the `correct' complexity measures in these problems.
>
> Regarding novelties in the analysis, a primary novelty is how we exploit local observability under the adversarial linear partial monitoring framework. This is elaborated on in Section 4, where we highlight the point of departure with the standard analysis of EXO in finite partial monitoring. In particular, the choice of exploration distribution in our analysis does not follow the water transfer method of [32, 33]; it constitutes a simpler and more natural manner of redistributing the weights between the actions in this setting, and importantly, it enabled achieving the game-dependent constants featured in our results.
>
> > The notation in the first paragraph is not consistent with the notation in the rest of the paper...Line 20 says: "All of these components are known to the learner,", which suggestes L is know, but then L is unobserved.
>
> The notation used in the beginning describes the partial monitoring framework in full generality, subsuming our setting and that of finite partial monitoring. Indeed, $L$, as defined there, is known to the learner. It describes how the loss is determined by the learner's action and the environment's (unobserved) outcome. In our setting, the outcome space corresponds to $\mathcal{L} \in \mathbb{R}^d$ (as mentioned in Lines 97 and 98), such that for $a \in \mathcal{A}$ and $\ell \in \mathcal{L}$, $L(a,\ell) = \psi_a^\top \ell$. This can be clarified further in the revised version of the manuscript.
>
> > Line 136 claims that (3) implies (2) for $C = \mathbb{R}^d$. Is this true? From its definition, the set $\mathcal{P}(\mathbb{R}^d)$ should only contain points in $\mathrm{ext}(\mathrm{co}(\mathcal{A}))$, whereas (2) invloves points for all $a,b \in \mathcal{A}$.
>
> $\mathcal{P}(\mathbb{R}^d)$ contains all actions since $\boldsymbol{0} \in \mathbb{R}^d$ and $\mathcal{P}(\boldsymbol{0}) = \mathcal{A}$. We also note (as mentioned in the paper) that this formulation of the observability conditions is due to [24].
>
> > Does the set of neighboors $N_{\mathcal{G}}(a)$ include the [action] itself?
>
> Not necessarily, if the reviewer is referring to the graph feedback problem. In the composite graph feedback problem, however, we do assume that every action has a self-loop as mentioned in Line 180.

---

> > ### Comment · Reviewer_5qSe · 2025-08-08
> > **Response**
> >
> > I thank the authors for their clarifications and encourage them to make the suggested changes. I will maintain my score.

---

### Decision · Program_Chairs · 2025-09-17

**Decision:**

Accept (poster)

**Comment:**

This paper considers the linear partial monitoring problem under the adversarial environment. The approach of exploration-by-optimization (EXO) is extended to the linear setting and an instance-dependent regret bounds (in the sense of the adversarial setting) are derived for locally- and globally-observable games.

While the reviewers appreciated the theoretical importance of the result, they also raised similar concerns on the presentation and the relation with existing work. I agree with these opinions; while it is somewhat unavoidable that the explanation on partial monitoring tends to notation-heavy due to its inherent complexity and generality, I don't think that the proposed adaptation of EXO is well-positioned in comparison with those for other settings. In particular, while I understand that the approach is different from the existing EXO for partial monitoring with finite feedback symbols in not using the water-transfer operator, the essential benefit of the proposed approach does not seem to be formally discussed enough. Still, the reviewers also shared the opinion that there is enough contribution in the clear characterization of the EXO for linear partial monitoring and the derived bound, based on which I determined to recommend acceptance. I expect that the authors carefully improve the manuscript addressing the raised concerns in the final version.

The writing issue I found is the wrong references. Some references are not cited in the main body. Nevertheless, the supplementary material has another reference list, while the reference numbers for papers are not the same between the main body and the supplementary. Please also correct this issue.